# Evolution of $^{231}$Pa and $^{230}$Th in overflow waters of the North Atlantic

Feifei Deng[1], Gideon M. Henderson[1], Maxi Castrillejo[2,3], Fiz F. Perez[4] and Reiner Steinfeldt[5]

[1]Department of Earth Sciences, University of Oxford, South Parks Road, Oxford, OX13AN, UK.

[2]LaboraLaboratory of Ion Beam Physics, ETH-Zurich, Otto Stern Weg 5, Zurich, 8093, Switzerland.

[3]Institut de Ciència i Tecnologia Ambientals & Departament de Física, Universitat Autònoma de Barcelona, Bellaterra, 08193, Spain.

[4]Departamento de Oceanografía Instituto Investigaciones Marinas (CSIC), Eduardo Cabello 6, E36208 Vigo, Spain.

[5]Institut fur Umweltphysik, Universitat Bremen, D-28334 Bremen, Germany.

*Correspondence to*: Feifei Deng (feifei.deng@earth.ox.ac.uk)

**Abstract.** Many paleoceanographic studies have sought to use the $^{231}$Pa/$^{230}$Th ratio as a proxy for deep ocean circulation rates in the North Atlantic. As yet, however, no study has fully assessed the concentration of, or controls on, $^{230}$Th and $^{231}$Pa in waters immediately following ventilation at the start of Atlantic meridional overturning. To that end, full water-column $^{231}$Pa and $^{230}$Th concentrations were measured along the GEOVIDE section, sampling a range of young North Atlantic deep waters. Th-230 and $^{231}$Pa concentrations in the water column are lower than those observed further south in the Atlantic, ranging

between 0.06 and 12.01 µBq/kg, and between 0.37 and 4.80 µBq/kg, respectively. Both $^{230}$Th and $^{231}$Pa profiles generally increase with water depth from surface to deep water, followed by decrease near the seafloor, with this feature most pronounced in the Labrador Sea (LA Sea) and Irminger Sea (IR Sea). Assessing this dataset using Extended Optimum Multi-Parameter (eOMP) analysis and CFC-based water mass age indicates that the low values of $^{230}$Th and $^{231}$Pa in water near the seafloor of the LA Sea and IR Sea are related to the young waters present in those regions. The importance of water age is

confirmed for $^{230}$Th by a strong correlation between $^{230}$Th and water mass age (though this relationship with age is less clear for $^{231}$Pa and the $^{231}$Pa/$^{230}$Th ratio). Scavenged $^{231}$Pa and $^{230}$Th were estimated and compared to their potential concentrations in the water column due to ingrowth. This calculation indicates that more $^{230}$Th is scavenged (~80%) than $^{231}$Pa (~40%), consistent with the relatively higher particle-reactivity of $^{230}$Th. Enhanced scavenging for both nuclides is demonstrated near the seafloor in young overflow waters. Calculation of meridional transport of $^{230}$Th and $^{231}$Pa with this new GEOVIDE dataset

enables a complete budget for $^{230}$Th and $^{231}$Pa for the North Atlantic. Results suggest that net transport southward of $^{230}$Th and $^{231}$Pa across GEOVIDE is smaller than transport further south in the Atlantic, and indicates that the flux to sediment in the North Atlantic is equivalent to 96% of the production of $^{230}$Th, and 74% of the production for $^{231}$Pa. This result confirms a significantly higher advective loss of $^{231}$Pa to the south relative to $^{230}$Th and supports the use of $^{231}$Pa/$^{230}$Th to assess meridional transport at a basin scale.

**Key words.** GEOTRACES; water-column $^{230}$Th and $^{231}$Pa; water mass ageing; scavenging; meridional transport.

# 1 Introduction

Several paleoceanographic proxies have been proposed that rely on the $^{231}$Pa/$^{230}$Th ratio in marine sediments, one of which is that $^{231}$Pa/$^{230}$Th may record the rate of deep-water circulation, particularly in the North Atlantic. Both $^{231}$Pa and $^{230}$Th are produced in seawater at a constant rate by decay of uranium, but have decay activities much lower than their parent uranium isotopes due to rapid removal by adsorption onto sinking marine particles. Both nuclides are also reversibly scavenged, leading to particularly low concentrations at the surface and increasing concentrations with depth (Nozaki et al., 1981). Advection of surface waters to depth transports water with low concentrations of $^{231}$Pa and $^{230}$Th into the deep ocean, where their concentrations subsequently increase towards an equilibrium value at a rate dependant on the residence time of the nuclide. The longer residence time of $^{231}$Pa relative to $^{230}$Th (~130 years versus ~20 years, Henderson and Anderson, 2003) means that the equilibrium concentration of $^{231}$Pa is closer to that expected from uranium decay, and that the time taken to reach this equilibrium is longer.

This oceanic behaviour of $^{231}$Pa and $^{230}$Th suggests that their measurement in marine sediments may reveal information about the past environment, with one common use being as a recorder of deep-water circulation, particularly in the North Atlantic (e.g. Gherardi et al., 2005, 2009; McManus et al., 2004; Roberts et al., 2014; Yu et al., 1996). The interpretation of sedimentary $^{231}$Pa/$^{230}$Th ratios for such past ocean circulation is based on two end-member conceptual models:

**Basin-scale Advection:** The longer residence time of $^{231}$Pa than $^{230}$Th means that deep-water contains more $^{231}$Pa than $^{230}$Th relative to production from decay. Advection of deep-waters out of the North Atlantic therefore removes more $^{231}$Pa than $^{230}$Th, leaving sediments in the basin with a $^{231}$Pa/$^{230}$Th ratio below the production ratio. If deep-water ventilation ceases, $^{231}$Pa removal from the North Atlantic also ceases, and sedimentary $^{231}$Pa/$^{230}$Th values reach their production ratio. This approach was first proposed by Yu et al. (1996) who measured $^{231}$Pa/$^{230}$Th in Holocene and Last Glacial Maximum (LGM) sediments from many core-top samples from the Atlantic and Southern Ocean. They found similar Holocene and LGM values at a basin scale, suggesting broadly similar overturning during the two periods. Subsequent application to sediments from Heinrich Stadial 1, initially in a single core (McManus et al., 2004) and progressively a geographical range of cores (Bradtmiller et al., 2014), revealed reduced advection of $^{231}$Pa out of the basin at that time, suggesting decreased overturning.

**Water-mass Evolution:** The longer residence time of $^{231}$Pa means that, following ventilation, it takes longer for deep-water $^{231}$Pa concentrations to reach equilibrium with respect to scavenging than is the case for $^{230}$Th. This leads to a systematic evolution of $^{231}$Pa/$^{230}$Th with age of the water. Sediments capture this ratio (with a fractionation due to different scavenging coefficients for the two nuclides), so capture information about the age of the water. Simple models suggest an increase of $^{231}$Pa/$^{230}$Th with age over about 400 years (e.g. several residence times of $^{231}$Pa). This approach to interpreting sedimentary $^{231}$Pa/$^{230}$Th allows for the possibility of calculating flow rates for a single water mass and from a single core rather than at a basin scale. It has been pursued by (Negre et al., 2010) to asses deep-water flow in both southerly and northerly directions by comparing sediments in the North and South Atlantic, and allowed these authors to apply a simple model (Thomas et al., 2007) to calculate flow rates.

Recent water-column measurements of $^{231}$Pa and $^{230}$Th on GEOTRACES cruises shed new light on the chemical behaviour and controls on these isotopes in seawater and provided evidence to assess the validity of the models underlying the use of sedimentary $^{231}$Pa/$^{230}$Th as a proxy for deep-water circulation. These measurements have indicated that there is considerably more net advection of $^{231}$Pa than $^{230}$Th out of the North Atlantic (Deng et al., 2014), supporting the Basin-scale Advection

model for $^{231}$Pa/$^{230}$Th. But these measurements also have suggested that there is no simple relationship between increasing $^{231}$Pa/$^{230}$Th and age of water, as would be expected for the Water-mass Evolution model (e.g. Deng et al., 2014). Studies using 2-D and 3-D ocean models (e.g. Marchal et al., 2010; Siddall et al., 2007) have also supported the use of sedimentary $^{231}$Pa/$^{230}$Th to constrain deep-water circulation at a basin scale, and suggested that the relationship between $^{231}$Pa/$^{230}$Th and water mass age is more complex than assumed in earlier studies (e.g. Luo et al., 2010).

Observations and model studies of $^{231}$Pa and $^{230}$Th have also suggested that other controls complicate $^{231}$Pa/$^{230}$Th as a dynamic tracer of deep-water circulation, such as the effect of boundary scavenging at seafloor and ocean margins (e.g. Anderson et al., 1994; Deng et al., 2014; Rempfer et al., 2017) and the influence of particle flux and composition (e.g. Chase et al., 2002; Hayes et al., 2014; Siddall et al., 2005).

To fully assess the behaviour of $^{231}$Pa/$^{230}$Th, and its potential as a dynamic tracer of deep-water circulation, knowledge of the

concentrations and variations of these isotopes as deep waters form and enter the deep Atlantic is required. Some measurements have placed initial constraints on $^{231}$Pa and $^{230}$Th values in young North Atlantic deep waters (e.g. Moran et al., 1997, 2002; Rutgers van der Loeff and Berger, 1993), but there has not yet been a systematic study of the composition of waters in the far north Atlantic. The GEOVIDE cruise allowed waters to be collected for such a study, along a line where significant other data are available, both from that cruise and previous occupations of OVIDE. GEOVIDE provided an ideal

opportunity to understand $^{231}$Pa and $^{230}$Th at the start of the ocean meridional overturning circulation, and to assess the hypotheses underlying the use of $^{231}$Pa/$^{230}$Th as a paleo-proxy for the rate of deep-water circulation.

## 2 Sampling strategies and analytical methods

Seawater samples were collected during the GEOVIDE cruise aboard the R/V Pourquoi Pas? from 15 May to 30 June, 2014 as part of GEOTRACES Section GA01. The cruise sampled four regions in the North Atlantic between 40º-60ºN: Labrador

Sea (LA Sea), Irminger Sea (IR Sea), Iceland Basin (IC Basin), and Western European Basin (WE Basin) (Fig. 1).

Full-depth water-column $^{231}$Pa and $^{230}$Th for this study were collected from 11 stations (Fig. 1). Sampling followed the procedure suggested by GEOTRACES intercalibration work (Anderson et al., 2012). Briefly, seawater samples of 5 Litres were directly filtered from Niskin bottles mounted on the Stainless Steel CTD Rosette through AcroPak$^{TM}$ capsules with Supor$^{®}$ Membrane (0.45 μm pore size). Filtered seawater samples were collected into acid cleaned HDPE plastic bottles, and

sealed with a screw cap and Parafilm to reduce evaporation and contamination. Samples were then double bagged for storage in boxes for transport back to the shore-based lab for analysis.

Once returned to the laboratory in Oxford, samples were weighed and then acidified with quartz distilled concentrated HCl to pH ~1.7, shaken and left for at least four days to ensure that Pa and Th was desorbed from the walls of the bottle. A mixed $^{229}$Th-$^{236}$U spike and a $^{233}$Pa spike were then added to each sample to allow measurement of Th, U (for another study), and Pa by isotope dilution MC-ICP-MS (Multi Collector-Inductively Coupled Plasma-Mass Spectrometry). The $^{233}$Pa spike was freshly made by milking from $^{237}$Np (following Regelous et al., 2004) and calibrated against a known $^{236}$U solution after complete decay of $^{233}$Pa to $^{233}$U, i.e. four to five half-lives of $^{233}$Pa ($t_{1/2}$=26.98 days, Usman and MacMahon, 2000) after spike production (Robinson et al., 2004). 50 mg of pure Fe as a chloride solution was also added to each water sample. Samples were left overnight to allow for spike equilibrium after which the pH was raised to ~8.5 using distilled NH$_4$OH to co-precipitate the actinides with insoluble Fe-oxyhydroxides. At least 48 hours were allowed for scavenging of the actinides onto Fe-oxyhydroxides. The precipitate was centrifuged and rinsed, and Th, Pa and U were separated using anion exchange chromatography following Thomas et al. (2006).

After chemical separation, Pa and Th were measured on a Nu instrument MC-ICP-MS at the University of Oxford. Mass discrimination and ion-counter gain were assessed with the measurement of a U standard, CRM-145 U, before each sample measurement. Use of a U standard for this purpose minimises memory problems that might be caused by use of a Th or Pa standard (Thomas et al., 2006). Measurements were also made 0.5 mass units either side of masses of interest to allow accurate correction for the effect of abundance sensitivity on small $^{231}$Pa and $^{230}$Th beams, and a correction for a small $^{232}$ThH interference on the $^{233}$Pa beam is made from assessment of the hydride formation rate on a $^{232}$Th standard. Concentrations of $^{231}$Pa, $^{230}$Th together with $^{232}$Th were obtained from the precise MC-ICP-MS measurement of $^{231}$Pa/$^{233}$Pa, $^{230}$Th/$^{229}$Th, and $^{232}$Th/$^{229}$Th ratios together with well-calibrated concentrations of $^{233}$Pa, and $^{229}$Th-$^{236}$U spikes.

Chemistry blanks were assessed by conducting the complete chemical procedure on ~100 ml of Milli-Q water with each batch of samples. Based on six blank measurements, the average blanks for dissolved $^{231}$Pa, $^{230}$Th and $^{232}$Th are 0.21±0.14 fg, 1.59±0.60 fg and 5.13±1.47 pg, respectively (uncertainties are 2 standard errors). Blank contributions account for 2-22%, 2-26%, and 0.2-16% of the dissolved $^{231}$Pa, $^{230}$Th and $^{232}$Th respectively (with the higher values being for surface samples due to their low concentrations).

**3 Results**

Measured $^{230}$Th and $^{231}$Pa concentrations were corrected for blanks, ingrowth from U in seawater since the time of sample collection, and detrital U-supported $^{230}$Th and $^{231}$Pa concentrations. Measured and corrected concentrations of $^{230}$Th, $^{231}$Pa, and $^{232}$Th, along with details of corrections, are provided in the Supplemental Information S1. Although analysis was conducted in terms of fg/kg, results are converted to the SI units adopted by GEOTRACES data product, i.e., µBq/kg for $^{230}$Th and $^{231}$Pa, and pmol/kg for $^{232}$Th. This conversion uses half-lives for $^{231}$Pa, $^{230}$Th and $^{232}$Th of 32,760 yr, 75,584 yr and 1.405×10$^{10}$ yr, respectively (Cheng et al., 2013; Holden, 1990; Robert et al., 1969). Uncertainties were propagated, including the contribution from sample weighing, spike calibration, impurities in the spikes, blank corrections, and mass spectrometric measurement,

and are reported as 2 standard errors (2 s.e.). Average total uncertainties for $^{231}$Pa, $^{230}$Th and $^{232}$Th are ±0.17 μBq/kg, ±0.17 μBq/kg, and ±0.0032 pmol/kg, respectively. Vertical profiles showing the results of corrected $^{230}$Th and $^{231}$Pa concentrations in the water column are plotted by region in Fig. 2.

Th-230 concentrations in the water column range between 0.06 and 12.01 μBq/kg, and initially generally increase with water depth from surface to deep water. Towards the seafloor, six of the eleven stations show a prominent decrease of $^{230}$Th, with this feature most pronounced in the LA and IR Seas.

Pa-231 concentrations in the water column range between 0.37 and 4.80 μBq/kg and also increase with water depth, but less rapidly than $^{230}$Th. $^{231}$Pa profiles also often exhibit a decrease near the seafloor at stations showing a $^{230}$Th decrease. Station 38 at the Reykjanes Ridge distinguishes itself from other $^{231}$Pa profiles in that an increase in $^{231}$Pa concentrations from low concentrations at 1000 m is observed, continuing towards the bottom.

Observed $^{230}$Th and $^{231}$Pa values at GEOVIDE are lower than those observed in inter-calibrated GEOTRACES data from further south in the Atlantic. Figure 3 compares average depth profiles for $^{230}$Th and $^{231}$Pa in the west Atlantic, covering high-latitude Northwest Atlantic (from GEOVIDE, west of the Mid-Atlantic Ridge), mid-latitude Northwest Atlantic (GEOTRACES section GA03_w, Hayes et al., 2015) and Southwest Atlantic (GEOTRACES section GA02, Deng et al., 2014). A southward increase of both $^{230}$Th and $^{231}$Pa concentrations is observed below 1000 m.

## 4 Discussion

Early studies of water-column $^{230}$Th and $^{231}$Pa reported a linear increase of both nuclides with water depth (e.g., Anderson et al., 1983b; Nozaki et al., 1981), and introduced a reversible scavenging model with exchange of both nuclides between their dissolved and particulate phases. Later studies observed a deviation of $^{230}$Th and $^{231}$Pa profiles from this reversible scavenging model, with the expected increase with depth often inverting near the seafloor (e.g., Anderson et al., 1983a; Bacon and Anderson, 1982). This feature has been further investigated in more recent studies. Rutgers van der Loeff and Berger (1993) observed that $^{230}$Th concentrations decrease in the bottom water in the South Atlantic south of the Antarctic Polar Front and interpreted this as the influence of relatively young bottom water in the region. Okubo et al. (2012) also found decreasing $^{230}$Th values near the seafloor in the North Pacific and, in the absence of ventilation in the area, interpreted these as due to bottom scavenging. Deng et al. (2014) observed low concentrations of both $^{230}$Th and $^{231}$Pa in near-bottom water coinciding with the presence of the nepheloid layer, and interpreted the low values as a result of enhanced scavenging by resuspended particles in the nepheloid layer.

In this study, recently ventilated overflow waters are sampled at depth, particularly in the Labrador and Irminger Seas. Low values of $^{230}$Th and $^{231}$Pa near the seafloor might be expected to relate to these young waters, but the effects of scavenging must also be considered.

### 4.1 Water mass distribution and influence

The presence of multiple water masses sampled by the GEOVIDE Section allows the influence of water mass (and age) on $^{230}$Th and $^{231}$Pa to be assessed. Extended Optimum Multi-Parameter (eOMP) Analysis (García-Ibáñez et al., 2018) for the GEOVIDE section maps the presence of 10 water-mass end-members in the section (Fig. 4), including three recently ventilated waters in the GEOVIDE section:

*i.* Labrador Sea Water (LSW), which is formed by deep convection (Talley and McCartney, 1982), is the dominant deep water along the section, extending from 1000 to 2500 m depth in the east and from surface to 3500 m in the west of the section.

*ii.* Iceland–Scotland Overflow Water (ISOW), which is formed in the Norwegian Sea and subsequently entrains overlying warmer and more salty waters. This water mass initially flows along the eastern flank of the Reykjanes Ridge before spreading back northwards, after crossing the Charlie-Gibbs Fracture Zone, into the Irminger and Labrador Seas (Dickson and Brown,

1994; Saunders, 2001). A pronounced layer of this water mass is observed immediately below the LSW, and extends as deep as 4000 m west of 20°W.

*iii.* Denmark Strait Overflow Water (DSOW), which is formed after the Nordic Seas deep waters overflow and entrains Atlantic waters (SPMW and LSW) (Yashayaev and Dickson, 2008) with dense Greenland shelf water cascading down to the DSOW layer in the Irminger Sea (Falina et al., 2012; Olsson et al., 2005; Tanhua et al., 2005). This water occupies the deepest

part of the IR and LA Seas.

In the east of the section, deep waters consist of the much older Lower North East Atlantic Deep Water (NEADW$_L$) which is formed with a significant southern component from Antarctic Bottom Water. A number of other water masses are also observed at shallow depths, including Mediterranean Water, and various mode waters.

Some control of water mass on $^{230}$Th and $^{231}$Pa concentration is evident in nuclide section plots (Fig. 5), particularly relatively

low $^{230}$Th and $^{231}$Pa concentrations in DSOW and high values in the old NEADW. In other places, the impact of water mass is less apparent. The challenge with these nuclides is that they are not conservative tracers of water mass, but evolve significantly during transport and water aging. On the GEOVIDE Section we can analyse this evolution because the ages of the water-masses can be assessed from CFC data.

CFC measurements are not available from the GEOVIDE cruise itself. de la Paz et al. (2017), however, measured CFC

concentrations along the east of the same section (covering WE Basin, IC Basin, and IR Sea) in 2012 (OVIDE/CATARINA cruise). This allowed the computing of CFC-based age with the Transit Time Distribution (TTD) method. Using the water mass distribution along GEOVIDE given by García-Ibáñez et al. (2018) and the distribution for the same water masses in 2012 (García-Ibáñez et al., 2015), we derived CFC-based ages for GEOVIDE waters (Fig. 6; further details in Supplemental Information S2). Uncertainties (1 standard error) associated with CFC-based age calculated with this approach range between

30   11-40%.

CFC-based water-mass ages range from ≈10 years, observed in DSOW at the bottom of the LA Sea, to ≈800 years, observed for NEADW at the bottom of the WE Basin. Because this study focuses on understanding controls on $^{231}$Pa and $^{230}$Th in recently ventilated waters, we omit detailed consideration of the upper 1km in subsequent discussion, and restrict our analysis to water sampled west of 35°W of the section where young waters (<50 years) dominate. A rescaled version of the CFC age

section indicates the variation in age of ventilated waters (Fig. 6b). DSOW, occupying the deepest LA and IR Seas, is the youngest water mass in this region, with an average age of ~19 years. ISOW and LSW are slightly older, with ages ranging from 26 to 45 years and 32 to 40 years respectively.

## 4.2 Evolution of $^{230}$Th and $^{231}$Pa with water age

5 The presence of recently ventilated deep-waters with constrained CFC ages allows analysis of the rates at which $^{230}$Th and $^{231}$Pa concentrations increase during transport, and the rates of scavenging of these nuclides. To conduct this analysis, we define five components in the budget of $^{230}$Th and $^{231}$Pa:

*i. Preformed component:* The $^{230}$Th or $^{231}$Pa transported from the surface into the interior. For this analysis, in the absence of measurements for the exact location of deep-water formation during winter convection, we assume the same preformed value

10 for all water masses and set this as the average of concentrations measured in surface waters <100 m depth along GEOVIDE section. This gives preformed concentrations of 1.66 µBq/kg for $^{230}$Th and 1.31 µBq/kg for $^{231}$Pa. We recognise that true preformed values may differ from these values and between water masses, and discuss the implications of uncertainty in preformed values in the following section. Preformed $^{230}$Th and $^{231}$Pa will decrease due to radioactive decay during transport. Although we take this decay into account in the following analysis, it is insignificant given the ages of waters involved and

15 the much longer half-lives of $^{230}$Th and $^{231}$Pa.

*ii. Ingrown component*: The ingrown $^{230}$Th or $^{231}$Pa from radioactive decay of U since the water was last in contact with the surface. This component increases as the water mass ages. The concentration of this component in a water mass of age t can be calculated as:

$$^{230}Th_{Ingrown} = {}^{234}U \times (1 - e^{-\lambda_{230}t}) \qquad (4.1)$$

$$^{231}Pa_{Ingrown} = {}^{235}U \times (1 - e^{-\lambda_{231}t}) \qquad (4.2)$$

where $^{230}$Th$_{Ingrown}$ and $^{231}$Pa$_{Ingrown}$ are the $^{230}$Th and $^{231}$Pa ingrown from their U parents, respectively; $^{234}$U and $^{235}$U are activities of $^{234}$U, and $^{235}$U in seawater (45551.2 µBq/kg (2801.4 dpm/1000l) and 1823.8 µBq/kg (112.2 dpm/1000l), respectively, assuming a constant seawater $^{238}$U activity of 39609.8 µBq/kg (2436 dpm/1000l) at salinity 35 psu, and seawater $^{234}$U/$^{238}$U activity ratio of 1.15 and natural $^{238}$U/$^{235}$U abundance ratio of 137.88); $\lambda_{230}$ and $\lambda_{231}$ are decay constants of $^{230}$Th and $^{231}$Pa

25 (9.17 ×10$^{-6}$ yr$^{-1}$ and 2.12×10$^{-5}$ yr$^{-1}$, respectively).

*iii. Potential Total component*: The $^{230}$Th and $^{231}$Pa expected in the water due to the combination of preformed and ingrown components, if there were no removal by scavenging.

*iv. Observed component*: The $^{230}$Th and $^{231}$Pa observed in the water column, i.e. dissolved $^{230}$Th and $^{231}$Pa measured in this study (after correction for detritus and ingrowth from U since sample collection).

*v. Scavenged component*: The net $^{230}$Th or $^{231}$Pa removed from the water since it left the surface due to scavenging. For each depth, this component is the net of nuclide added from above by desorption from settling particles, and the removal downwards by scavenging.

These components are related to each other as follows:

$$Preformed + Ingrown = Potential\ Total\ ^{230}Th\ (or\ ^{231}Pa) = Observed + Scavenged \qquad (4.3)$$

The difference between the Potential Total and the Observed concentration of $^{230}$Th (or $^{231}$Pa), therefore provides a measure of the amount of nuclide scavenged since the water left the surface (Fig. 7).

We examine the evolution of both the observed and scavenged components of $^{230}$Th and $^{231}$Pa with water mass age (Fig. 8). Both $^{230}$Th and $^{231}$Pa show an increase in observed concentration with age of water, with the increase for $^{230}$Th much more

regular than for $^{231}$Pa. This strong $^{230}$Th relationship, regardless of depth of the sample (Fig. 8a), indicates a primary control of water-mass age on the increase of $^{230}$Th in these younger waters.

For $^{230}$Th, the rate of increase with age (i.e. slope in Fig. 8a) indicates that about one quarter of the $^{230}$Th formed from U decay remains in the water, with the other three quarters being removed by scavenging. This ratio is consistent with the average $^{230}$Th for these waters, which requires that about three times more $^{230}$Th than remains in water has been removed by scavenging

(Fig. 8a, 8b). The scatter between $^{231}$Pa and age (Fig. 8c) precludes the use of the slope to assess the relative proportion of scavenged $^{231}$Pa, but the average values (Fig. 8c, 8d) indicate that about half of the $^{231}$Pa remains in the water, while half is removed by scavenging. The relative behaviour of $^{230}$Th and $^{231}$Pa is consistent with previous expectations, with a higher fraction of scavenging of $^{230}$Th than $^{231}$Pa.

The hypothesis that $^{231}$Pa/$^{230}$Th ratios increase monotonically as water mass ages forms the foundation of the Water-mass

Evolution model for interpretation of sedimentary $^{231}$Pa/$^{230}$Th in terms of the rate of deep water circulation. For these young waters, however, there is no clear relationship between observed $^{231}$Pa/$^{230}$Th and age (Fig. 8e), nor between the $^{231}$Pa/$^{230}$Th value scavenged to the sediment and age (Fig. 8f), calling the Water-mass Evolution model into question.

## 4.3 The importance of preformed $^{230}$Th and $^{231}$Pa in young waters

To assess the controls on $^{230}$Th, $^{231}$Pa, and particularly the resulting $^{231}$Pa/$^{230}$Th ratio, we apply a simple scavenging-mixing

model following Moran et al. (1997). This model was first created to assess the evolution of $^{230}$Th in a 1D water column as it ages following ventilation. Here we adopt it by modelling the nuclide evolution with age for each depth, and by also modelling $^{231}$Pa. This assumes that waters have remained at the same depth since ventilation which, though not correct in detail, still allows the model to provide insights about controls on these nuclides.

Following Moran et al. (1997), dissolved concentration of $^{230}$Th and $^{231}$Pa is given by,

$$c_d = \frac{C_{pre,t} + P\ \ \tau_w}{(K_d\ SPM + 1)} \times \left[1 - \exp\left(-\frac{(K_d\ SPM + 1)}{SK_d\ \tau_w SPM} \times z\right)\right] \quad (4.4)$$

where $C_d$ is the dissolved concentration of the nuclide ; $P$ is the production rate of [230]Th and [231]Pa, 0.42 µBq/kg/yr ($2.57 \times 10^{-2}$ dpm/1000l/yr ) and 0.039 µBq/kg/yr ($2.37 \times 10^{-3}$ dpm/1000l/yr), respectively; $K_d$ is the distribution coefficient of the nuclide; $\lambda$ is the decay constant of the nuclide; $C_{pre,t}$ is the preformed total concentration of [230]Th (or [231]Pa); SPM is the suspended particle concentration; and S is the particle settling speed, which represents the net effect of particle sinking, disaggregation and aggregation; $\tau_w$ is water mass age; z is the water depth.

The model requires values for four parameters: particle settling speed (S), suspended particle concentration (SPM), and distribution coefficients for [230]Th ($K_d^{Th}$) and [231]Pa ($K_d^{Pa}$). We select these parameters to give a good fit to the [230]Th and [231]Pa observations at an open ocean station, Station 13, on the east of the section (i.e. a station sampling older waters, which are close to equilibrium) and use these values to interpret the younger waters to the west. Best fits to Station 13 suggested S = 800 m/yr; SPM = 25 µg/l; $K_d^{Th}$ =$1.1 \times 10^7$ ml/g; and $K_d^{Pa}$ = $1.4 \times 10^6$ ml/g (the first three of these are close to those of Moran et al. (1997)). A fuller description of the model is given in Supplemental Information S3.

We show two sets of output from the model, one with a preformed component (C_pre) equal to the nuclide concentrations observed in the upper 100 m of the GEOVIDE section (as in 4.2 above), and one with the preformed component set to zero for both nuclides. For both cases, the modelled evolution of nuclide concentrations with age between 0-50 years at 2000 m and 3500 m water depths is plotted in Fig. 9, and compared to data. As expected, modeled [230]Th and [231]Pa concentrations increase with age, with deeper waters having higher concentrations and [230]Th increasing more rapidly initially (Fig. 9), but the preformed concentration is seen to be important in setting total nuclide concentration for several decades after ventilation. The fit of the model to observations in young waters from GEOVIDE is improved in the model run with zero preformed nuclide, particularly for [230]Th. This is surprising, given that surface-water [230]Th and [231]Pa values are generally non-zero, and typically close to the value observed in the GEOVIDE surface waters. For [230]Th in young deep waters, even the model with zero preformed nuclide overestimates the observed value, possibly indicating additional scavenging from these waters close to the seafloor, or as a result of differing biological productivity and particle fluxes between stations.

The most striking effect of changing the assumed preformed values in the model is on [231]Pa/[230]Th (Fig. 9c). When preformed values are set at zero, [231]Pa/[230]Th ratios always increase with water age, but when set at the average surface value from GEOVIDE, [231]Pa/[230]Th ratios initially decrease before increasing. The impact of preformed concentrations has a long-lasting impact on water-column and scavenged [231]Pa/-Th, lasting for hundreds of years following ventilation (Supplementary Information Figure S2 (c), (d)). This indicates that knowledge of the nuclide concentration at the site of deep-water formation is critical to understanding the early evolution of [231]Pa/[230]Th in waters and their underlying sediments.

## 4.4 Scavenging of [230]Th and [231]Pa

Knowledge of the CFC ages of the waters analysed on the GEOVIDE cruise allow an assessment of the scavenging rates of [230]Th and [231]Pa. To do so, we compare the Scavenged component to the Potential Total component (as defined in Section 4.2).

The percentage of the Scavenged component relative to the Potential Total component is higher for $^{230}$Th, at an average of 80%, than for $^{231}$Pa at an average of 40% (Fig. 10), consistent with the relatively higher particle-reactivity of $^{230}$Th. For both nuclides, there is a higher fraction of scavenging in samples from near the seafloor, particularly those from DSOW in the deepest LA Sea. Bottom scavenging has been indicated in previous studies (e.g. Bacon and Anderson, 1982; Deng et al., 2014;
Okubo et al., 2012), but this study indicates that this enhanced nuclide scavenging occurs even in the very young overflow waters at the start of the meridional circulation.

## 4.5 Uncertainty analysis of the assessment of the scavenging of $^{230}$Th and $^{231}$Pa

Uncertainty in both CFC-based ages and in preformed values of $^{230}$Th and $^{231}$Pa contribute to uncertainty when calculating the scavenging of $^{230}$Th and $^{231}$Pa. Uncertainties associated with CFC-based age range between 11-40% (1 standard error). This
uncertainty leads to an average uncertainty of 23% and 13% in potential total $^{230}$Th and $^{231}$Pa respectively, corresponding to an average uncertainty of 30% in the scavenged component of $^{230}$Th and 40% in the scavenged component of $^{231}$Pa.
A two-fold increase in preformed values results in an increase by a factor of 1.2 and 1.6 in the total potential $^{230}$Th and $^{231}$Pa, respectively, leading to an increase by a factor of 1.2 in the scavenged component of $^{230}$Th and of 2.6 in the scavenged component of $^{231}$Pa. The impact of preformed uncertainty is less significant when comparing the scavenging to the potential
total components. The ratio of scavenged/potential total increases by a factor of 1.1 and 1.4 for $^{230}$Th and $^{231}$Pa, respectively. This sensitivity analysis indicates that a better knowledge of preformed values will benefit the assessment of the scavenging of both nuclides.

## 4.6 Meridional transport of $^{230}$Th and $^{231}$Pa in the North Atlantic

Previous calculations have indicated removal of $^{230}$Th and $^{231}$Pa from the North Atlantic by meridional transport southward.
Deng et al. (2014) calculated net southward transport of 6% of the $^{230}$Th and 33% of $^{231}$Pa, relative to production of these nuclides in the water column. That calculation, however, did not provide a complete budget for $^{230}$Th and $^{231}$Pa for the North Atlantic because observations at the time did not constrain input of these nuclides from the north. Data in this study allow this calculation, and therefore a more complete budget for the modern North Atlantic.
García-Ibáñez et al. (2018) calculated volume transports for the Portugal to Greenland section of the GEOVIDE section by
combining the water mass fractions from eOMP analysis with the absolute geostrophic velocity field calculated using inverse model constrained by Doppler current profiler velocity measurements (Zunino et al., 2017). They separated northward flowing upper, and southward flowing lower limbs of the AMOC at isopycnal $\sigma_1$ (potential density referenced to 1000 dbar) = 32.15 kg/m$^3$, with +18.7 ± 2.4 Sv and -17.6 ± 3.0 Sv flow across the section above and below this value (positive value indicates northward transport). With average $^{230}$Th and $^{231}$Pa concentrations in the upper limb ($\sigma_1$ < 32.15 kg/m$^3$) of 1.60 and 1.32
μBq/kg respectively, northward transport of $^{230}$Th is $3.07 \times 10^{10}$ and of $^{231}$Pa is $2.53 \times 10^{10}$ μBq/s. Average $^{230}$Th and $^{231}$Pa

concentrations in the lower limb ($\sigma_1 > 32.15$ kg/m$^3$) are 3.44 and 2.07 µBq/kg , respectively, indicating transports of $^{230}$Th and $^{231}$Pa are -6.22 × 10$^{10}$ and -3.74 × 10$^{10}$ µBq/s, respectively.

Net transport of $^{230}$Th and $^{231}$Pa across GEOVIDE is therefore to the south, and supplies 3.15 × 10$^{10}$ µBq/s $^{230}$Th and 1.21 × 10$^{10}$ µBq/s $^{231}$Pa to the North Atlantic (Fig. 11). This is a smaller net transport than further south in the Atlantic (Fig. 11), due to the lower $^{230}$Th and $^{231}$Pa concentrations in the water column close of the site of deep-water formation.

The budget for these nuclides for the North Atlantic consists of:  production in the water column; addition by advection from the North; loss by advection to the South and removal to the sediment.  The data from this study allows this budget to be fully assessed, and indicates that the flux to the sediment is equivalent to 96% of the production of $^{230}$Th, and 74% of the production for $^{231}$Pa (Supplemental Information Table S4).  For both nuclides, these fluxes are higher than in previous calculations (Deng et al., 2014) which ignored advective fluxes from the North.  There is, however, still a significantly higher advective loss of $^{231}$Pa relative to $^{230}$Th.  At a basin scale, therefore, $^{231}$Pa/$^{230}$Th in the sediment must be lower than the production ratio.  This lower value is generated by the meridional transport of the North Atlantic, and likely to be sensitive to changes in this transport. Using the Basin-scale Advection model to interpret sedimentary $^{231}$Pa/$^{230}$Th to assess meridional transport, as initially proposed by Yu et al. (1996), is therefore still supported by a full modern North Atlantic budget for these nuclides.

**5 Conclusion**

Measurement of $^{230}$Th and $^{231}$Pa in waters from GEOVIDE show some control of water mass on $^{230}$Th and $^{231}$Pa concentrations, particularly low concentrations in DSOW and high values in the old NEADW. There is, however, no close mapping of nuclide concentration to water mass.

With the availability of CFC-based ages on this section, the evolution of $^{230}$Th and $^{231}$Pa concentration with age is possible.  A systematic increase of $^{230}$Th concentration is observed over the first 50 years following ventilation, and a similar though more scattered relationship seen for $^{231}$Pa.  There is no clear relationship between the $^{231}$Pa/$^{230}$Th ratio and age for these young waters The long-term evolution of $^{231}$Pa/$^{230}$Th is found from a simple model to be highly dependent on the preformed concentrations for these nuclides. These results complicate the interpretation of sedimentary $^{231}$Pa/$^{230}$Th as a paleo-proxy for deep water circulation based on systematic evolution of water $^{231}$Pa/$^{230}$Th with age, and point to the importance of a better knowledge of preformed $^{230}$Th and $^{231}$Pa concentrations to improve interpretation. This analysis of the $^{230}$Th and $^{231}$Pa concentration relative to the age of the water not only demonstrates the influence of water mass aging on $^{231}$Pa and $^{230}$Th, but also points to the influence of scavenging. Scavenged $^{230}$Th is much more extensive than $^{231}$Pa, as expected, and enhanced removal of both nuclides is seen immediately above the seafloor, particularly for young waters.

Calculation of meridional transport of $^{230}$Th and $^{231}$Pa indicates a southward net transport of both nuclides across the GEOVIDE section. This advection is smaller than that further south in the Atlantic as a result of lower $^{230}$Th and $^{231}$Pa concentrations at GEOVIDE. Calculation of the flux across GEOVIDE allows a more complete budget for the North Atlantic to be constructed and demonstrates a significantly higher advective loss of $^{231}$Pa to the south relative to $^{230}$Th, with 26% of the

$^{231}$Pa produced advected southward (relative to only 4% for $^{230}$Th). This calculation supports the interpretation of sedimentary $^{231}$Pa/$^{230}$Th measurements as a proxy for overturning circulation, when based on advective loss of $^{231}$Pa at a basin scale.

**Data availability.** GEOVIDE $^{231}$Pa, $^{230}$Th and $^{232}$Th data are available at British Oceanographic Data Centre - Natural Environment Research Council, UK: http://doi.org/10.5285/7a150d33-956b-0fec-e053-6c86abc0b35c (Deng et al., 2018).

**Author contributions.** GMH conceptualized and acquired funding for the research. MC collected seawater samples for $^{231}$Pa, $^{230}$Th and $^{232}$Th at sea and provided expertise about other measurements on the cruise. FD conducted the chemical analysis and MC-ICP-MS measurement of $^{231}$Pa, $^{230}$Th and $^{232}$Th. FFP and RS conducted the analysis, calculation and interpretation of CFC ages. FD conducted the interpretation and analysis of $^{231}$Pa, $^{230}$Th and $^{232}$Th data, with extensive contribution from GMH. FD wrote the paper and prepared all figures, with GMH contributing extensively and contributions from other co-authors.

**Competing interests.** The authors declare that they have no conflict of interest.

**Special issue statement.** This article is part of the special issue "GEOVIDE, an international GEOTRACES study along the OVIDE section in the North Atlantic and in the Labrador Sea (GA01)". It is not associated with a conference.

**Acknowledgements**

Géraldine Sarthou and Pascale Lherminier are thanked for leading the GEOVIDE cruise, along with the captain, Gilles Ferrand, and crew of the R/V Pourquoi Pas?. We would like to give a special thanks to Pierre Branellec, Floriane Desprez de Gésincourt, Michel Hamon, Catherine Kermabon, Philippe Le Bot, Stéphane Leizour, Olivier Ménage, Fabien Pérault and Emmanuel de Saint Léger for their technical expertise and to Catherine Schmechtig for the GEOVIDE database management. This work was supported by the French National Research Agency (ANR-13-BS06-0014, ANR-12-PDOC-0025-01), the French National Centre for Scientific Research (CNRS-LEFE-CYBER), the LabexMER (ANR-10-LABX-19), and Ifremer. It was supported for the logistic by DT-INSU and GENAVIR. Thank you also to Yi Tang who helped with sampling during the cruise. We are grateful to Mercedes de la Paz Arandiga, and Pascale Lherminier for providing valuable input during analysis of water masses and ages, and to Maribel García-Ibáñez for early provision of the eOMP analysis presented elsewhere in this volume. We also thank Yves Plancherel and Jerry McManus for valuable insight during discussion of the results presented. We thank Roger Francois, and two anonymous reviewers, and the editor Gilles Reverdin, for their constructive comments on the manuscript.

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

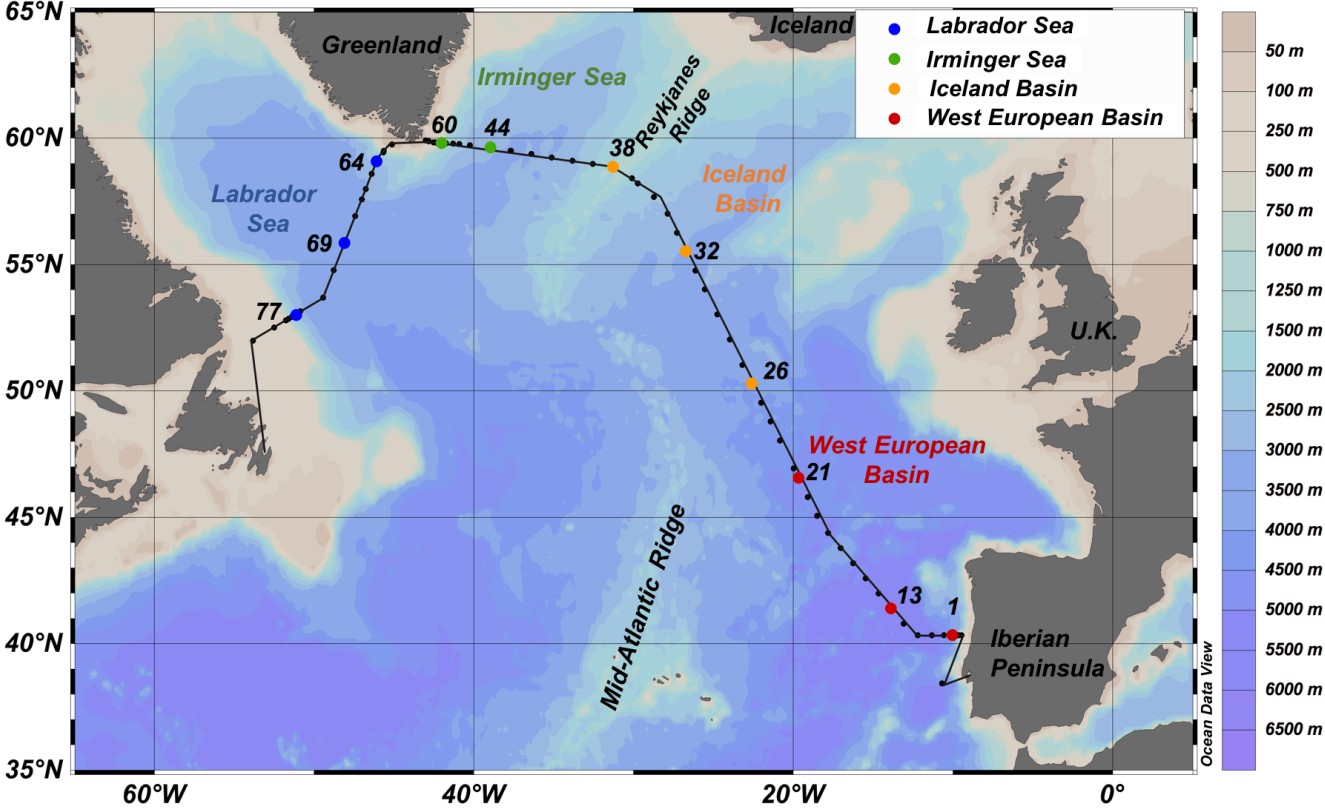

Figure 1: Map showing GEOVIDE cruise track (black line) and station locations (black dots). Colour bars indicate water depth. Sampling locations for water-column $^{231}$Pa and $^{230}$Th in this study are shown by coloured dots, with colours representing the ocean regions they are located in.

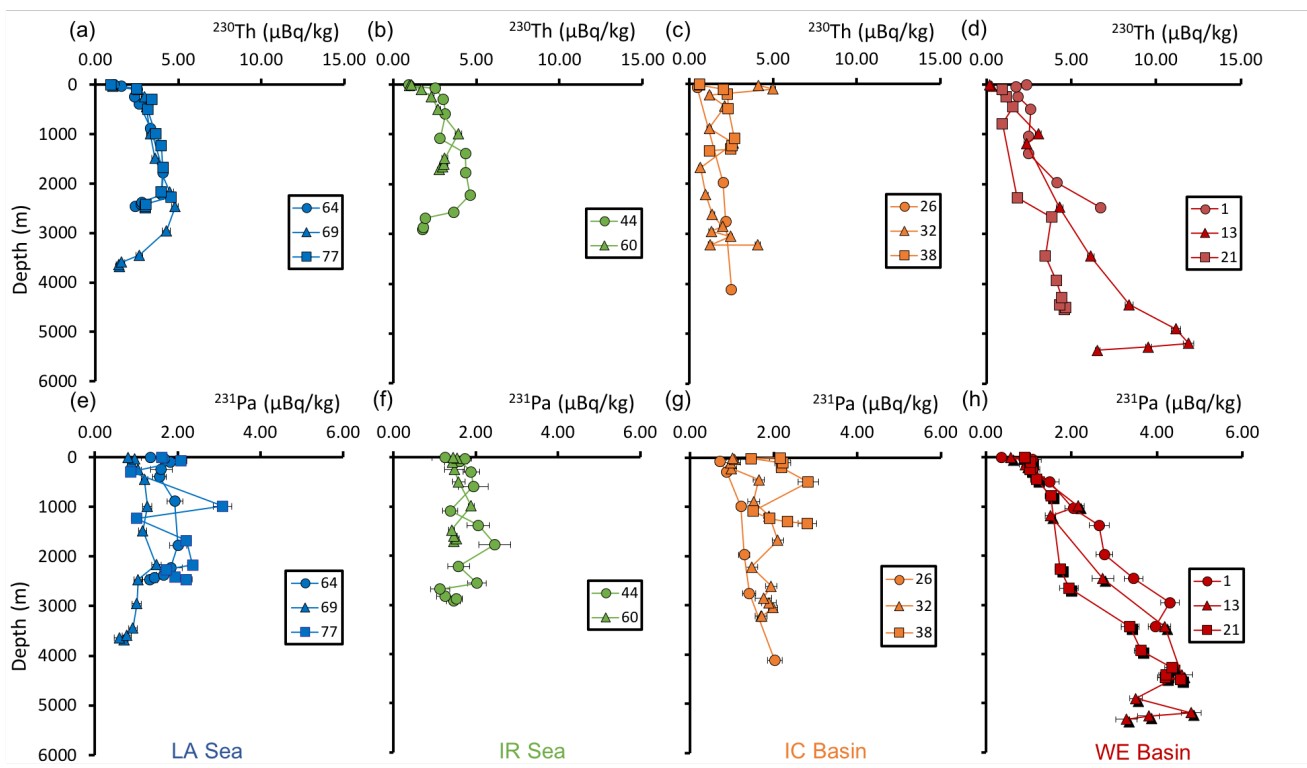

**Figure 2: Vertical profiles of $^{230}$Th (a-d) and $^{231}$Pa (e-h) in the water column along the GEOVIDE section. Colours corresponds to the region (as in Fig. 1). LA = Labrador, IR = Irminger, IC = Iceland, WE = West European). Uncertainties represent 2 standard error (2 s.e.).**

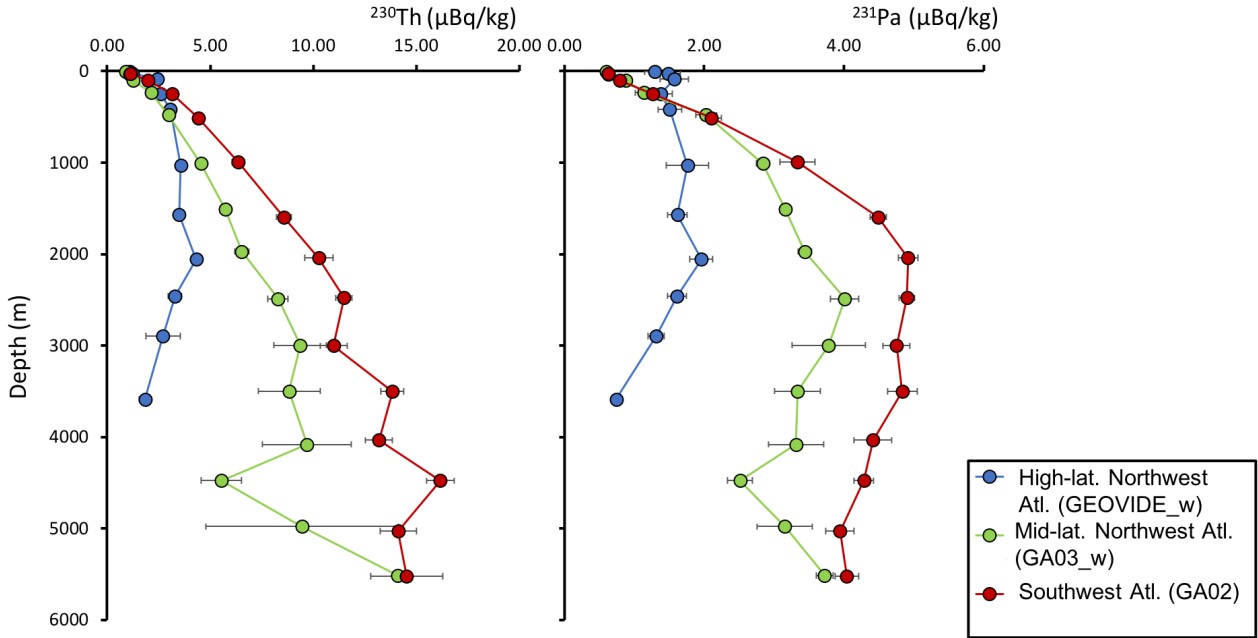

**Figure 3: Vertical profiles of <sup>230</sup>Th and <sup>231</sup>Pa from high-latitude Northwest Atlantic (west section of GEOVIDE), mid-latitude Northwest Atlantic (GA03_w), and Southwest Atlantic (GA02). Data from all stations were sorted by water depth and averages for depth and <sup>230</sup>Th and <sup>231</sup>Pa concentrations were taken for surface, 25 m,100 m, 250 m, 500 m, and every 500 m interval below. Error bars on <sup>230</sup>Th and <sup>231</sup>Pa concentrations averages reflect standard deviation of the mean of measurements.**

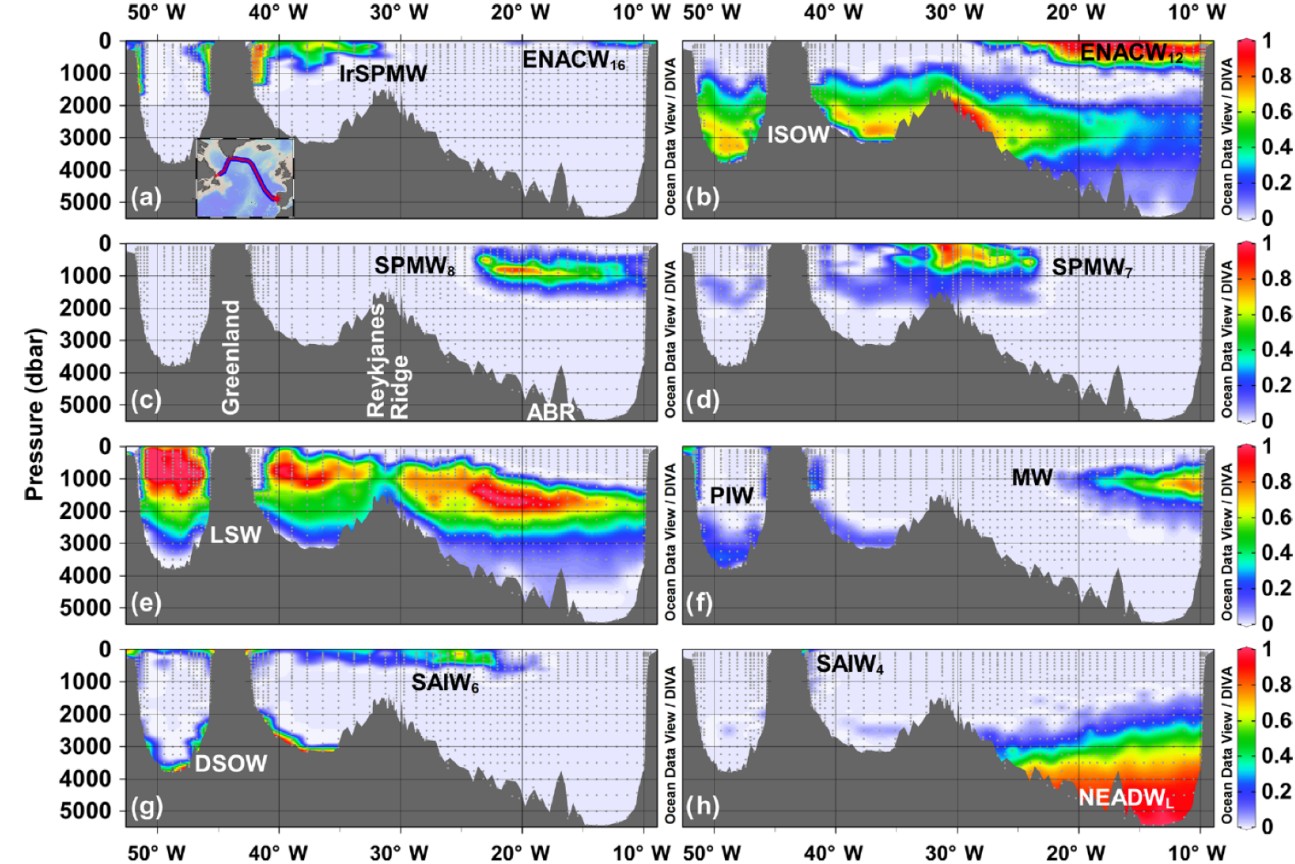

**Figure 4: Results of Extended Optimum MultiParameter (eOMP) analysis for the GEOVIDE section (García-Ibáñez et al., 2018). Colours reflect the fraction of water at each location assigned to the water mass shown in that panel: ENACW$_{16}$ and ENACW$_{12}$ = East North Atlantic Central Water of 16ºC and 12ºC; SPMW$_8$, SPMW$_7$, IrSPMW = Subpolar Mode Water of 8ºC, 7ºC and of the Irminger Sea; SAIW$_6$ and SAIW$_4$ = Subarctic Intermediate Water of 6ºC and 4ºC; MW = Mediterranean Water; PIW = Polar Intermediate Water; ISOW=Iceland–Scotland Overflow Water; LSW=Labrador Sea Water; DSOW: Denmark Strait Overflow Waters; and NEADW$_L$: Lower North East Atlantic Deep Water; ABR= Azores-Biscay Rise.**

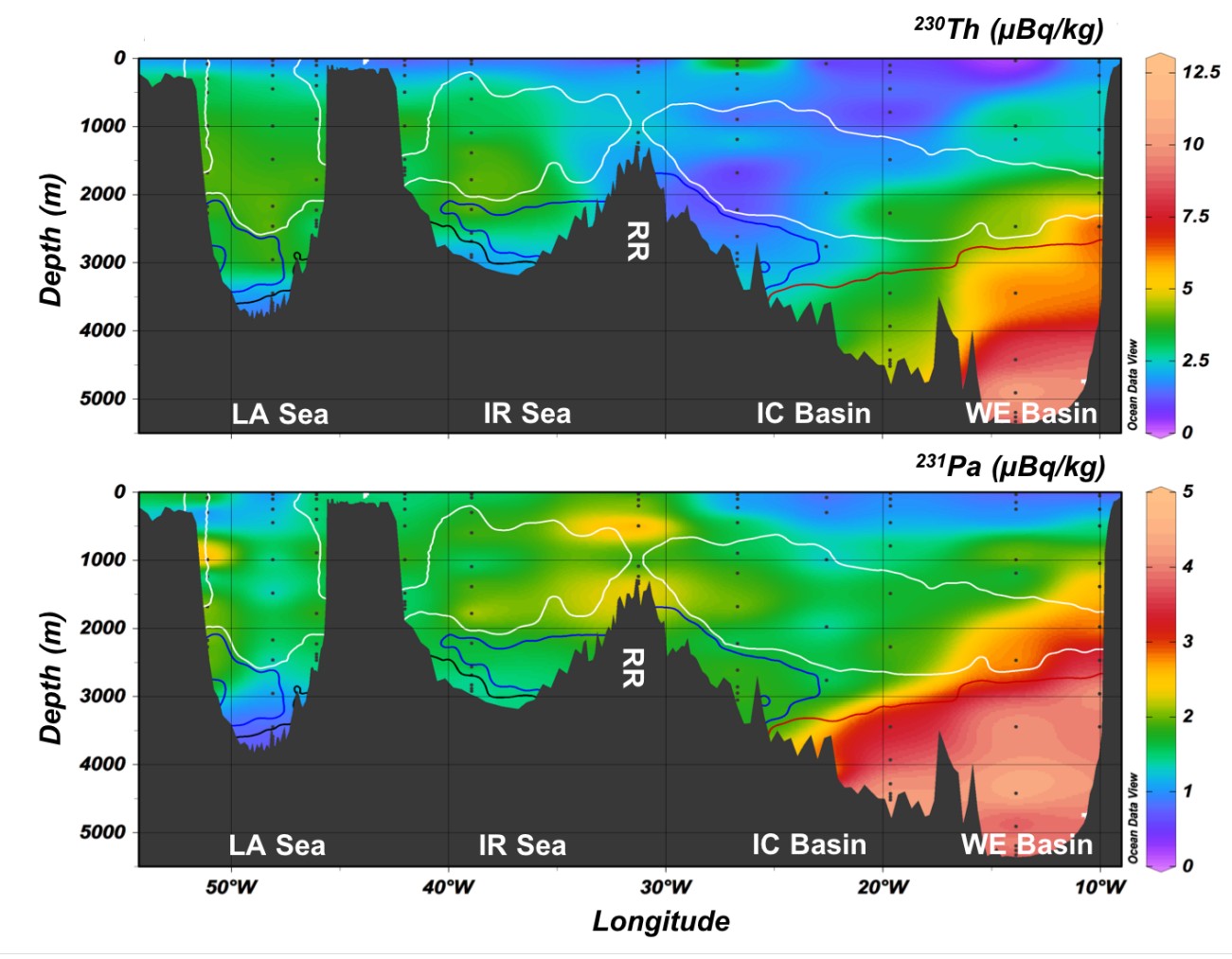

**Figure 5: Distribution of [230]Th and [231]Pa along the GEOVIDE section. Water masses were indicated by contours (black: DSOW; blue: ISOW; white: LSW; red: NEADW.) based on 50% level percentage composition of source water types from eOMP analysis.**

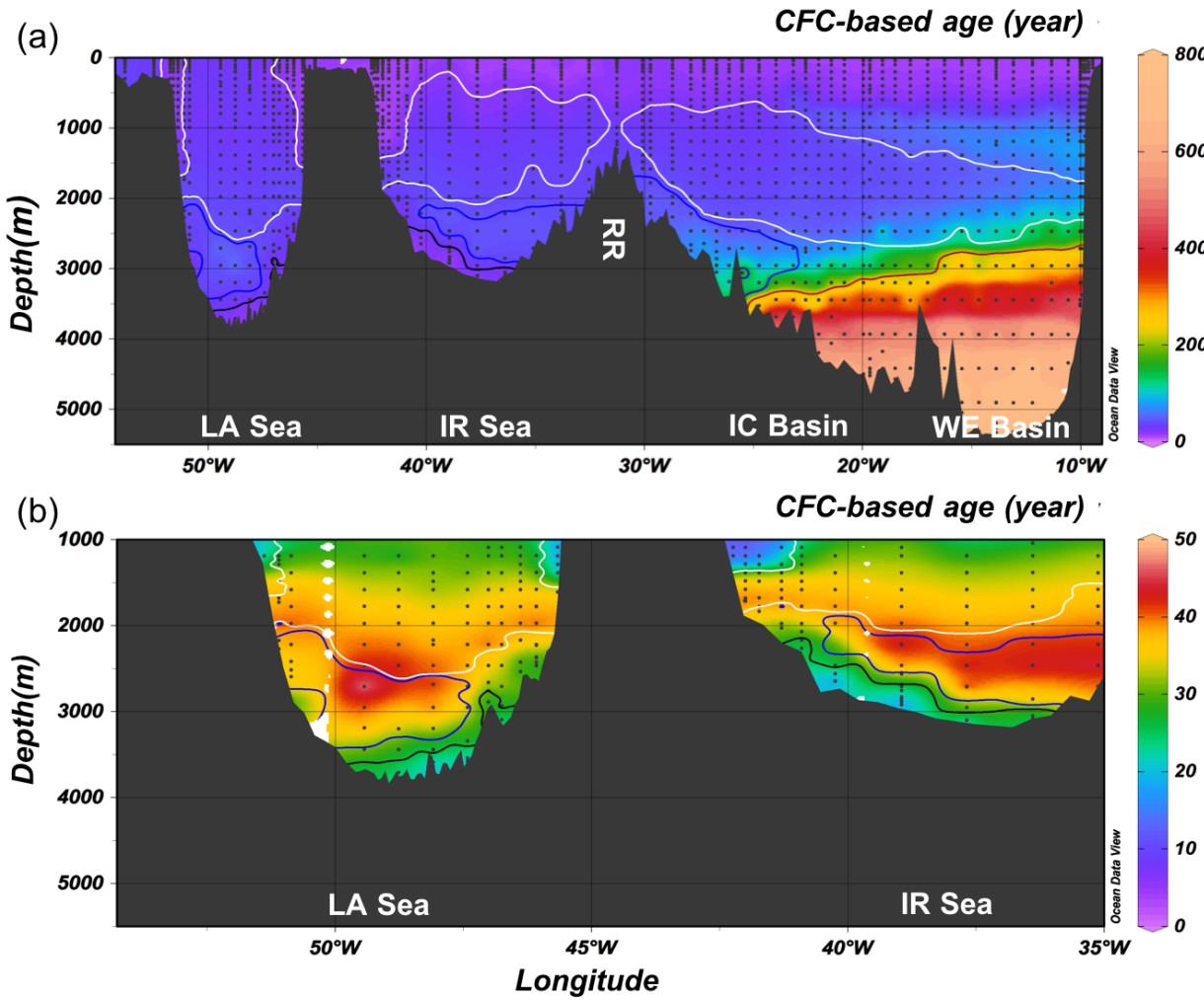

**Figure 6: Water mass age based on CFC data along the GEOVIDE section. (a) Full water-column data for the entire section, showing waters from 10 to 800 years in age; (b) A rescaled version of (a) omitting the upper 1000m and the older waters east of 35ºW to show age variation in recently ventilated deep-waters. Water masses were indicated by contours (black: DSOW; blue: ISOW; white: LSW; red: NEADW.) based on 50% level percentage composition of source water types from eOMP analysis.**

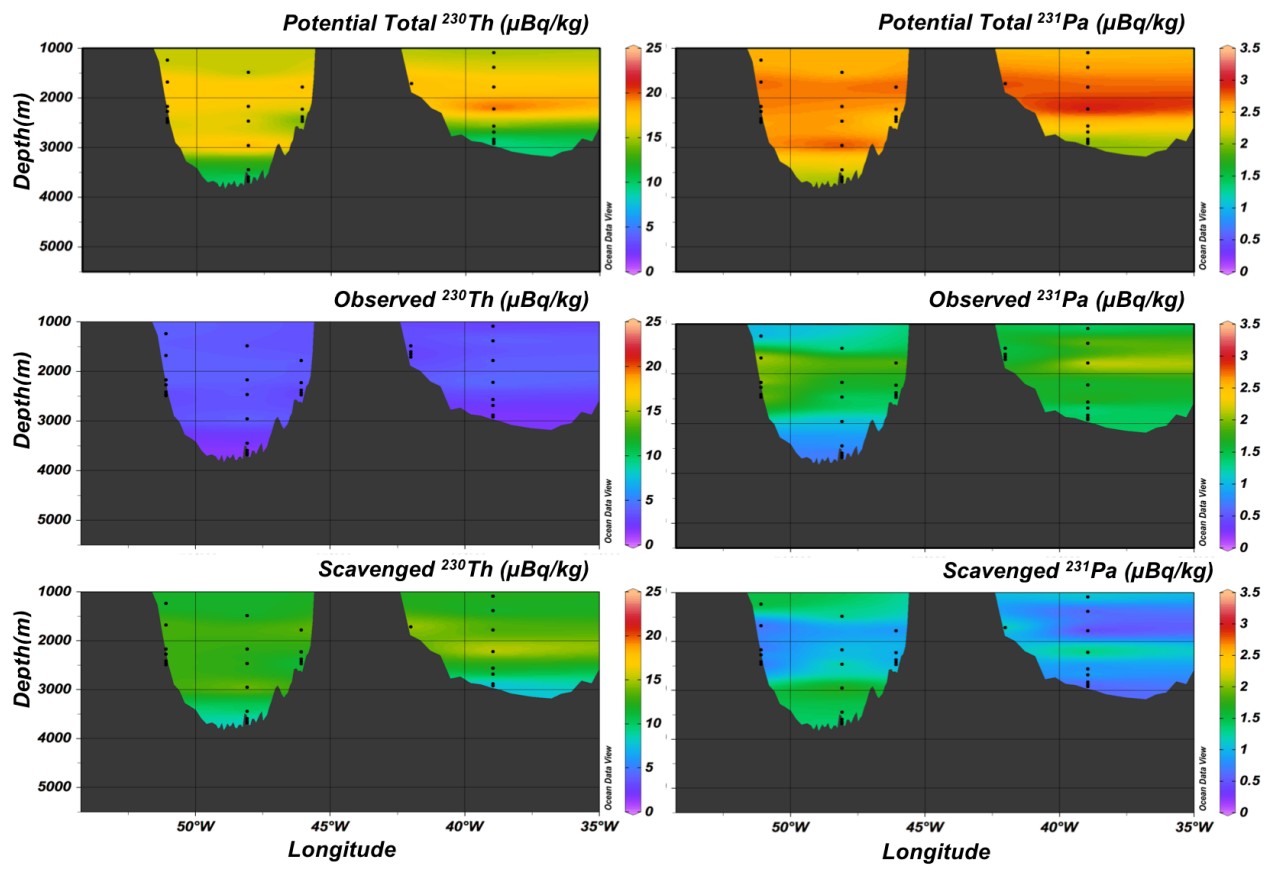

**Figure 7: Potential Total, Observed, and Scavenged components of [230]Th and [231]Pa in waters >1000 m water depth and west of 35ºW.**

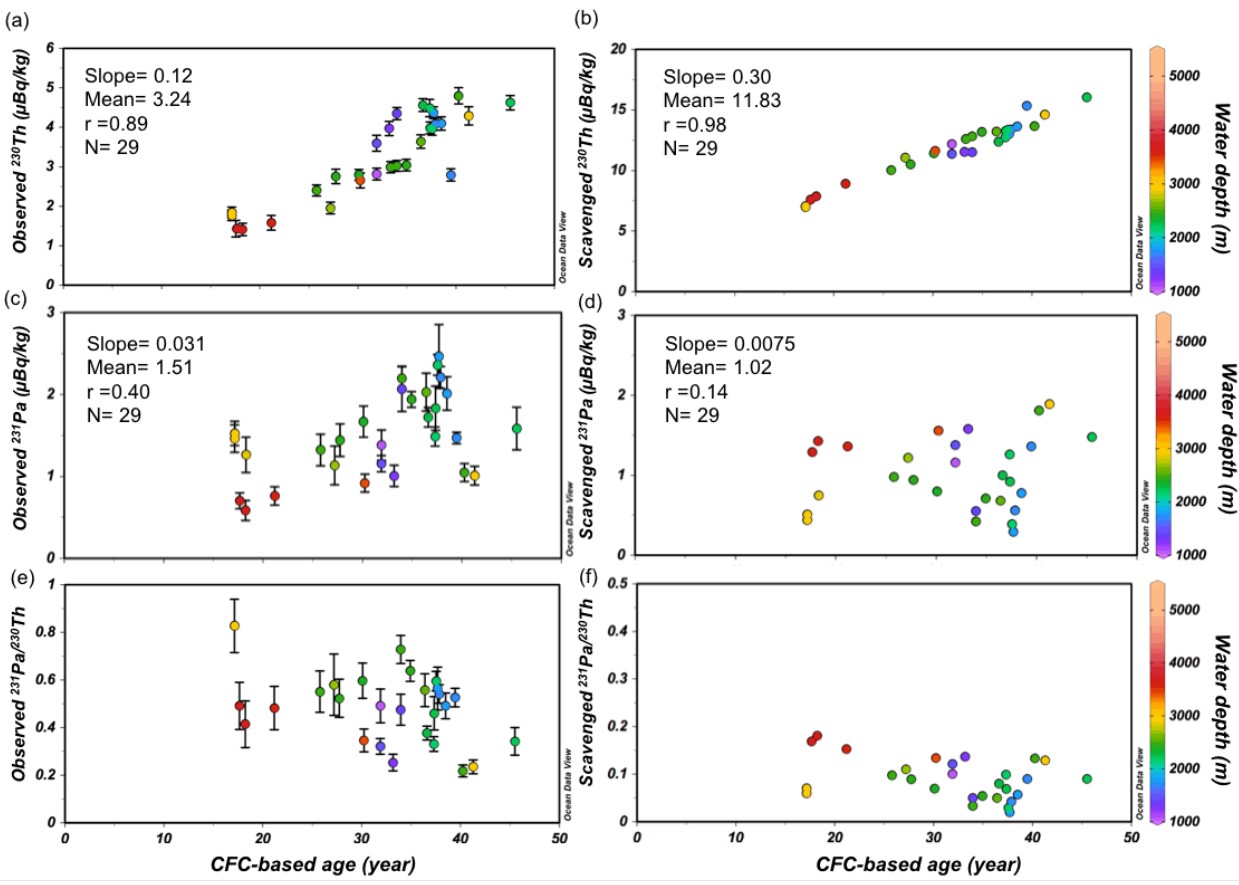

**Figure 8: Relationship between water mass age and the Observed and Scavenged components of $^{230}$Th, $^{231}$Pa and $^{231}$Pa/$^{230}$Th (colour coded by water depth). Least square fitting statistics were also given, i.e. slope and correlation coefficient r of the least square line, mean value and number of the data points. Note the increase of observed concentrations for both nuclides with age. Comparison of average values indicates that about three quarters of $^{230}$Th produced by decay is scavenged, compared with about half of the $^{231}$Pa.**

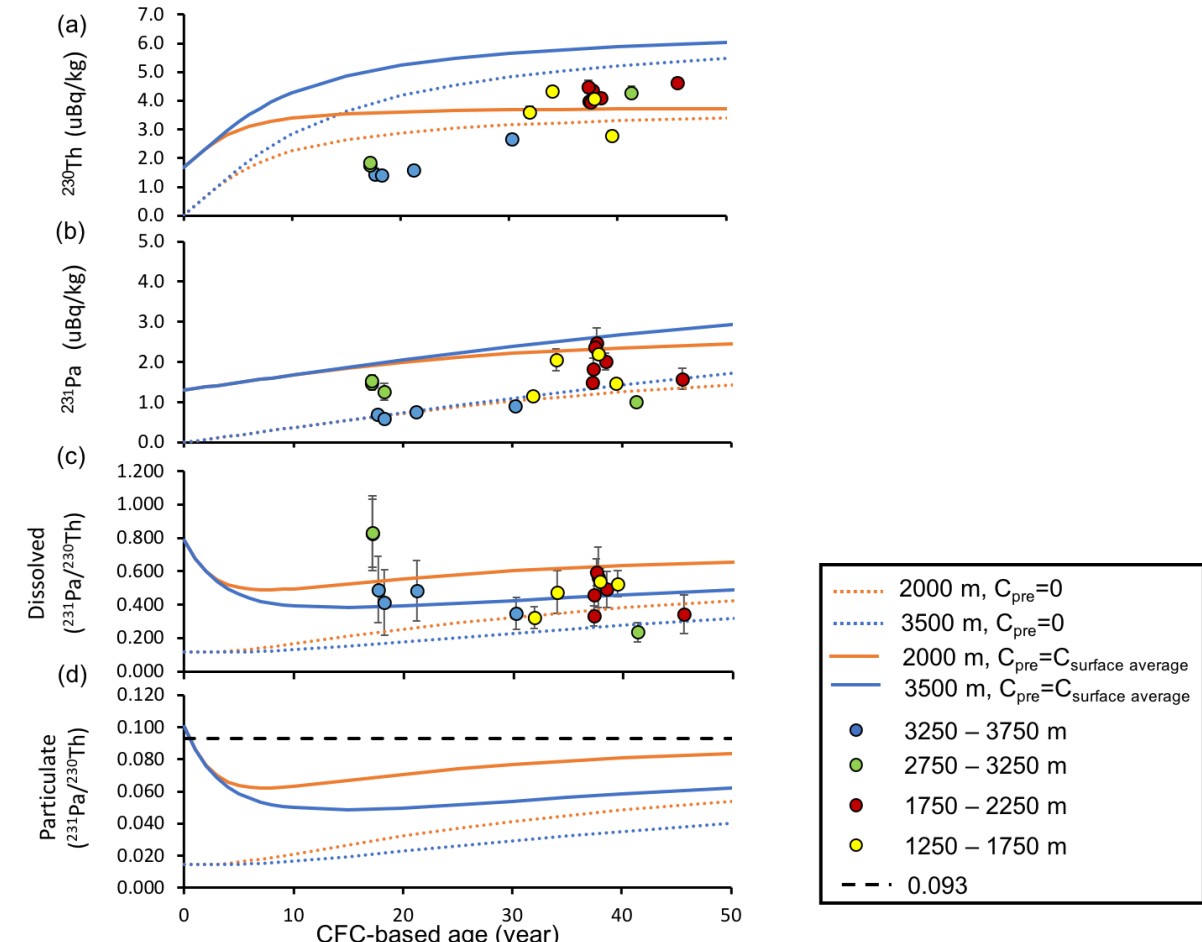

**Figure 9:** Results from a scavenging-mixing model of $^{230}$Th, $^{231}$Pa, Dissolved $^{231}$Pa/$^{230}$Th and particulate $^{231}$Pa/$^{230}$Th compared to observations. Preformed concentration ($C_{pre}$) were set at 0 (dashed line) and at the average surface concentration ($C_{surface\ average}$) from GEOVIDE section (solid line), i.e. $^{230}$Th= 1.66 µBq/kg, $^{231}$Pa= 1.31 µBq/kg. A version of this figure extending to older waters is available in the Supplemental Information Figure S2.

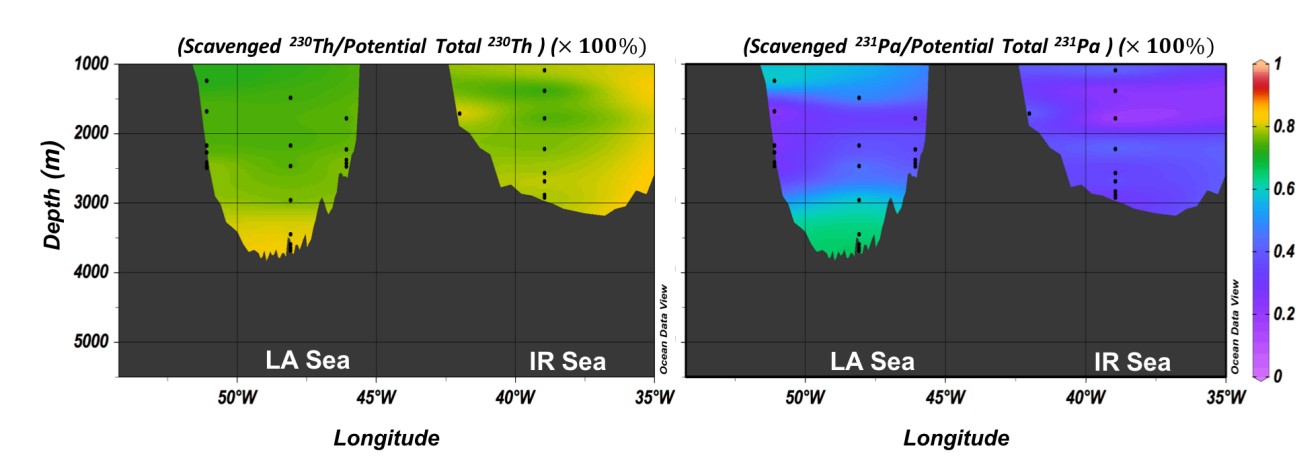

**Figure 10: Ratio of Scavenged component to Potential Total component for $^{230}$Th and $^{231}$Pa, providing an assessment of the relative importance of scavenging for the two nuclides, and of the location of scavenging.**

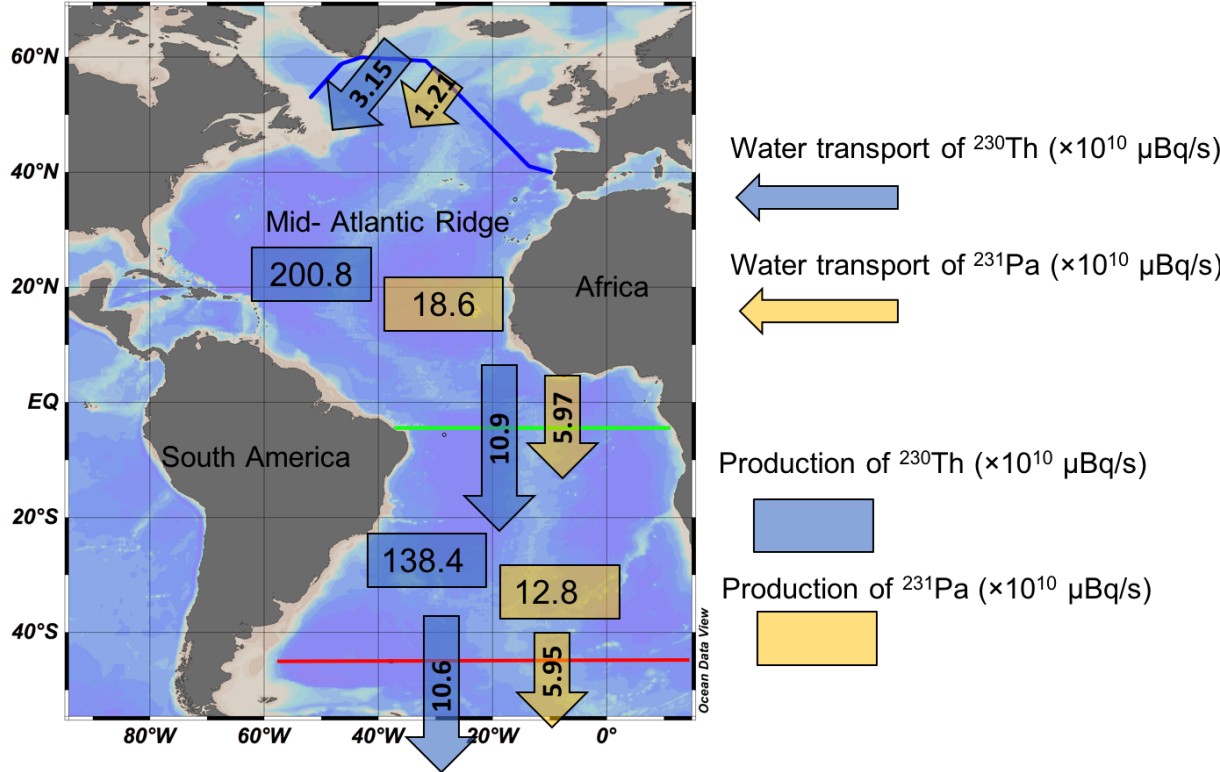

**Figure 11: Fluxes of $^{230}$Th (blue arrow) and $^{231}$Pa (yellow arrow) across the GEOVIDE section (blue solid line), 4.5°S (green solid line) and 45°S (red solid line). Also shown are production of $^{230}$Th (blue box) and $^{231}$Pa (yellow box) in the North Atlantic (between GEOVIDE section and 4.5°S) and in the South Atlantic (between 4.5°S and 45°S), based on calculation in Deng et al. (2014). These fluxes indicate that 4% of the $^{230}$Th produced in the North Atlantic is exported southward by ocean circulation, and 26% of the $^{231}$Pa.**