# Peer review of "Evolution of 231Pa and 230Th in overflow waters of the North Atlantic"

_Biogeosciences, 2018_

## Referee Comment (RC1) · R. Francois (Referee) · 13 May 2018

**R. Francois (Referee)**

rfrancois@eos.ubc.ca

Received and published: 13 May 2018

This paper is an important contribution to the growing database describing the distribution of Th-230 and Pa-231 in the world ocean. I recommend publication after considering the comments and questions listed below:

Abstract; line 19-20: The reason for a weaker correlation between Pa-231 and water mass age is likely due to the much lower concentrations and rate of ingrowth of Pa compared to Th. It may be worth adding error bars to Fig. 8(a),(c),(e) to highlight this point.

P 3; line 13: why was a 236U spike added to the sample?

P 4; Line 20 - 23; Fig. 3: If the purpose of this figure is to show that the concentrations

measured on the GEOVIDE line are much lower than measured further "downstream", I don't think that the data from 40°S are useful because they raise questions that are likely beyond the scope of this paper. The wording used in line 20 - 23 is incorrect since there is an apparent southward decrease instead of an increase between the North Atlantic and 40°S. Hayes et al.'s data show a strong longitudinal gradient for both Th-230 and Pa-231. Were the averages calculated over the entire longitudinal GA03 section? Were the averages calculated for  $40^{\circ}$ S comparable?

Fig. 5: ISOW is not indicated on this figure, even though it is one of the most prominent water mass. I would suggest replacing the salinity contour lines by the contour of the main water masses reported on Fig. 4, since the purpose of this figure is to relate these water masses to the distribution of the radionuclides

Line 31; p5 and Fig. 6: It is not entirely clear to me how the CFC ages were determined for the GEOVIDE section. Water masses and CFC ages were determined in 2012 of the eastern side of the section. From these data, the CFC ages of the different water masses could be determined for 2012 and these ages were used to determine the CFC ages of all the samples collected in 2014 from their water mass constituents.

My first problem is how you could estimate the CFC age of the water masses west of Greenland. Surely, the age of ISOW and DSOW in this section of the transect must be older than in the eastern section since these water masses are farther removed from their site of formation.

My second problem is how can you distinguish ages between 100 and 800 years with CFCs considering that CFC manufacture only started 80 years ago?

Finally, and maybe more fundamentally, I question whether you can apply CFC-age to Th-230 and Pa-231. I am not an expert on this question but here are my concerns:

Each (most?) of your samples consists of a mixture of different water masses. Even the red zones on Fig. 4, indicate up to 20% mixing. If you mix 2 water masses of different
ages, the CFC age of the mixed sample will be biased toward the younger age because there is more CFCs in the younger end member (although CFC concentrations have quasi plateaued since 1990). On the other hand, for the same mixed sample, the Th-230 age will be biased towards the older end-member because it contains more Th-230. So, in a mixed sample, Th-230 had more time for in-growth than indicated by the CFC age. Because Pa-231 grows in slower than Th-230, this effect is less pronounced. Therefore, I believe that the ingrown Th and Pa calculated with CFC ages using equations 4.1 and 4.2 must be taken as minima and more so for Th than for Pa.

Fig. 6: Adding water mass contour lines would be helpful

Line 16; p6 "Preformed component" "..in the absence of measurement at the exact location of deep water formation...".

While this statement is correct for the formation of DSOW and ISOW, I would say that station 69 is essentially at the location of LSW formation

"...we set the preformed value as the average of concentrations measured in waters

Alternatively the reasoning followed by the authors may be that cooling happens at the surface, and therefore that must be the starting point, and the higher Th concentrations found in the water below at the sites of deep water formation reflects the residence time of this surface water in the convection cell. If this is the case, that should be more explicit in the paper. I would also use the available data for surface water in the Labrador and Nordic Seas only, which are available from this study and Moran's.

Line 30; p6: as per discussion above, calculated potential total concentration should be viewed as minima (because CFC ages underestimate the Th and Pa "ages")

Line 4; p7: "..is the net of nuclide added from above by [add: DESORPTION FROM] settling particles..."

Line 8-9; p7: Considering that the CFC ages are younger than the Th ages, potential and scavenged Th-230 must be viewed as minima. Also, the underestimation of fraction scavenged is larger for Th than Pa.

Line 14; p7: I guess the authors deduce that about 75% of the Th-230 produced is scavenged because the slope of slope of observed Th-230 vs time is 25% of the rate of production, right? If so, it would be helpful to be explicit and to report the value of the slope in Fig. 8a. Wouldn't the intercept also provide an independent estimate of preformed? I am not sure what the intercept on scavenged Th-230 figure means. I would also reiterate here that 75% scavenging is a minimum, and it could be that scavenging is more underestimated in older water than younger ones, producing this apparent intercept. It is not clear to me why the same can't be attempted for Pa, at least for the observed values.

Line 15-14; p7: "This ratio is consistent with the average 230Th for these waters, which requires that about three times more 230Th has been removed by scavenging"

Line 18; p7:" ... the average [Pa] values indicate that about half of the 231Pa remains in the water"
I can't follow the line of reasoning here. I think the authors need to be more explicit.

P2; line 23: ".. there is no simple relationship between increasing Pa/Th and age, as would be expected"

P7; Line 21; p7:"..The hypothesis that Pa/Th ratios increase as water mass ages.."

We should not expect seawater Pa/Th to simply increase with water mass age. If dissolved Th and Pa activities are initially low, Pa/Th in the water column (and underlying sediments) should initially decrease with age of water mass because Th grows in faster than Pa. It's only once Th has regained levels closer to equilibrium with scavenging (and therefore grows more slowly) that Pa starts growing faster, resulting in increasing Pa/Th. This effect is clearly illustrated in the paper by Luo et al. (2010) (see Fig. 14 in the paper). Fig. 8e,f may show a hint of this initial decreasing trend.

(Luo et al also argued "Clearly, it is impossible to constrain the history of changes in the AMOC from the evolution of 231Pa/230Th at one site, as was attempted by McManus et al.; p 395; last sentence of first paragraph)

Section 4.3: I am puzzled by Fig 9. The intercepts don't seem to match with the preformed values. The fit with the data is terrible. I am not sure what this section is telling us.

Line 28; p7: wouldn't it be better to follow density rather than depth?

Line 4; p8: why did you choose station 1 and 13? Is it because they are the least ventilated and therefore these parameters could be estimated from a linear fit? If this is the case, that should be indicated. However, is it reasonable to apply scavenging parameters from a margin to open ocean stations?

Fig. 9: How can Pa/Th (zero preformed) increase with age (c) if Th (a) grows faster than Pa (b)?

Section 4.4:

BGD
This discussion should take into account that scavenged Th and Pa are underestimated and proportionally more for Th than Pa

Since potential and scavenged are underestimated by exactly the same absolute amount, calculated scavenged/potential ratios should also be taken as minima.

P 9; line 20-22; I think it would be helpful for the reader to provide the details of this mass balance calculation. For instance, what volume for the North Atlantic did you use to calculate production? It seems that the results from this mass balance corroborate that scavenged (and therefore potential) Th and Pa calculated from equation 4.3 are underestimated because of the use of CFC ages.

I am a little surprised that so much Pa is scavenged in the North Atlantic. If 77% of the production is scavenged, then the average Pa/Th of N. Atlantic sediments should be ca.0.073, which is higher than measured in the central N. Atlantic basin. That means that boundary scavenging must be more significant than I thought.

Roger Francois

---

## Referee Comment (RC2) · Anonymous Referee #2 · 14 May 2018

Here Deng et al. provide a very nice and concise piece of science. They examine crucial assumptions made for the application of 231Pa/230Th as an AMOC proxy by providing an extensive new data set of water 231Pa and 230Th concentrations. This is hard won data and the authors deserve credit for their efforts. The new data extends the former GEOTRACES transects by Hayes15 and Deng14 towards the northern North Atlantic representing a definite reality-check for the assumptions made when using 231Pa/230Th as a proxy. These assumptions have been made based on the elegant approach of measuring a kinetic tracer with a constant and well-defined input-function not involved in the carbon-cycle (Yu1996). While previous studies already proofed the consistence and capability of 231Pa/230Th as an AMOC proxy the novelty of this study is the systematic examination of the behaviour of 231Pa and 230Th in the

northern North Atlantic in the water masses recently influenced by NADW formation with a new set of samples and by state-of-the art analytical methods. Therefore, this manuscript certainly deserves publication in Biogeosciences.

Given the published results from the 231Pa/230Th proxy of the last decades I would have wondered if this study would have come to a different conclusion. But here Deng et al. make very good cases by confirming the prerequisites for using 231Pa/230Th as AMOC proxy. The only little weakness of the manuscript is the missed opportunity of setting the new findings into the context of the attempts of using 231Pa/230Th as a large scale AMOC proxy. There are several of papers dealing with a single 231Pa/230Th profile from on location, but the results of the few which deal with comprehensive compilation approaches could be better assessed and discussed here. Besides the already mentioned Yu et al. 1996, I think in particular of Bradtmiller et al. 2014, which use the large scale 231Pa deficit for analyzing the HS1 AMOC. It would be worth of shortly recapitulate their results in the light of the new results presented here. Besides the connection to observational paleo data I also miss the comparison to theoretical or model studies. The authors should in particular present a short comparison to the predictions made by Marchal et al. 2000 and the most recent attempt by Rempfer et al. 2017. Further, most of the features reported here have been anticipated by the simple box-model approach by Luo et al. 2010. They already found a very weak correlation of 231Pa/230Th with water mass age, highlighting a vertical gradient not a horizontal in the presence of an active AMOC (see specific points below).

page 3, line 2: recurring typo of "R/V Pourquoi Pas?"

page 3, line 15: please specify how many several months are

page 3, line 19: what was the analytical yield ($\sim$ range)of the anion chromatography for Pa and Th?

page 3, line 25: what was the 232Th/231Pa in the Pa-samples? Was the correction for the 232ThH interference necessary, if yes, how big was the contribution to the Pasignal?

page 5, line 14: ISOW is mentioned but not shown in Fig. 5.

supplement: please add a column specifying the errors to the given concentrations

supplement page 7 line 7: I'm aware that this is of marginal importance given the final result, but maybe the authors could elaborate on the value 0.7 in (3) and (4). 0.7 seems a little bit high for an average, or at least unnecessarily high at the high end of the possible range according to Henderson and Anderson 2003 or Bourne et al. 2012. Further, is the detrital correction required for particles slipping through the 0.45 $\mu$m filters? Is the added HCl capable of leaching of the 232Th (and 230Th and 231Pa) from these particles?

Fig. 3,8,9: error bars are misssing

page 7, line 21: It is not surprising that 231Pa/230Th does not correlate with water mass age very much. This has been already predicted by Luo et al. 2010. Much more important is the vertical decrease within one circulating mater mass (e.g. Burckel et al. 2016). Thus, the sentence that "231Pa/230 ratios increase as water mass ages forms the foundation of using 231Pa/230Th in discrete cores" is not completely accurate.

page 10, line 13: typo. two times "demostrates".

Fig. 6 : I assume the x-axis has changed between (a) and (b), but they are both shown on the same longitudinal scale.

Fig. 3: maybe it would be worth of showing 231Pa/230Th as well in an additional panel (c). Fig. 8: please indicate water depth at the colour bar.

References: Bourne, M., et al., 2012. Improved determination of marine sedimentation rates using 230Thxs. Geochemistry Geophysics Geosystems 13. Bradtmiller, L., et al., 2014. 231Pa/230Th evidence for a weakened but persistent Atlantic meridional over-turning circulation during Heinrich Stadial 1. Nature Communications 5. Burckel, P., et

al., 2016. Changes in the geometry and strength of the Atlantic Meridional Overturning Circulation during the last glacial (20-50 ka). Climate of the Past 12. Deng, F., et al., 2014. Controls on seawater 231Pa, 230Th and 232Th concentrations along the flow paths of deep waters in the Southwest Atlantic. Earth and Planetary Science Letters 390. Hayes, C., et al., 2015. 230Th and 231Pa on GEOTRACES GA03, the U.S. GEO-TRACES North Atlantic transect, and implications for modern and paleoceanographic chemical fluxes. Deep Sea Research Part II: Topical Studies in Oceanography 116. Henderson, G., et al., 2003. The U-series toolbox for paleoceanography, Uranium Series Geochemistry. Reviews in Mineralogy and Geochemistry 128. Luo, Y., et al., 2010. Sediment 231Pa/230Th as a recorder of the rate of the Atlantic meridional overturning circulation: insights from a 2-D model. Ocean Science 6. Marchal, O., et al., 2000. Ocean thermohaline circulation and sedimentary 231Pa/230Th ratio. Paleoceanography 15. Rempfer, J., et al., 2017. New insights into cycling of 231Pa and 230Th in the Atlantic Ocean. Earth and Planetary Science Letters 468. Yu, E., et al., 1996. Similar rates of modern and last-glacial ocean thermohaline circulation inferred from radiochemical data. Nature 379.

---

## Referee Comment (RC3) · Anonymous Referee #3 · 19 May 2018

Deng and coworkers have produced an important data set by analyzing samples collected on GEOTRACES Section GA01 (GEOVIDE) for 231Pa and 230Th. These results hold valuable implications for the use of these radionuclides as tracers of North Atlantic deep water ventilation, and its variability through time (via the analysis of 231Pa/230Th ratios, henceforth "Pa/Th", archived in marine sediments). However, there are some major issues that should be addressed before I can recommend that the manuscript be published, as detailed in the following.

Major Comments:

1) Clarify and emphasize the principal take home message. The concluding sentence of the manuscript states "and continues to support the use of sedimentary 231Pa/230Th measurements at a basin scale to constrain overturning circulation." This

statement is based on the calculated southward export of dissolved Pa being substantially greater than the southward transport of Th. However, as clearly stated in the manuscript, there is no observable relationship between dissolved Pa/Th ratio and water mass age. This observation is in direct contradiction to the principles underlying the use of sedimentary Pa/Th ratios to reconstruct past variability of the ventilation of deep water in the North Atlantic Ocean, where it is assumed that dissolved Pa/Th ratios will increase monotonically with age after water mass formation due to the longer residence time of Pa compared to Th. How can the authors conclude that their results support the use of sedimentary Pa/Th ratios to constrain overturning circulation when there is no relationship between dissolved Pa/Th ratio and water mass age?

This issue becomes even more important if one considers the evolution over time of dissolved Pa/Th ratios down the length of the western Atlantic Ocean. Although the authors do not present Pa/Th ratios for the mid-latitude North Atlantic or at 40°S along with the dissolved 230Th and 213Pa data in Figure 3, eyeballing the dissolved 230Th and 231Pa profiles for these regions suggests very little change in the dissolved Pa/Th ratio from north to south, from GEOVIDE near the formation region to GA10 at 40°S. If a more rigorous analysis of the data reveals this to be true, i.e., that there is no change with water mass age in the dissolved Pa/Th ratio down the entire length of the Atlantic Ocean, then I do not see how Pa/Th ratios can be related to ventilation rate, either in the modern ocean or to reconstruct climate-related changes in ventilation rate in the past.

The manuscript would have much greater impact if this point were discussed at length, incorporating data from the entire Atlantic Ocean.

2) The development and application of CFC ages are unclear. The description in the Supplementary material (page 8) is helpful, but some of the output is not meaningful. CFC ages are not valid for time periods older than the initial introduction of CFCs into the environment in the middle of the 20th century. My colleagues who are experts in the use of CFC ages generally decline to interpret apparent ages greater than about 40

to 50 years due to the uncertainties inherent in interpreting CFC ages in water masses last exposed to the atmosphere during the earliest days when CFCs were tagging water masses. Therefore, I do not understand how Mediterranean Water can be assigned an age of 91±8 years, or NEADW can be assigned an age of 989±48 years (Table S2). Unless there is something not explained in the paper that allows CFC ages this old to be computed, the old ages should be removed from the paper. Accordingly, Figure 6a can be removed, leaving only Figure 6b in the paper.

Related to Figure 6, it is very confusing that the two panels have different longitude scales, but the scale for Figure 6a is not shown. If there is a reason to retain Figure 6a, then include the longitude scale and note that Figure 6b incorporates only the western half (approximately) of Figure 6a.

3) The calculation and interpretation of scavenging (rates and percentages) relies strongly on the estimated CFC ages (Figures 7 through 9). Given this important sensitivity to estimated age, I recommend that the authors include a discussion of the uncertainty in the CFC ages, and how that may affect their interpretation.

4) As discussed on page 6 of the manuscript, the initial (preformed) concentrations of dissolved 230Th and 231Pa at the time the water masses formed are unknown, so the authors assume that the preformed concentrations are either equal to average concentrations in surface waters sampled along the GEOVIDE transect or that preformed concentrations are zero. Unfortunately, until data are available for the Nordic Seas and the Labrador Sea during times of winter convection, these may be the only options available for the type of analysis described here. Nevertheless, it would be helpful if the authors provided additional discussion of the sensitivity of their derived products (e.g., fraction scavenged for each isotope) to the values assumed for the preformed concentration.

In this context, it would also be helpful to discuss the possibility that each water mass has a different preformed concentration, and how this might affect the interpretation of

the data presented in Figure 8. Implicit in the presentation of the data presented in Figure 8 is the assumption that all water masses have the same preformed concentrations. What if this is not the case? How would that alter the interpretation of the data?

5) There seems to be a problem with the model curves shown in Figure 9, where the solid lines depict model results for the case where average surface water concentrations determined for samples collected on the GEOVIDE cruise were used in place of the preformed concentrations. If that were the case, then why do the projected model concentrations at zero age (solid lines) intersect the Y axis at concentrations about double the values reported for average surface concentrations on page 6 (0.108 dpm/1000 L for 230Th and 0.089 dpm/1000 L for 231Pa)? If I understand the model correctly, then the concentrations at zero age should equal the assigned preformed concentrations. Is this not the case? Is the problem that the preformed concentrations are introduced twice in equation 6 (supplementary material)? Note that C(pre) and C(surface) are one and the same. Should both terms be in equation 6?

6) Summary of major comments: Given all of the uncertainties in CFC age and in initial (preformed) concentrations of Pa and Th, it seems that a stronger paper than the one under review would be produced by integrating the new data from GEOVIDE with other data from GEOTRACES cruises down the length of the Atlantic Ocean (GA02, GA03, GA10 and, perhaps, other sections with data in the GEOTRACES IDP2017, if there are any) to establish firmly whether or not the dissolved Pa/Th ratio in deep Atlantic water evolves over time as assumed in the application of sedimentary Pa/Th ratios to constrain past changes in the rate of ventilation of North Atlantic Deep Water.

Minor comments:

1) The authors report their results using historical units (dpm/1000 L) but their results will be converted to SI units when included in the next GEOTRACES data product. All of the Th and Pa data currently in the IDP2017 are presented using SI units, so

why not make this conversion before publishing the GEOVIDE data? 2) page 6 line 1 "Steinfeldt" is misspelled. 3) page 6, definition of "Ingrown component": Are U concentrations normalized to a constant salinity? To the salinity measured for each sample? Something else? 4) page 7 line 16: "three times more 230Th has been removed by scavenging" than "what?" Complete the description of the comparison being made. 5) page 8 line 7: Change "run" to "ran" 6) page 8 line 27: delete the "in" prior to "from DSOW" 7) page 8 line 28, and elsewhere: Bottom scavenging of 230Th was first noted by Bacon and Anderson (1982) and by Anderson et al. (1983; EPSL 66(1-3), 73-90, not the paper cited by Deng et al.) in their study of the eastern tropical Pacific. These early indications of bottom scavenging should be cited. 8) page 9 line 12: What is the source of the average 230Th and 231Pa concentrations in the upper limb? The values given here are not those given on page 6 for GEOVIDE surface waters, so the source should be given. 9) page 10 line 7: Change "that" to "than"

―――――――――――――――――――――

---

## Author Response (AR1)

**"Evolution of 231Pa and 230Th in overflow waters of the North Atlantic" by Feifei Deng et al.**

**Response to referees**

We would like to thank all referees for their time reading the manuscript and giving constructive suggestions to improve the paper. We are pleased that all three referees appreciate the dataset and broadly welcome publication of this paper.

Four issues were raised by more than one reviewer. Before we respond to the points raised in individual reviews, we address these four issues.

1. Disagreement between the various referees about the key nature of findings led to a change in the introduction section, and the conclusion section accordingly.

It is interesting that referee comments varied from those that said our work confirmed the use of 231Pa/230Th as a paleoproxy, to those that thought we have proved the proxy does not work. It is clearly important to more clearly state how the proxy might be interpreted, and whether such interpretation is justified following our work. So we have clarified in the introduction section that there are two conceptual models that form the foundation of the interpretation of sedimentary 231Pa/230Th ratios in terms of past rates of deep water circulation.

Model 1 relies on a net export of 231Pa out of the Atlantic due to the residence time 231Pa being longer than 230Th (an approach adopted by studies such as McManus et al., 2014, and Bradtmiller et al., 2014).

Model 2 is based on the systematic evolution of 231Pa/230Th with water mass age, which has seen its application in Negre et al. (2010).

Our study provides an opportunity to assess the validity of these models. In the conclusion section, we made clear that our result supports the Model 1 interpretation that there is a northward export of 231Pa out of the Atlantic, but raises questions about model 2 because there is no simple relationship between 231Pa/230Th and water mass age.

2. Reviewers questioned the reliability of CFC ages, especially for older waters, and asked for more details about how these ages were calculated.

We have clarified that CFC-based ages were calculated with Transit Time Distribution (TTD) method, and were different from the CFC concentration/tracer ages based on the atmospheric history of CFC.

Briefly, we computed CFC-based ages combining CFC concentrations and water mass composition obtained from extended Optimum Multi-Parameter (eOMP) analysis. First, TTD mean ages for each source water type (SWT) were calculated from CFC concentrations and eOMP analysis from OVIDE cruise 2012. These mean ages for each SWT were then combined with water mass composition obtained from

eOMP analysis for GEOVIDE 2014 to give an age for water at depths where water mass compositions are available. This approach assumes that the mixing of the ages (not the CFC concentrations) is linear, and decides that the aging of water is due to variations in water mass composition, rather than the increase of spreading time of the water. In further considering and discussing this calculation we have relied heavily on input from Reiner Steinfeldt and we have therefore added him as an author to the manuscript.

3. Reviewers questioned why 231Pa and 230Th concentrations given by the model did not reflect quoted preformed values at zero water age.

We have considered the modelling work carefully. On reflection, we consider our introduction of a surface term to the model to be incorrect and have consequently removed it during revision, to rely on the model exactly as originally presented in Moran et al. (1997).

**4. Reviewers suggested the use of SI units adopted in GEOTRACES data product.**

We have changed the units in the data table and throughout the text using  $\mu$ Bq/kg for 231Pa and 230Th, and pmol/kg for 232Th.

Below, we respond to the referees point by point. Reviewers' comments are in blue, and our responses are in black.

**Reviewer 1: R. Francois (Referee)**

This paper is an important contribution to the growing database describing the distribution of Th-230 and Pa-231 in the world ocean. I recommend publication after considering the comments and questions listed below:

Abstract; line 19-20: The reason for a weaker correlation between Pa-231 and water mass age is likely due to the much lower concentrations and rate of ingrowth of Pa compared to Th. It may be worth adding error bars to Fig. 8(a),(c),(e) to highlight this point.

**Author's response:**

We agree with the reviewer that the lower concentration and slower ingrowth rate of Pa might be the reason for a weaker correlation. We have added error bars to Fig. 8(a),(c),(e), and pointed out this likely reason in the text as suggested.

P 3; line 13: why was a 236U spike added to the sample?

**Author's response:**

We add the spike because we sometimes recover U during chemical separation of Th and Pa. We have made that clear in the revised manuscript.

P4; Line 20 - 23; Fig. 3: If the purpose of this figure is to show that the concentrations measured on the GEOVIDE line are much lower than measured further "downstream", I don't think that the data from 40S are useful because they raise questions that are likely beyond the scope of this paper. The wording used in line 20 - 23 is incorrect since there is an apparent southward decrease instead of an increase between the North Atlantic and 40S. Hayes et al.'s data show a strong longitudinal gradient for bothTh-230 and Pa-231. Were the averages calculated over the entire longitudinal GA03 section? Were the averages calculated for 40S comparable?

**Author's response:**

We agree that 40°S data raise points outside the scope of this manuscript. We have replaced Fig 3 with a figure that compares only published data from west of the Mid-Atlantic Ridge from GA01 (i.e. this study), GA03 (Hayes) and GA02 (South Atlantic). This approach more closely follows deep-waters as they age along their flow path, and shows an increase of concentration of both nuclides.

Fig. 5: ISOW is not indicated on this figure, even though it is one of the most prominent water mass. I would suggest replacing the salinity contour lines by the contour of the main water masses reported on Fig. 4, since the purpose of this figure is to relate these water masses to the distribution of the radionuclides.

**Author's response:**

We agree with the reviewer that plotting contours of the main water masses instead of salinity contours would be more useful and have modified figures accordingly.

Line 31; p5 and Fig. 6: It is not entirely clear to me how the CFC ages were determined for the GEOVIDE section. Water masses and CFC ages were determined in 2012 of the eastern side of the section. From these data, the CFC ages of the different water masses could be determined for 2012 and these ages were used to determine the CFC ages of all the samples collected in 2014 from their water mass constituents.

My first problem is how you could estimate the CFC age of the water masses west of Greenland. Surely, the age of ISOW and DSOW in this section of the transect must be older than in the eastern section since these water masses are farther removed from their site of formation.

My second problem is how can you distinguish ages between 100 and 800 years with CFCs considering that CFC manufacture only started 80 years ago?

Finally, and maybe more fundamentally, I question whether you can apply CFC-age to Th-230 and Pa-231. I am not an expert on this question but here are my concerns: Each (most?) of your samples consists of a mixture of different water masses. Even the red zones on Fig. 4, indicate up to 20% mixing. If you mix 2 water masses of different ages, the CFC age of the mixed sample will be biased toward the younger age because there is more CFCs in the younger end member (although CFC concentrations have quasi plateaued since 1990). On the other hand, for the same mixed sample, the Th-230 age will be biased towards the older end-member because it contains more Th-230. So, in a mixed sample, Th-230 had more time for in-growth than indicated by the CFC age. Because Pa-231 grows in slower than Th-230, this effect is less pronounced. Therefore, I believe that the ingrown Th and Pa calculated

**with CFC ages using equations 4.1 and 4.2 must be taken as minima and more so for Th than for Pa.**

**Author's response:**

Thank you for raising the question. We have included a clear description of how CFC ages are determined, as explained as point 2 on page 1 of this Response to Reviewer (RtR). This approach allows projection of water age west of Greenland where CFC measurements were not available.

We could understand the reviewer's concern about the ISOW and DSOW ages not showing an increase in the eastern section. The approach we adopted to calculate CFC ages determines that the aging of water is due to the change in water mass composition (i.e. the mixture of water masses), rather than the increase of water spreading time.

Unlike the CFC apparent age calculated based on pCFC and the atmospheric history of CFC, the TTD approach inherently solves the mixing bias problem and is not limited by the fact that atmospheric history of CFC only started in 1930s. However, our age model does work better in the young age range between 0-80 years, and we restrict our discussion within that age range.

We agree that there exists a bias due to mixing of waters of different ages for CFC concentration age calculated based on the atmospheric history of CFC. CFC-based ages in our paper however is computed with TTD method and takes into account advection and diffusion in the mathematical framework it employs and therefore will not be biased due to mixing. As we combine TTD-derived CFC age for each SWT with water mass composition, the water mass mixing of the CFC ages is linear (unlike the CFC concentrations). This addresses the reviewer's concern that CFC ages are biased due to mixing from this perspective. There is also no bias for 230Th as the mixing is linear.

Fig. 6: Adding water mass contour lines would be helpful

Author's response:

As above, we have added water mass contour lines.

Line 16; p6 "Preformed component" "..in the absence of measurement at the exact location of deep water formation: : :".

While this statement is correct for the formation of DSOW and ISOW, I would say that station 69 is essentially at the location of LSW formation.

**Author's response:**

We cannot totally agree with the reviewer that station 69 is the location of LSW formation. Pickart et al. (2003) for instance suggested that Labrador Basin is not the only Basin of LSW formation, and that Irminger Basin is a second formation site. We agree though that the wording needs some correction to make it clear that the preformed value of 230Th and 231Pa is very uncertain.

"..we set the preformed value as the average of concentrations measured in waters <100m depth for this section.."

I think the authors need to elaborate on their rationale for doing this. It is not like deepwater is formed uniquely from water sinking from a depth < 100m. Deep water convection homogenizes the water column in the Labrador and Nordic seas, which then spreads laterally at depth. Therefore, the homogenized water column at the sites of deepwater formation could be the starting point providing preformed concentrations. If so, data from station 69, as well as earlier data from Moran et al indicate that preformed Th-230 should be about 0.3 dpm/1000l both for the Labrador and Nordic seas, instead of 0.1 dpm/1000l. Such value for the Nordic Sea (based on one profile from Moran et al., 1995) is of course problematic since it is higher than measured here in the IC basin. In addition, Moran's data from the Labrador Sea shows substantial intern-annual variability. I think the authors should be more nuanced in their choice of preformed values and consider how the uncertainties on this number could affect their conclusions.

Alternatively, the reasoning followed by the authors may be that cooling happens at the surface, and therefore that must be the starting point, and the higher Th concentrations found in the water below at the sites of deep water formation reflects the residence time of this surface water in the convection cell. If this is the case, that should be more explicit in the paper. I would also use the available data for surface water in the Labrador and Nordic Seas only, which are available from this study and Moran's.

**Author's response:**

1. We did not mean to give the impression that deep water is formed uniquely from water sinking from a depth <100m, nor that the values we selected for preformed values are necessarily correct. An important conclusion from our work is that the preformed values of 231Pa and 230Th make a significant and long-lasting difference to the evolution of the 231Pa/230Th ratio as waters then age. Our selection of a particular preformed values serves to illustrate this point, rather than being a statement about true preformed values. It is, of course, also possible that the various water masses have different preformed values, making the use of a single value for the whole section inappropriate if seeking a "correct" value. We have made this logic clearer during revision; that we are not implying we know the preformed values, but that whatever they are will make a substantial difference to deep-water 231Pa/230Th.

Instead, as we pointed on page 6, the preformed concentrations of dissolved 230Th and 231Pa at the time the water masses formed are unknown. We can only assume that the preformed concentrations are either equal to average concentrations in surface waters sampled along the GEOVIDE transect or that preformed concentrations are zero, until data are available for the Nordic Seas and the Labrador Sea during times of winter convection. The reason for the choice of < 100 m is that we consider <100 m as the surface mixed layer, where Pa and Th is well mixed.

2. This is why we discussed the effect of preformed values on the evolution of Pa and Th with water age in section 4.3. We have also added how preformed values might affect our calculation of scavenged components.

Line 30; p6: as per discussion above, calculated potential total concentration should be viewed as minima (because CFC ages underestimate the Th and Pa "ages")

Author's response:

We hope that our opening comments above clarify that there is no age bias in our CFC ages calculated with TTD approach.

Line 4; p7: "..is the net of nuclide added from above by [add: DESORPTION FROM] settling particles: : :"

Author's response: Changes are made as suggested.

Line 8-9; p7: Considering that the CFC ages are younger than the Th ages, potential and scavenged Th-230 must be viewed as minima. Also, the underestimation of fraction scavenged is larger for Th than Pa.

**Author's response:**

We hope that our opening comments above clarify that there is no age bias in our CFC ages calculated with TTD approach.

Line 14; p7: I guess the authors deduce that about 75% of the Th-230 produced is scavenged because the slope of slope of observed Th-230 vs time is 25% of the rate of production, right? If so, it would be helpful to be explicit and to report the value of the slope in Fig. 8a.

**Author's response:**

This is how we deduce the number. We have added the linear fit equations, giving the slope, on the figure.

Wouldn't the intercept also provide an independent estimate of preformed? I am not sure what the intercept on scavenged Th-230 figure means.

**Author's response:**

We agree with the reviewer that the intercept does provide some estimate of the preformed, especially for Th with a very good linearity. However, we would expect that at t=0, there is no scavenging taking place, and the y intercept of the scavenged Th figure should be 0, while that of the observed Th figure should be preformed value. However, we have a positive y intercept in scavenged Th figure, and a negative y intercept in observed figure. We attribute this to a combination of a lack of 230Th data in younger waters and the uncertainty associated with 230Th measurement and calculation.

I would also reiterate here that 75% scavenging is a minimum, and it could be that scavenging is more underestimated in older water than younger ones, producing this apparent intercept. It is not clear to me why the same can't be attempted for Pa, at least for the observed values.

Author's response:

As above, we do not think there is an age bias in our CFC ages based on TTD approach. We therefore do not think a change is required here.

Line 15-14; p7: "This ratio is consistent with the average 230Th for these waters, which requires that about three times more 230Th has been removed by scavenging" Line 18; p7:" : : :the average [Pa] values indicate that about half of the 231Pa remains in the water"

I can't follow the line of reasoning here. I think the authors need to be more explicit.

Author's response: From Fig. 8 (a) and (c), we have: Average of Th in these waters=  $3.24 \mu$ Bq/kg and average of Pa in these waters=  $1.51 \mu$ Bq/kg From Fig. 8 (b) and (d), we have: Average of scavenged Th = 11.8Average of scavenged Pa = 1.02

To maintain an average Th of 3.3  $\mu$ Bq/kg in water requires 12.1  $\mu$ Bq/kg to be scavenged, which is three time more Th to be removed by scavenging. Similarly, to maintain an average of Pa 1.5  $\mu$ Bq/kg in water requires 1.2  $\mu$ Bq/kg to be scavenged. That is, half of the 231Pa remains in water and half removed by scavenging.

P2; line 23: ".. there is no simple relationship between increasing Pa/Th and age, as would be expected"

P7; Line 21; p7:"..The hypothesis that Pa/Th ratios increase as water mass ages.."

We should not expect seawater Pa/Th to simply increase with water mass age. If dissolved Th and Pa activities are initially low, Pa/Th in the water column (and underlying sediments) should initially decrease with age of water mass because Th grows in faster than Pa. It's only once Th has regained levels closer to equilibrium with scavenging (and therefore grows more slowly) that Pa starts growing faster, resulting in increasing Pa/Th. This effect is clearly illustrated in the paper by Luo et al. (2010) (see Fig. 14 in the paper). Fig. 8e,f may show a hint of this initial decreasing trend. (Luo et al also argued "Clearly, it is impossible to constrain the history of changes in the AMOC from the evolution of 231Pa/230Th at one site, as was attempted by McManus et al.; p 395; last sentence of first paragraph).

**Author's response:**

We agree with the reviewer that we should not expect seawater Pa/Th to simply increase with water mass age. The model results plotted on Fig. 8e,f indicate an expectation that Pa/Th ratios can either increase or decrease in the first 10 to 15 years, depending on the preformed values. Beyond this age, Pa/Th ratios increase with age before reaching equilibrium. We have made this clearer in the manuscript accordingly.

Section 4.3: I am puzzled by Fig 9. The intercepts don't seem to match with the

**preformed values. The fit with the data is terrible. I am not sure what this section is telling us.**

**Author's response:**

As in our opening comment 3: We thank the reviewer for pointing out this issue, which contributed to our reconsideration of the model and removal of the incorrect surface term. This change gives Y intercept matching the preformed values at t=0.

We are aware of the 'terrible' fit of the model with data, even in the new model with surface term removed. It however supports what we aim to convey in this section that preformed values can influence the evolution of Pa, Th and Pa/Th with water mass age. Without a better knowledge of preformed values, we cannot give a simple description of how Pa/Th evolve with water mass age, and the model cannot generate profiles that better fit observations.

**Line 28; p7: wouldn't it be better to follow density rather than depth?**

**Author's response:**

We followed depth as the model is a function of depth, reflecting that the scavenging of 230Th and 231Pa takes place with particles sinking with depth instead of density. This is also the way that 230Th and 231Pa data have traditionally been considered.

Line 4; p8: why did you choose station 1 and 13? Is it because they are the least ventilated and therefore these parameters could be estimated from a linear fit? If this is the case, that should be indicated. However, is it reasonable to apply scavenging parameters from a margin to open ocean stations?

**Author's response:**

We chose station 1 and 13 as their profiles are the closest to the equilibrium profile with linear increase with water depth. We however realized, as the reviewer pointed out, that Station 1 is close to the margin, on the continental slope, and therefore used station 13 only, an open ocean station, to optimize model parameters.

Fig. 9: How can Pa/Th (zero preformed) increase with age (c) if Th (a) grows faster than Pa (b)?

**Author's response:**

With zero preformed Pa/Th is expected to increase from first ventilation. Despite the fact that Pa approaches an equilibrium value more slowly than Th, the Pa concentration increases more than Th concentration, i.e., more of the Th produced by decay is removed by scavenging than for Pa, at each time step. So the Pa/Th ratio increases monotonically from T=0.

**Section 4.4:**

This discussion should take into account that scavenged Th and Pa are underestimated and proportionally more for Th than Pa. Since potential and scavenged are underestimated by exactly the same absolute amount, calculated scavenged/potential ratios should also be taken as minima.

**Author's response:**

As above, we do not think there is underestimation of scavenged Th and Pa due to bias in CFC ages and Th ages as a result of mixing.

P 9; line 20-22; I think it would be helpful for the reader to provide the details of this mass balance calculation. For instance, what volume for the North Atlantic did you use to calculate production?

**Author's response:**

As we followed the Deng et al. (2014) for the calculation and used the number for the volume of the North Atlantic in that paper, which is referred to in the discussion manuscript, we did not give details here. However, we understand the reviewer's point that it would be helpful for readers so we now repeat this information in the revised supplementary material.

It seems that the results from this mass balance corroborate that scavenged (and therefore potential) Th and Pa calculated from equation 4.3 are underestimated because of the use of CFC ages. I am a little surprised that so much Pa is scavenged in the North Atlantic. If 77% of the production is scavenged, then the average Pa/Th of N. Atlantic sediments should be ca.0.073, which is higher than measured in the central N. Atlantic basin. That means that boundary scavenging must be more significant than I thought.

**Author's response:**

The 96% scavenged relative to produced for Th and 77% for Pa obtained from the mass balance calculation represents a basin-scale average in the North Atlantic. The lower scavenged to produced values from equation 4.3. represent more localized results when water masses are young, and nuclide concentrations and scavenging correspondingly low.

**Roger Francois**

**Anonymous Referee #2**

Here Deng et al. provide a very nice and concise piece of science. They examine crucial assumptions made for the application of 231Pa/230Th as an AMOC proxy by providing an extensive new data set of water 231Pa and 230Th concentrations. This is hard won data and the authors deserve credit for their efforts. The new data extends the former GEOTRACES transects by Hayes15 and Deng14 towards the northern North Atlantic representing a definite reality-check for the assumptions made when using 231Pa/230Th as a proxy. These assumptions have been made based on the elegant approach of measuring a kinetic tracer with a constant and well-defined inputcfunction not involved in the carbon-cycle (Yu1996). While previous studies already proofed the consistence and capability of 231Pa/230Th as an AMOC proxy the novelty of this study is the systematic examination of the behaviour of 231Pa and 230Th in the northern North Atlantic in the water masses recently influenced by NADW formation with a new set of samples and by state-of-the art analytical methods. Therefore, this manuscript certainly deserves publication in Biogeosciences. Given the published results from the 231Pa/230Th proxy of the last decades I would have wondered if this study would have come to a different conclusion. But here Deng et al. make very good cases by confirming the prerequisites for using 231Pa/230Th as AMOC proxy. The only little weakness of the manuscript is the missed opportunity of setting the new findings into the context of the attempts of using 231Pa/230Th as a large scale AMOC proxy. There are several of papers dealing with a single 231Pa/230Th profile from on location, but the results of the few which deal with comprehensive compilation approaches could be better assessed and discussed here. Besides the already mentioned Yu et al. 1996, I think in particular of Bradtmiller et al. 2014, which use the large scale 231Pa deficit for analyzing the HS1 AMOC. It would be worth of shortly recapitulate their results in the light of the new results presented here. Besides the connection to observational paleo data I also miss the comparison to theoretical or model studies. The authors should in particular present a short comparison to the predictions made by Marchal et al. 2000 and the most recent attempt by Rempfer et al. 2017. Further, most of the features reported here have been anticipated by the simple box-model approach by Luo et al. 2010. They already found a very weak correlation of 231Pa/230Th with water mass age, highlighting a vertical gradient not a horizontal in the presence of an active AMOC (see specific points below).

Thank you for suggestions of references to be included. We agree, and have included them during a rewrite of the introduction.

page 3, line 2: recurring typo of "R/V Pourquoi Pas?"

Author's response:

"R/V Pourquoi Pas?" is the correct spelling for the name of the French research vessel which undertook sampling in this study.

page 3, line 15: please specify how many several months are Author's response:

We have specified that it is four to five half-lives of  ${}^{233}$ Pa (t1/2=26.98 days, Usman and MacMahon, 2000) after spike production.

page 3, line 19: what was the analytical yield (range)of the anion chromatography for Pa and Th?

**Author's response:**

Analytical yield ranges from approximately 41-91% for Th, and 30-52% for Pa.

page 3, line 25: what was the 232Th/231Pa in the Pa-samples? Was the correction for the 232ThH interference necessary, if yes, how big was the contribution to the Pa-signal?

**Author's response:**

232Th/231Pa in the Pa samples are at about 1800. 232ThH interference contributes 0.1% of 233Pa signal. From this perspective, it does not seem necessary to correct for ThH interference in our case. However, the Th signal in the Pa sample was not known before the Pa measurement and conducted in case Th was not very well separated from Pa.

page 5, line 14: ISOW is mentioned but not shown in Fig. 5.

**Author's response:**

Thank you for pointing out. We have labelled ISOW in Fig.5.

**Supplement: please add a column specifying the errors to the given concentrations**

Author's response:

We assume that the reviewer meant Table S1 in the supplement. In this table, it is the 232Th-corrected Pa and Th concentrations that are used for the discussion. We therefore have only given errors (2se) associated with these concentrations, and do not think it is necessary to report errors for the measured concentrations without correction.

Supplement page 7 line 7: I'm aware that this is of marginal importance given the final result, but maybe the authors could elaborate on the value 0.7 in (3) and (4). 0.7 seems a little bit high for an average, or at least unnecessarily high at the high end of the possible range according to Henderson and Anderson 2003 or Bourne et al. 2012. Further, is the detrital correction required for particles slipping through the 0.45 m filters? Is the added HCI capable of leaching of the 232Th (and 230Th and 231Pa) from these particles?

**Author's response:**

We agree with the reviewer. Henderson and Anderson (2003) suggested average of 238/232 activity ratio to be  $0.6\pm0.1$  in the Atlantic, and  $0.7\pm0.1$  in the Pacific Ocean. We therefore have adopted the value of 0.6 in the revised paper as 238U/232Th activity ratio to correct for the detrital contribution of 231Pa to 230Th. Equation (3) and (4) were rewritten and the data were recalculated accordingly.

Detrital correction is to correct for the contribution from the partial dissolution of lithogenic minerals to 230Th pool in seawater, rather than in the sample after being collected. It is therefore necessary regardless of the filtration and the acidification.

**Fig. 3,8,9: error bars are missing**

Author's response:

We have added error bars to the measured data in these figures.

page 7, line 21: It is not surprising that 231Pa/230Th does not correlate with water mass age very much. This has been already predicted by Luo et al. 2010. Much more important is the vertical decrease within one circulating mater mass (e.g. Burckel et al. 2016). Thus, the sentence that "231Pa/230 ratios increase as water mass ages forms the foundation of using 231Pa/230Th in discrete cores" is not completely accurate.

**Author's response:**

Thank you for pointing this out. We found this comment very helpful to improve our paper. We agree with the reviewer on the relationship between 231Pa/230Th and water mass age. We have clarified the two conceptual models forming the foundation of the interpretation of Pa/Th in terms of rates of deep water circulation (as explained in point 1 of our opening comments above).

page 10, line 13: typo. two times "demostrates".

Author's response: We have corrected the typo.

Fig. 6: I assume the x-axis has changed between (a) and (b), but they are both shown on the same longitudinal scale.

Author's response:

Thank you for pointing out this error. We have corrected the longitudinal scale for Fig.6 (a) in the revised manuscript.

Fig. 3: maybe it would be worth of showing 231Pa/230Th as well in an additional panel

Author's response:

We do not think it is not necessary as we focus on discussing the latitudinal gradient of 231Pa and 230Th rather than 231Pa/230Th at this stage.

(c). Fig. 8: please indicate water depth at the colour bar.

Author's response:

We have added to the colour bar the water depth (m).

References: Bourne, M., et al., 2012. Improved determination of marine sedimentation

rates using 230Thxs. Geochemistry Geophysics Geosystems 13.

Bradtmiller, L., et al.,2014. 231Pa/230Th evidence for a weakened but persistent Atlantic meridional overturning

circulation during Heinrich Stadial 1. Nature Communications 5.

Burckel, P., et al., 2016. Changes in the geometry and strength of the Atlantic Meridional Overturning

Circulation during the last glacial (20-50 ka). Climate of the Past 12.

Deng, F., et al., 2014. Controls on seawater 231Pa, 230Th and 232Th concentrations along the flow paths of deep waters in the Southwest Atlantic. Earth and Planetary Science Letters 390.

Hayes, C., et al., 2015. 230Th and 231Pa on GEOTRACES GA03, the U.S. GEOTRACES North Atlantic transect, and implications for modern and paleoceanographic chemical fluxes. Deep Sea Research Part II: Topical Studies in Oceanography 116.

Henderson, G., et al., 2003. The U-series toolbox for paleoceanography, Uranium Series Geochemistry. Reviews in Mineralogy and Geochemistry 128.

Luo, Y., et al., 2010. Sediment 231Pa/230Th as a recorder of the rate of the Atlantic meridional overturning

circulation: insights from a 2-D model. Ocean Science 6.

Marchal, O., et al., 2000. Ocean thermohaline circulation and sedimentary 231Pa/230Th ratio. Paleoceanography

15.

Rempfer, J., et al., 2017. New insights into cycling of 231Pa and 230Th in the Atlantic Ocean. Earth and Planetary Science Letters 468.

Yu, E., et al., 1996. Similar rates of modern and last-glacial ocean thermohaline circulation inferred from

radiochemical data. Nature 379.

**Anonymous Referee #3**

Deng and coworkers have produced an important data set by analyzing samples collected on GEOTRACES Section GA01 (GEOVIDE) for 231Pa and 230Th. These results hold valuable implications for the use of these radionuclides as tracers of North Atlantic deep water ventilation, and its variability through time (via the analysis of 231Pa/230Th ratios, henceforth "Pa/Th", archived in marine sediments). However, there are some major issues that should be addressed before I can recommend that the manuscript be published, as detailed in the following.

**Major Comments:**

Clarify and emphasize the principal take home message. The concluding sentence of the manuscript states "and continues to support the use of sedimentary 231Pa/230Th measurements at a basin scale to constrain overturning circulation." This statement is based on the calculated southward export of dissolved Pa being substantially greater than the southward transport of Th. However, as clearly stated in the manuscript, there is no observable relationship between dissolved Pa/Th ratio and water mass age. This observation is in direct contradiction to the principles underlying the use of sedimentary Pa/Th ratios to reconstruct past variability of the ventilation of deep water in the North Atlantic Ocean, where it is assumed that dissolved Pa/Th ratios will increase monotonically with age after water mass formation due to the longer residence time of Pa compared to Th. How can the authors conclude that their results support the use of sedimentary Pa/Th ratios to constrain overturning circulation when there is no relationship between dissolved Pa/Th ratio and water mass age? This issue becomes even more important if one considers the evolution over time of dissolved Pa/Th ratios down the length of the western Atlantic Ocean. Although the authors do not present Pa/Th ratios for the mid-latitude North Atlantic or at 40S along with the dissolved 230Th and 213Pa data in Figure 3, eyeballing the dissolved 230Th

and 231Pa profiles for these regions suggests very little change in the dissolved Pa/Th ratio from north to south, from GEOVIDE near the formation region to GA10 at 40S. If a more rigorous analysis of the data reveals this to be true, i.e., that there is no change with water mass age in the dissolved Pa/Th ratio down the entire length of the Atlantic Ocean, then I do not see how Pa/Th ratios can be related to ventilation rate, either in the modern ocean or to reconstruct climate-related changes in ventilation rate in the past.

The manuscript would have much greater impact if this point were discussed at length, incorporating data from the entire Atlantic Ocean.

**Author's response:**

We have taken on board the reviewer's comments here. As explained in point 1 of our opening comments above, we have now clarified that there are two conceptual models that form the foundation of the interpretation of 231Pa/230Th. This has significantly helped us to make clear the take-home messages in the conclusion section.

The development and application of CFC ages are unclear. The description in the Supplementary material (page 8) is helpful, but some of the output is not meaningful. CFC ages are not valid for time periods older than the initial introduction of CFCs into the environment in the middle of the 20th century. My colleagues who are experts in the use of CFC ages generally decline to interpret apparent ages greater than about 40 to 50 years due to the uncertainties inherent in interpreting CFC ages in water masses last exposed to the atmosphere during the earliest days when CFCs were tagging water masses. Therefore, I do not understand how Mediterranean Water can be assigned an age of 91±8 years, or NEADW can be assigned an age of 989±48 years (Table S2). Unless there is something not explained in the paper that allows CFC ages this old to be computed, the old ages should be removed from the paper. Accordingly, Figure 6a can be removed, leaving only Figure 6b in the paper.

Related to Figure 6, it is very confusing that the two panels have different longitude scales, but the scale for Figure 6a is not shown. If there is a reason to retain Figure 6a, then include the longitude scale and note that Figure 6b incorporates only the western half (approximately) of Figure 6a.

**Author's response:**

We have corrected the missing longitude mistake.

Thank you for suggesting clarification for the calculation of CFC-based ages. We have included more detailed information describing how CFC-based ages were calculated and uncertainties associated with it, as explained as point 2 on page 1 in this RtR.

The calculation and interpretation of scavenging (rates and percentages) relies strongly on the estimated CFC ages (Figures 7 through 9). Given this important sensitivity to estimated age, I recommend that the authors include a discussion of the uncertainty in the CFC ages, and how that may affect their interpretation.

**Author's response:**

Thank you for the suggestion. We have included the uncertainty of the CFC ages and how they affect our interpretation as supplemental information in the revised manuscript.

As discussed on page 6 of the manuscript, the initial (preformed) concentrations of dissolved 230Th and 231Pa at the time the water masses formed are unknown, so the authors assume that the preformed concentrations are either equal to average concentrations in surface waters sampled along the GEOVIDE transect or that preformed concentrations are zero. Unfortunately, until data are available for the Nordic Seas and the Labrador Sea during times of winter convection, these may be the only options available for the type of analysis described here. Nevertheless, it would be helpful if the authors provided additional discussion of the sensitivity of their derived products (e.g., fraction scavenged for each isotope) to the values assumed for the preformed concentration.

**Author's response:**

**Thank you for this comment. We have added the suggested sensitivity analysis in the Supplemental Information.**

In this context, it would also be helpful to discuss the possibility that each water mass has a different preformed concentration, and how this might affect the interpretation of the data presented in Figure 8. Implicit in the presentation of the data presented in Figure 8 is the assumption that all water masses have the same preformed concentrations. What if this is not the case? How would that alter the interpretation of the data?

**Author's response:**

Thank you for your comment. This is a good point, and we have made clear that our results are based on the assumption of the same preformed value for different water masses, and addressed the influence of this assumption based on our discussion of how preformed values affect the results.

There seems to be a problem with the model curves shown in Figure 9, where the solid lines depict model results for the case where average surface water concentrations determined for samples collected on the GEOVIDE cruise were used in place of the preformed concentrations. If that were the case, then why do the projected model concentrations at zero age (solid lines) intersect the Y axis at concentrations about double the values reported for average surface concentrations on page 6 (0.108 dpm/1000 L for 230Th and 0.089 dpm/1000 L for 231Pa)? If I understand the model correctly, then the concentrations at zero age should equal the assigned preformed concentrations. Is this not the case? Is the problem that the preformed concentrations are introduced twice in equation 6 (supplementary material)? Note that C(pre) and C(surface) are one and the same. Should both terms be in equation 6?

**Author's response:**

Thank you for pointing this out. We ran some tests with the model and realized that introducing the surface term in the model is the reason causing the 231Pa and 230Th concentration twice of the preformed values at zero age. Considering this weakness in the model, we removed the surface term in the model as explained as point 3 of our opening comments. The new model gives y intercepts at preformed values at t=0.

Summary of major comments: Given all of the uncertainties in CFC age and in initial (preformed) concentrations of Pa and Th, it seems that a stronger paper than the one under review would be produced by integrating the new data from GEOVIDE with other data from GEOTRACES cruises down the length of the Atlantic Ocean (GA02, GA03, GA10 and, perhaps, other sections with data in the GEOTRACES IDP2017, if there are any) to establish firmly whether or not the dissolved Pa/Th ratio in deep Atlantic water evolves over time as assumed in the application of sedimentary Pa/Th ratios to constrain past changes in the rate of ventilation of North Atlantic Deep Water.

**Author's response:**

We agree that it would be great to integrate all the 231Pa and 230Th data available from GEOTRACES in the Atlantic Ocean. We are thinking of putting together a paper with that aim. With the present manuscript paper, however, we would prefer to focus on the data from GEOVIDE and how this new addition of data can provide evidence to assess the conceptual models adopted for the interpretation of 231Pa/230Th. However, we continue to include data from GA02 and GA03 to put our new GEOVIDE data in the broader context and to show horizontal gradient of 231Pa and 230Th in the Atlantic.

Minor comments:

The authors report their results using historical units (dpm/1000 L) but their results will be converted to SI units when included in the next GEOTRACES data product. All of the Th and Pa data currently in the IDP2017 are presented using SI units, so why not make this conversion before publishing the GEOVIDE data?

Author's response:

We have reported our data in SI units adopted in the GEOTRACES data product as suggested.

page 6 line 1 "Steinfeldt" is misspelled.

Author's response:

We have corrected the misspelling.

page 6, definition of "Ingrown component": Are U concentrations normalized to a constant salinity? To the salinity measured for each sample? Something else?

Author's response:

We used constant U activities for all the sample when calculating the "ingrown component", and calculated 238U following the equation in Owens et al., 2017, assuming a salinity of 35 permil. U-235 and U-234 activity was then calculated assuming natural abundance ratio of 238U/235U=137.88 and the 234U/238U activity ratio in seawater is ~1.15.

page 7 line 16: "three times more 230Th has been removed by scavenging" than "what?" Complete the description of the comparison being made.

Author's response: We have added: '... more than it remains in the water...'

page 8 line 7: Change "run" to "ran"

Author's response: We have made changes as suggested.

page 8 line 27: delete the "in" prior to "from DSOW"

Author's response:

**We have made changes as suggested.**

page 8 line 28, and elsewhere: Bottom scavenging of 230Th was first noted by Bacon and Anderson (1982) and by Anderson et al. (1983; EPSL 66(1-3), 73-90, not the paper cited by Deng et al.) in their study of the eastern tropical Pacific. These early indications of bottom scavenging should be cited.

Author's response:

We have included the suggested reference.

page 9 line 12: What is the source of the average 230Th and 231Pa concentrations in the upper limb? The values given here are not those given on page 6 for GEOVIDE surface waters, so the source should be given.

Author's response:

Here, the boundary between upper and lower was defined by potential density at 32.15 kg/m3 as described in the discussion manuscript on page 9 line 10. The average 230Th and 231Pa in the upper and lower limb are average concentrations of these nuclides in water with potential density < 32.15 kg/m3 and > 32.15 kg/m3, respectively.

page 10 line 7: Change "that" to "than" i

Author's response: We have made changes as suggested.

**Additional references (added by authors ):**

Bradtmiller, L. I., McManus, J. F. and Robinson, L. F.: 231Pa/230Th evidence for a weakened but persistent Atlantic meridional overturning circulation during Heinrich Stadial 1, Nat. Commun., 5, 5817 [online] Available from: http://dx.doi.org/10.1038/ncomms6817, 2014.

Negre, C., Zahn, R., Thomas, A. L., Masqué, P., Henderson, G. M., Martínez-Méndez, G., Hall, I. R. and Mas, J. L.: Reversed flow of Atlantic deep water during the Last Glacial Maximum, Nature, 468, 84 [online] Available from: http://dx.doi.org/10.1038/nature09508, 2010.

Owens, S. A., Buesseler, K. O. and Sims, K. W. W.: Re-evaluating the 238U-salinity relationship in seawater: Implications for the 238U–234Th disequilibrium method, Mar. Chem., 127(1), 31–39, doi:https://doi.org/10.1016/j.marchem.2011.07.005, 2011.

Pickart, R. S., Straneo, F. and Moore, G. W. K.: Is Labrador Sea Water formed in the Irminger basin?, Deep Sea Res. Part I Oceanogr. Res. Pap., 50(1), 23–52, doi:https://doi.org/10.1016/S0967-0637(02)00134-6, 2003.

Usman, K. and MacMahon, T. D.: Determination of the half-life of 233Pa, Appl. Radiat.

lsot., 52(3), 585–589, doi:https://doi.org/10.1016/S0969-8043(99)00214-6, 2000.

**Evolution of 231Pa and 230Th in overflow waters of the North Atlantic**

Feifei Deng1, Gideon M Henderson1, Maxi Castrillejo2,3, and Fiz F. Perez4 and Reiner Steinfeldt5

1Department of Earth Sciences, University of Oxford, South Parks Road, Oxford, OX13AN, UK.

2LaboraLaboratory of Ion Beam Physics, ETH-Zurich, Otto Stern Weg 5, Zurich, 8093, Switzerland

3Institut de Ciència i Tecnologia Ambientals & Departament de Física, Universitat Autònoma de Barcelona, Bellaterra, 08193, Spain

4Departamento de Oceanografia Instituto Investigaciones Marinas (CSIC), Eduardo Cabello 6, E36208 Vigo, Spain. 5Institut fur Umweltphysik, Universitat Bremen, D-28334 Bremen, Germany.

Correspondence to: Feifei Deng (feifei.deng@earth.ox.ac.uk)

5

- 10 Abstract. Many paleoceanographic studies have sought to use the 231Pa/230Th ratio as a proxy for deep ocean circulation rates in the North Atlantic. As yet, however, no study has fully assessed the concentration of, or controls on, 230Th and 231Pa in waters immediately following ventilation at the start of Atlantic meridional overturning. To that end, full water-column 231Pa and 230Th concentrations were measured along the GEOVIDE section, sampling a range of young North Atlantic deep waters. Th-230 and 231Pa concentrations in the water column are lower than those observed further south in the Atlantic, ranging
- 15 between 0.06 and 12.01 μBq/kg, and between 0.37 and 4.80 μBq/kg, respectively. Both 230Th and 231Pa profiles generally increase with water depth from surface to deep water, followed by decrease near the seafloor, with this feature most pronounced in the Labrador Sea (LA Sea) and Irminger Sea (IR Sea). Analyzing this dataset with Extended Optimum Multi-Parameter (eOMP) Analysis and CFC-based water mass age indicates that the low values of 230Th and 231Pa in water near the seafloor of the LA Sea and IR Sea are related to the young waters present in those regions. This importance of water age is
- 20 confirmed for 230Th by a strong correlation between 230Th and water mass age (though this relationship with age is less clear, for 231Pa and 231Pa/230Th ratio). Scavenged 231Pa and 230Th were estimated and compared to their potential concentrations in the water column due to ingrowth. This calculation indicates that more 230Th is scavenged (~80%) relative to 231Pa (~40%), consistent with the relatively higher particle-reactivity of 230Th. Enhanced scavenging for both nuclides is demonstrated near the seafloor in young overflow waters. Calculation of meridional transport of 230Th and 231Pa with this new GEOVIDE dataset
- 25 enables a complete budget for 230Th and 231Pa for the North Atlantic. Results suggest that net transport southward of 230Th and 231Pa across GEOVIDE is smaller than transport further south in the Atlantic, and indicates that the flux to sediment in the North Atlantic is equivalent to 96% of the production of 230Th, and 74% of the production for 231Pa. This result confirms a significantly higher advective loss of 231Pa to the south relative to 230Th and supports the use of 231Pa/230Th to assess meridional transport at a basin scale.
- 30 Key words. GEOTRACES; water-column 230Th and 231Pa; water mass ageing; scavenging; meridional transport.

| Delet | ted: ,                                                   |
|-------|----------------------------------------------------------|
| Delet | ted: their                                               |
| Delet | ted: Potential Total concentrations in the water column. |
| Delet | ted: The result shows                                    |

[revised manuscript text omitted]
. Deleted: Both 231Pa and 230Th are generated continuously by decay of uranium in the water column and are rapidly removed, leading to very low concentrations at the surface and increasing concentrations with depth due to reversible scavenging (Nozaki et al., 1981). Advection of surface waters to depth transports the low concentrations into the deep ocean, where they subsequently increase due to scavenging from above at a rate dependant on the residence time of the nuclide. Because 231Pa has a longer residence time than 230Th (~130 years versus ~20 years, Henderson and Anderson, 2003), the concentration of 231Pa increases more slowly and its resulting removal to the seafloor also increases more slowly. This behaviour means that, in basins such as the North Atlantic with vigorous formation and advection of deep-waters, less 231Pa is removed to the sediment than is produced in the water column. Sediments underlying the North Atlantic will therefore have a 231 Pa/230 Th below the production ratio, with lower ratios caused by stronger advection. The use of 231Pa/230Th to constrain past North Atlantic overturning was first proposed by Yu et al. (1996) who measured 231Pa/230Th in Holocene and Last Glacial Maximum (LGM) sediments from many cores from the Atlantic and Southern Ocean. They found similar Holocene and LGM values at a basin scale, suggesting broadly similar overturning during the two periods. Later studies have generally relied on a smaller number of cores (e.g. McManus et al 2004; Gherardi et al. 2009; Roberts et al. 2014), chosen from particular depths to sample important water masses. This approach relies on the systematic evolution of the 231 Pa/230 Th ratio in each water mass

Recent water-column measurements of 231Pa and 230Th on GEOTRACES cruises have shed new light on the chemical controls on these isotopes in seawater. These have indicated that there is considerably more net advection of 231Pa than 230Th out of the North Atlantic (Deng et al., 2014), supporting the basin-scale application of 231Pa/J260Th as initially pursued (Yu et al. 1996). But these

measurements have also suggested that there is no simple relationship between increasing  $^{231}\text{Pa}^{/230}\text{Th}$  and age of water, as would be expected if the  $^{231}\text{Pa}^{/230}\text{Th}$  at single cores is to be interpreted as reflecting the rate of ventilation (e.g. Deng et al., 2014).  $\P$

**Deleted:** One important constraint on understanding 231Pa/230Th is to learn about the controls on these isotopes as deep waters form and enter the deep Atlantic. Some measurements have placed initial constraints on 231Pa and 230Th values in young North Atlantic deep waters (e.g. Moran et al., 1997, 2002; Rutgers van der Loeff and Berger, 1993), but there has not yet been a systematic study of the composition of waters in the far north Atlantic. The GEOVIDE cruise allowed waters to be collected for such a study, along a line where significant other data are available, both from that cruise and previous occupations of OVIDE. GEOVIDE provided an ideal opportunity to understand 231Pa and 230Th at the start of the ocean meridional overturning circulation. ¶

Once returned to the laboratory in Oxford, samples were weighed and then acidified with quartz distilled concentrated HCl to  $pH \sim 1.7$ , shaken and left for at least four days to ensure that Pa and Th was desorbed from the walls of the bottle. A mixed  $^{229}Th^{-236}U$  spike and a  $^{233}Pa$  spike were then added to each sample to allow measurement of Th, U (for another study), and Pa by isotope dilution MC-ICP-MS (Multi Collector-Inductively Coupled Plasma-Mass Spectrometry). The  $^{233}Pa$  spike was

- 5 freshly made by milking from 237Np (following Regelous et al., 2004) and calibrated against a known 236U solution after complete decay of 233Pa to 233U, i.e. four to five half-lives of 233Pa (tr2=26.98 days, Usman and MacMahon, 2000) after spike production (Robinson et al., 2004). 50 mg of pure Fe as a chloride solution was also added to each water sample. Samples were left overnight to allow for spike equilibrium after which the pH was raised to ~8.5 using distilled NH4OH to co-precipitate the actinides with insoluble Fe-oxyhydroxides. At least 48 hours were allowed for scavenging of the actinides onto Fe-
- 10 oxyhydroxides. The precipitate was centrifuged and rinsed, and Th, Pa and U were separated using anion exchange chromatography following Thomas et al., 2006.

After chemical separation, Pa and Th were measured on a Nu instrument MC-ICP-MS at the University of Oxford. Mass discrimination and ion-counter gain were assessed with the measurement of a U standard, CRM-145 U, before each sample measurement. Use of a U standard for this purpose minimises memory problems that might be caused by use of a Th or Pa

- 15 standard (Thomas et al., 2006). Measurements were also made 0.5 mass units either side of masses of interest to allow accurate correction for the effect of abundance sensitivity on small 231Pa and 230Th beams, and a correction for a small 232ThH interference on the 233Pa beam is made from assessment of the hydride formation rate on a 232Th standard. Concentrations of 231Pa, 230Th together with 232Th were obtained from the precise MC-ICP-MS measurement of 231Pa,230Th/229Th, and 232Th/229Th ratios together with well-calibrated concentrations of 233Pa, and 229Th-236U spikes.
- 20 Chemistry blanks were assessed by conducting the complete chemical procedure on ~100 ml of Milli-Q water with each batch of samples. Based on six blank measurements, the average blanks for dissolved 231Pa, 230Th and 232Th are 0.21±0.14 fg, 1.59±0.60 fg and 5.13±1.47 pg, respectively (uncertainties are 2 standard errors). Blank contributions account for 2-22%, 2-26%, and 0.2-16% of the dissolved 231Pa, 230Th and 232Th respectively (with the higher values being for surface samples due to their low concentrations).

**25 3 Results**

Measured 230Th and 231Pa concentrations were corrected for blanks, ingrowth from U in seawater since the time of sample collection, and detrital U-supported 230Th and 231Pa concentrations. Measured and corrected concentrations of 230Th, 231Pa, and 232Th, along with details of corrections, are provided in the Supplemental Information S1. Although analysis was conducted in terms of fg/kg, results are converted to the S1 units adopted by GEOTRACES data product, i.e., µBq/kg for 230Th and 231Pa.

30 and pmol/kg for 232Th, This conversion uses half-lives for 231Pa, 230Th and 232Th of 32,760 yr, 75,584 yr and 1.405×1010 yr, respectively (Robert et al., 1969; Cheng et al., 2013; Holden, 1990). Uncertainties were propagated, including the contribution from sample weighing, spike calibration, impurities in the spikes, blank corrections, and mass spectrometric measurement,

and are reported as 2 standard errors (2 s.e.). Average total uncertainties for  $^{231}$ Pa,  $^{230}$ Th and  $^{232}$ Th are  $\pm 0.17 \mu$ Bq/kg,  $\pm$

Th-230 concentrations in the water column range between 0.06 and 12.01  $\mu$ Bq/kg and initially generally increase with water 5 depth from surface to deep water. Towards the seafloor, six of the eleven stations show a prominent decrease of 230Th, with this feature most pronounced in the LA and IR Seas.

Pa-231 concentrations in the water column range between 0.37 and 4.80  $\mu$ Bq/kg and also increase with water depth, but less rapidly than 230Th. 231Pa profiles also often exhibit a decrease near the seafloor at stations showing a 230Th decrease. Station 38 at the Reykjanes Ridge distinguishes itself from other 231Pa profiles in that an increase in 231Pa concentrations from low

10 concentrations at 1000 m is observed, continuing towards the bottom. Observed 230Th and 231Pa values at GEOVIDE are lower than those observed in inter-calibrated GEOTRACES data from further south in the Atlantic. Figure 3 compares average depth profiles for 230Th and 231Pa in the west Atlantic, covering highlatitude Northwest Atlantic (from GEOVIDE, west of the Mid-Atlantic Ridge), mid-latitude Northwest Atlantic (GEOTRACES section GA03\_w, Hayes et al., 2015) and Southwest Atlantic (GEOTRACES section GA02, Deng et al., 2014).

15 A southward increase of both 230Th and 231Pa concentrations is observed below 1000 m.

**4 Discussion**

[revised manuscript text omitted]

 ${}^{231}Pa_{lngrown} = {}^{235}U \times (1 - e^{-\lambda_{231}t})$ (4.2)

 ${}^{230}Th_{lngrown} = {}^{234}U \times (1 - e^{-\lambda_{230}t})$ (4.1)

where 230ThIngrown and 231PaIngrown are the 230Th and 231Pa ingrown from their U parents, respectively; 238U and 235U are activities of 234U, and 235U in seawater (45551.2  $\mu$ Bq/kg (2801.4 dpm/1000l) and 1823.8  $\mu$ Bq/kg (112.2 dpm/1000l), respectively, assuming a constant seawater 238U activity of 39609.8  $\mu$ Bq/kg (2436 dpm/1000l) at salinity 35 psu, and seawater 234U/238U activity ratio of 1.15 and natural 238/235U abundance ratio of 137.88);  $\lambda_{230}$  and  $\lambda_{231}$  are decay constants of 230Th and 231Pa (9.17

 $25 \times 10^{-6}$  yr-1 and  $2.12 \times 10^{-5}$  yr-1, respectively).

*iii. Potential Total component:* The 230Th and 231Pa expected in the water due to the combination of preformed and ingrown components, if there were no removal by scavenging.

*iv. Observed component*: The 230Th and 231Pa observed in the water column, i.e. dissolved 230Th and 231Pa measured in this study (after correction for detritus and ingrowth from U since sample collection).

[revised manuscript text omitted]

50 circulation.

25

4.5 Meridional transport of  $^{230}\mathrm{Th}$  and  $^{231}\mathrm{Pa}$  in the North Atlantic

| Deleted: at Station 1 and 13 on the east of the section (i.e. those with older waters) |
|-----------------------------------------------------------------------------------------------|
| Deleted: the                                                                                  |
| Deleted: 1 and                                                                                |
| Deleted: data                                                                                 |
| Deleted: 2.0                                                                                  |
| Deleted: We run the model twice                                                               |
| Deleted: $C_{pre} = 0$                                                                 |
| Deleted: of nuclide concentrations                                                            |
| Deleted: ¶                                                                                    |

Previous calculations have indicated removal of 230Th and 231Pa from the North Atlantic by meridional transport southward. Deng et al. (2014) calculated net southward transport of 6% of the 230Th and 33% of 231Pa, relative to production of these nuclides in the water column. That calculation, however, did not provide a complete budget for 230Th and 231Pa for the North Atlantic because observations at the time did not constrain input of these nuclides from the north. Data in this study allow this calculation, and therefore a more complete budget for the modern North Atlantic.

- García-Ibáñez et al. (2018) calculated volume transports for the Portugal to Greenland section of the GEOVIDE section by combining the water mass fractions from eOMP analysis with the absolute geostrophic velocity field calculated using inverse model constrained by Doppler current profiler velocity measurements (Zunino et al., 2017). They separated northward flowing upper, and southward flowing lower limbs of the AMOC at isopycnal  $\sigma_1$  (potential density referenced to 1000 dbar) = 32.15
- 10 kg/m3, with +18.7 ± 2.4 Sv and -17.6 ± 3.0 Sv flow across the section above and below this value (positive value indicates northward transport). With average 230Th and 231Pa concentrations in the upper limb ( $\sigma_{L} < 32.15 \text{ kg/m^3}$ ) of 1.60 and 1.32  $\mu$ Bq/kg respectively, northward transport of 230Th is 3.07 × 1010 and of 231Pa is 2.53 × 1010  $\mu$ Bq/s. Average 230Th and 231Pa concentrations in the lower limb ( $\sigma_{I} > 32.15 \text{ kg/m^3}$ ) are 3.44 and 2.07  $\mu$ Bq/kg, respectively, indicating transports of 230Th and 231Pa are -6.22 × 1010 and -3.74 × 1010  $\mu$ Bq/s, respectively.
- 15 Net transport of 230Th and 231Pa across GEOVIDE is therefore to the south, and supplies  $3.15 \times 10^{10} \mu$ Bq/s 230Th and  $1.21 \times 10^{10} \mu$ Bq/s 231Pa to the North Atlantic (Fig. 11). This is a smaller net transport than further south in the Atlantic (Fig. 11), due to the lower 230Th and 231Pa concentrations in the water column close of the site of deep-water formation. The budget for these nuclides for the North Atlantic consists of: production in the water column; addition by advection from the North; loss by advection to the South and removal to the sediment. The data from this study allows this budget to be fully
- 20 assessed, and indicates that the flux to the sediment is equivalent to 96% of the production of 230Th, and 74% of the production for 231Pa (Supplemental Information Table S5). For both nuclides, these fluxes are higher than in previous calculations (Deng et al. 2014) which ignored advective fluxes from the North. There is, however, still a significantly higher advective loss of 231Pa relative to 230Th. At a basin scale, therefore, 231Pa/230Th in the sediment must be lower than the production ratio. This lower value is generated by the meridional transport of the North Atlantic, and likely to be sensitive to changes in this transport.
- 25 Use the Basin-scale Advection model to interpret sedimentary 231Pa/230Th to assess meridional transport, as initially proposed by Yu et. al (1996), is therefore still supported by the full modern North Atlantic budget for these nuclides.

**5** Conclusion**

30

5

Measurement of 230Th and 231Pa in waters from GEOVIDE show some control of water mass on 230Th and 231Pa concentrations, particularly low concentrations in DSOW and high values in the old NEADW. There is, however, no close mapping of nuclide concentration to water mass.

With the availability of CFC ages on this section, the evolution of 230Th and 231Pa concentration with age is possible. A systematic increase of 230Th concentration is observed over the first 50 years following ventilation, and a similar though more

| Deleted: | defined | betwee |
|----------|---------|--------|
|----------|---------|--------|

| Deleted: | 0.098 and 0.082 dpm/10001                                               |
|----------|-------------------------------------------------------------------------|
| Deleted: | 1.83 × 10 6                                                  |
| Deleted: | 1.53 × 10 6 dpm/s                                            |
| Deleted: | 0.22 and 0.15 dpm/10001                                                 |
| Deleted: | are $-3.87 \times 10^6$ and $-2.64 \times 10^6$ dpm/s                   |
| Deleted  | $2.04 \times 10^6$ dnm/s 230 Th and $1.11 \times 10^6$ dnm/s |

|     | e   |      |     |   |
|-----|-----|------|-----|---|
| ·ſ  | Del | lete | ٠d: | 7 |
| - 1 |     |      |     |   |

scattered relationship seen for  $^{231}$ PavThere is no clear relationship between the  $^{231}$ Pa/ $^{230}$ Th ratio and age for these young waters. The long-term evolution of  $^{231}$ Pa/ $^{230}$ Th is found from a simple model to be highly dependent on the preformed concentrations for these nuclides. These results complicate the interpretation of sedimentary  $^{231}$ Pa/ $^{230}$ Th as a paleo-proxy for deep water circulation based on systematic evolution of water  $^{231}$ Pa/ $^{230}$ Th with age, and point to the importance of a better knowledge of

- 5 preformed 230Th and 231Pa concentrations to improve interpretation. This analysis of the 230Th and 231Pa concentration relative to the age of the water not only demonstrates the influence of water mass aging on 231Pa and 230Th, but also points to the influence of scavenging. Scavenged 230Th is much more extensive that 231Pa, as expected, and enhanced removal of both nuclides is seen immediately above the seafloor, particularly for young waters.
- Calculation of meridional transport of 230Th and 231Pa indicates a southward net transport of both nuclides across the GEOVIDE section. This advection is smaller than that further south in the Atlantic as a result of lower 230Th and 231Pa concentrations at GEOVIDE. Calculation of the flux across GEOVIDE allows a more complete budget for the North Atlantic to be constructed and demonstrates a significantly higher advective loss of 231Pa to the south relative to 230Th, with 26% of the 231Pa produced advected southward (relative to only 4% for 230Th). This calculation, supports the interpretation of sedimentary 231Pa/230Th measurements as a proxy for overturning circulation, when based on advective loss of 231Pa at a basin scale.

**15 Acknowledgements**

Géraldine Sarthou and Pascale Lherminier are thanked for leading the GEOVIDE cruise, along with the captain, Gilles Ferrand, and crew of the R/V Pourquoi Pas?. We would like to give a special thanks to Pierre Branellec, Floriane Desprez de Gésincourt, Michel Hamon, Catherine Kermabon, Philippe Le Bot, Stéphane Leizour, Olivier Ménage, Fabien Pérault and Emmanuel de Saint Léger for their technical expertise and to Catherine Schmechtig for the GEOVIDE database management.

20 This work was supported by the French National Research Agency (ANR-13-BS06-0014, ANR-12-PDOC-0025-01), the French National Centre for Scientific Research (CNRS-LEFE-CYBER), the LabexMER (ANR-10-LABX-19), and Ifremer. It was supported for the logistic by DT-INSU and GENAVIR. Thank you also to Yi Tang who helped with sampling during the cruise. We are grateful to Mercedes de la Paz Arandiga, and Pascale Lherminier for providing valuable input during analysis of water masses and ages, and to Maribel García-Ibáñez for early provision of the eOMP analysis presented elsewhere

25 in this volume. We also thank Yves Plancherel for valuable insight during discussion of the results presented.

**Deleted:**

| -( | Deleted: . This                                                                                                                                  |
|----|--------------------------------------------------------------------------------------------------------------------------------------------------|
| -( | Deleted: demonstrates                                                                                                                            |
| `( | Deleted: 3                                                                                                                                       |
| `( | Deleted: from the basin                                                                                                                          |
| Y  | Deleted: , and continues to                                                                                                                      |
| Y  | Deleted: support                                                                                                                                 |
| -( | Deleted: the use of sedimentary 231 Pa/ 230 Th measurements at a basin scale to constrain overturning circulation.¶ |
| Y  | Formatted: Space Before: 24 pt, After: 12 pt                                                                                                     |

Deleted: . Formatted: English (US) Anderson, R. F., Fleisher, M. Q., Biscaye, P. E., Kumar, N., Dittrich, B., Kubik, P. and Suter, M.: Anomalous boundary scavenging in the Middle Atlantic Bight: evidence from 230Th, 231Pa, 10Be and 210Pb, Deep Sea Res. Part II Top. Stud. Oceanogr., 41(2), 537–561, doi:https://doi.org/10.1016/0967-0645(94)90034-5, 1994.

5 Anderson, R. F., Fleisher, M. Q., Robinson, L. F., Edwards, R. L., Hoff, J. A., Moran, S. B., van der Loeff, M. R., Thomas, A. L., Roy-Barman, M. and Francois, R.: GEOTRACES intercalibration of 230Th, 232Th, 231Pa, and prospects: For 10Be, Limnol. Oceanogr. Methods, 10, 179–213, doi:10.4319/lom.2012.10.179, 2012.

[revised manuscript text omitted]

---

## Author Response (AR2)

**"Evolution of 231Pa and 230Th in overflow waters of the North Atlantic"**
**by Feifei Deng et al.**

**Response to referees**

We would like to thank all referees for their time reading the revised manuscript and giving constructive suggestions to further improve the paper. We are pleased that the referees appreciate our clarification and revision of the initial manuscript, and give favourable reports for the paper.

Below, we respond to the referees according to report 1 point by point. Reviewer's comments from report 1 are in blue, and our responses are in black.

Report 1:

Deng et al. have provided a thoughtful response to comments from the referees, and the authors have made helpful clarifications in revising their text. I recommend publication with just a few further changes.

The following comments refer to page and line numbers in the authors' response version of the revised manuscript.

1) Section 4.3, p. 8, around line 29: Although the model equation from Moran et al. (1997) is given in the Supplementary Material, it would be easier for readers to understand the findings described in the text if the model equation were included in the main text. It was not immediately obvious to me how the model curves in Figure 9 were obtained until I realized that the curves were derived by plugging the preformed concentrations, S, SPM and Kd into Moran's equation (originally from Rutgers van der Loeff?) along with the reference depth (either 2000 m or 3500 m) to obtain the curves as a function of water mass age. Anticipating that other readers may similarly find this hard to follow, I recommend that the equation be added to the main text so that readers can see immediately how the curves in Figure 9 were obtained.

Author's response:
Thank you for pointing this out. As suggested, we have added a brief description of the equation from Moran et al. (1997) in the main text, and have kept a detailed version of the description of the model and parameterization in the SI.

2) p. 9, lines 11-14: It is not only possible that scavenging of 230Th is more intense than would be inferred from the parameters derived using Station 13 data, the model-data offsets observed here require this. I suggest adding a sentence or two to speculate about why scavenging may be greater than at Station 13 at these locations.

Author's response:
Thank you for pointing this out. In the last revised manuscript, we suggested additional scavenging close to the seafloor in LA and IR Seas for deep water as a possible cause of the model-data offsets. But, we would add a sentence on p.9, line 23 to indicate the possibility of differences in productivity between stations causing some changes in scavenging as another cause.

3) p. 9, line 27: Delete "from" in "those in from DSOW"

Author's response:
The text in the last revised manuscript was "…, particularly those from DSOW in the deepest LA Sea". We do not see the gramma issue here, and have kept the text as it was.

4) Figure 6, caption, change "older waters west of" to "older waters east of"

Author's response:
Thank you for correcting. We have made the correction as suggested.

5) Figure 7, bottom left panel, in the heading change "Observed" to "Scavenged"

Author's response:
Thank you for spotting the mistake. We have corrected the mistake as suggested.

6) Figure S1, caption, change "profiles of and 13" to "profiles of Station 13"

Author's response:
Thank you for spotting the typo. We have corrected as suggested. We have also made it consistent by capitalizing S in the word station. This change is made in p10, line 10 in SI.

7) Section S4, providing information about sources of uncertainty in the assessment of scavenging, is very important. I recommend moving it to the Discussion section of the main text.

Author's response:
Thank you for your suggestion. We agree with the reviewer and have moved this information to the main text as a new short Section 4.5 in the discussion section of the main text. The sentence on p. 9, line 24 in the last revised manuscript indicating such information was in SI has been removed. Section 4.5 in the last revised manuscript is therefore renumbered as section 4.6. Accordingly, this information has been removed entirely from SI, and Supplemental Information S5 is renumbered as S4, and where it is referred is also corrected ( p11, line 9 in this revised manuscript).

8) Supplementary material, the Journal, Volume and page numbers is missing for the first reference.

Author's response:
Thank you for pointing out. We have made the correction.

Summary: As stated in my original review, the authors present a valuable new data set. The discussion of the sensitivity of Pa/Th ratios to preformed (initial) dissolved 231Pa and 230Th concentrations, to my knowledge, has not been presented before. This is a particularly useful discussion to help direct future studies of the Pa/Th ratio as a paleo circulation proxy when applied to North Atlantic sediments.

[revised manuscript text omitted]

**S1. Data of water-column [231]Pa, [230]Th and [232]Th concentrations, and [231]Pa/[230]Th ratios along GEOVIDE section and details of correction**

**Table S1  Water-column [231]Pa, [230]Th and [232]Th concentrations, and [231]Pa/[230]Th ratios along GEOVIDE section**

| Station | Depth m | [231]Pa fg/kg | [231]Pa$_{corr}$ fg/kg | [231]Pa$_{corr}$ µBq/kg | 2se | [230]Th fg/kg | [230]Th fg/kg | [230]Th$_{corr}$ µBq/kg | 2se | [232]Th pg/kg | [232]Th pmol/kg | 2se | [231]Pa/[230]Th | 2se |
|---|---|---|---|---|---|---|---|---|---|---|---|---|---|---|
| 1 | 3445.3 | 2.270 | 2.269 | 3.96 | 0.17 | | | | | | | | | |
| 40.33ºN | 2955.3 | 2.465 | 2.463 | 4.30 | 0.22 | | | | | | | | | |
| 10.04ºW | 2465.8 | 1.988 | 1.982 | 3.46 | 0.22 | 9.00 | 8.83 | 6.72 | 0.20 | 52.33 | 0.2248 | 0.0046 | 0.516 | 0.036 |
| | 1974.9 | 1.592 | 1.588 | 2.77 | 0.19 | 5.64 | 5.50 | 4.18 | 0.17 | 43.47 | 0.1867 | 0.0036 | 0.663 | 0.053 |
| | 1385.3 | 1.522 | 1.517 | 2.65 | 0.23 | 3.43 | 3.28 | 2.50 | 0.14 | 44.96 | 0.1931 | 0.0041 | 1.061 | 0.108 |
| | 1039.6 | 1.176 | 1.170 | 2.04 | 0.20 | 3.48 | 3.29 | 2.50 | 0.20 | 59.80 | 0.2568 | 0.0049 | 0.818 | 0.105 |
| | 495.6 | 0.861 | 0.853 | 1.49 | 0.22 | 3.67 | 3.43 | 2.61 | 0.16 | 72.77 | 0.3125 | 0.0058 | 0.571 | 0.090 |
| | 246.9 | 0.635 | 0.624 | 1.09 | 0.18 | 2.79 | 2.47 | 1.88 | 0.14 | 99.60 | 0.4278 | 0.0098 | 0.581 | 0.104 |
| | 49.6 | 0.641 | 0.623 | 1.09 | 0.20 | 2.84 | 2.29 | 1.74 | 0.18 | 171.16 | 0.7351 | 0.0133 | 0.625 | 0.132 |
| | 4.7 | 0.238 | 0.210 | 0.37 | 0.08 | 3.93 | 3.11 | 2.36 | 0.09 | 255.90 | 1.0991 | 0.0186 | 0.155 | 0.033 |
| 13 | 5330.4 | 2.093 | 1.880 | 3.28 | 0.24 | 8.69 | 8.62 | 6.56 | 6.56 | 22.79 | 0.0979 | 0.0017 | 0.501 | 0.039 |
| 41.38ºN | 5263.1 | 2.421 | 2.178 | 3.80 | 0.26 | 12.66 | 12.55 | 9.55 | 9.55 | 33.12 | 0.1422 | 0.0022 | 0.398 | 0.028 |
| 13.89ºW | 5194.3 | 2.751 | 2.747 | 4.80 | 0.24 | 15.80 | 15.67 | 11.92 | 11.92 | 39.61 | 0.1701 | 0.0033 | 0.403 | 0.023 |
| | 4903.9 | 2.013 | 2.009 | 3.51 | 0.15 | 14.79 | 14.68 | 11.17 | 11.17 | 35.27 | 0.1515 | 0.0031 | 0.314 | 0.016 |
| | 4417.8 | 2.627 | 2.624 | 4.58 | 0.24 | 11.16 | 11.07 | 8.42 | 8.42 | 28.70 | 0.1233 | 0.0026 | 0.545 | 0.033 |
| | 3444 | 2.399 | 2.396 | 4.19 | 0.14 | 8.15 | 8.08 | 6.14 | 6.14 | 22.33 | 0.0959 | 0.0022 | 0.681 | 0.031 |
| | 2464.7 | 1.569 | 1.565 | 2.73 | 0.26 | 5.77 | 5.67 | 4.31 | 4.31 | 30.57 | 0.1313 | 0.0028 | 0.634 | 0.066 |
| | 1187.3 | 0.869 | 0.865 | 1.51 | 0.12 | 3.24 | 3.12 | 2.38 | 2.38 | 37.16 | 0.1596 | 0.0032 | 0.637 | 0.065 |
| | 989.1 | 1.239 | 1.234 | 2.16 | 0.15 | 4.16 | 4.03 | 3.06 | 3.06 | 41.38 | 0.1777 | 0.0035 | 0.704 | 0.063 |
| | 248.4 | 0.602 | 0.602 | 1.05 | 0.11 | | | | | | | | | |
| | 148.5 | 0.525 | 0.525 | 0.92 | 0.13 | | | | | | | | | |
| | 29.7 | 0.508 | 0.507 | 0.89 | 0.11 | 0.44 | 0.43 | 0.32 | 0.15 | 5.60 | 0.0241 | 0.0015 | 2.736 | 1.315 |
| | 4.3 | 0.330 | 0.330 | 0.58 | 0.08 | 0.28 | 0.26 | 0.19 | 0.14 | 6.70 | 0.0288 | 0.0014 | 2.968 | 2.154 |

| Station | Depth | $^{231}$Pa | $^{231}$Pa | $^{231}$Pa$_{corr}$ | 2se | $^{230}$Th | $^{230}$Th | $^{230}$Th$_{corr}$ | 2se | $^{232}$Th | $^{232}$Th | 2se | $^{231}$Pa/$^{230}$Th | 2se |
|---|---|---|---|---|---|---|---|---|---|---|---|---|---|---|
| | m | fg/kg | fg/kg | µBq/kg | | fg/kg | fg/kg | µBq/kg | | pg/kg | pmol/kg | | | |
| 21 | 4514.9 | 2.605 | 2.603 | 4.55 | 0.15 | 6.09 | 6.03 | 4.58 | 0.19 | 20.99 | 0.0902 | 0.0017 | 0.992 | 0.052 |
| 46.54ºN | 4475 | 2.404 | 2.401 | 4.20 | 0.18 | 6.21 | 6.13 | 4.66 | 0.28 | 25.11 | 0.1079 | 0.0025 | 0.900 | 0.066 |
| 19.67ºW | 4426.3 | 2.413 | 2.411 | 4.21 | 0.17 | 5.76 | 5.70 | 4.33 | 0.14 | 19.57 | 0.0841 | 0.0017 | 0.972 | 0.051 |
| | 4279.6 | 2.497 | 2.495 | 4.36 | 0.18 | 5.91 | 5.85 | 4.45 | 0.19 | 20.48 | 0.0880 | 0.0018 | 0.980 | 0.059 |
| | 3929.3 | 2.081 | 2.079 | 3.63 | 0.17 | 5.47 | 5.41 | 4.11 | 0.16 | 20.11 | 0.0864 | 0.0016 | 0.884 | 0.053 |
| | 3443.3 | 1.930 | 1.928 | 3.37 | 0.20 | 4.63 | 4.57 | 3.47 | 0.14 | 18.16 | 0.0780 | 0.0015 | 0.970 | 0.070 |
| | 2268.9 | 1.114 | 1.114 | 1.95 | 0.22 | 5.16 | 5.06 | 3.85 | 0.17 | 30.49 | 0.1309 | 0.0021 | 0.506 | 0.060 |
| | 1482.7 | 0.995 | 0.993 | 1.73 | 0.12 | 2.48 | 2.39 | 1.82 | 0.14 | 26.83 | 0.1152 | 0.0016 | 0.953 | 0.097 |
| | 788.4 | 0.873 | 0.871 | 1.52 | 0.14 | 1.31 | 1.23 | 0.93 | 0.16 | 25.41 | 0.1092 | 0.0018 | 1.628 | 0.312 |
| | 445.5 | 0.679 | 0.676 | 1.18 | 0.14 | 2.18 | 2.05 | 1.56 | 0.14 | 39.91 | 0.1714 | 0.0023 | 0.756 | 0.115 |
| | 246.9 | 0.595 | 0.592 | 1.03 | 0.11 | 1.64 | 1.53 | 1.17 | 0.13 | 33.41 | 0.1435 | 0.0019 | 0.886 | 0.139 |
| | 98.1 | 0.607 | 0.603 | 1.05 | 0.14 | 1.32 | 1.23 | 0.93 | 0.12 | 28.01 | 0.1203 | 0.0019 | 1.129 | 0.209 |
| | 13.9 | 0.536 | 0.532 | 0.93 | 0.15 | | | | | 10.81 | 0.0464 | 0.0015 | | |
| | 3.7 | 0.522 | 0.519 | 0.91 | 0.14 | | | | | 7.95 | 0.0342 | 0.0014 | | |
| | | | | | | | | | | | | | | |
| 26 | 4116.3 | 1.162 | 1.158 | 2.02 | 0.18 | 3.43 | 3.29 | 2.50 | 0.17 | 42.15 | 0.1810 | 0.0035 | 0.808 | 0.088 |
| 50.28ºN | 2758.5 | 0.813 | 0.807 | 1.41 | 0.15 | 3.04 | 2.85 | 2.17 | 0.14 | 57.40 | 0.2465 | 0.0047 | 0.650 | 0.081 |
| 22.60ºW | 1973.5 | 0.745 | 0.739 | 1.29 | 0.13 | 2.83 | 2.65 | 2.01 | 0.16 | 56.40 | 0.2422 | 0.0045 | 0.642 | 0.081 |
| | 989.1 | 0.695 | 0.695 | 1.22 | 0.11 | | | | | | | | | |
| | 296.7 | 0.503 | 0.503 | 0.88 | 0.10 | | | | | | | | | |
| | 74.4 | 0.408 | 0.407 | 0.71 | 0.12 | 0.67 | 0.62 | 0.47 | 0.17 | 13.57 | 0.0583 | 0.0018 | 1.498 | 0.603 |

| Station | Depth m | $^{231}$Pa fg/kg | $^{231}$Pa fg/kg | $^{231}$Pa$_{corr}$ μBq/kg | 2se | $^{230}$Th fg/kg | $^{230}$Th fg/kg | $^{230}$Th$_{corr}$ μBq/kg | 2se | $^{232}$Th pg/kg | $^{232}$Th pmol/kg | 2se | $^{231}$Pa/$^{230}$Th | 2se |
|---|---|---|---|---|---|---|---|---|---|---|---|---|---|---|
| 32 | 3218.7 | 0.973 | 0.968 | 1.69 | 0.15 | 5.49 | 5.35 | 4.07 | 0.15 | 43.48 | 0.1867 | 0.0023 | 0.416 | 0.039 |
| 55.51ºN | 3218.5 | 0.984 | 0.982 | 1.72 | 0.12 | 1.68 | 1.63 | 1.24 | 0.14 | 15.25 | 0.0655 | 0.0016 | 1.387 | 0.184 |
| 26.71ºW | 3049.9 | 1.131 | 1.129 | 1.97 | 0.11 | 3.27 | 3.21 | 2.44 | 0.13 | 20.44 | 0.0878 | 0.0016 | 0.809 | 0.063 |
| | 2949.6 | 1.076 | 1.075 | 1.88 | 0.17 | 1.74 | 1.70 | 1.29 | 0.13 | 12.96 | 0.0557 | 0.0014 | 1.456 | 0.198 |
| | 2854.3 | 1.004 | 1.002 | 1.75 | 0.20 | 2.65 | 2.60 | 1.97 | 0.17 | 16.46 | 0.0707 | 0.0016 | 0.887 | 0.128 |
| | 2610.4 | 1.109 | 1.108 | 1.94 | 0.13 | 1.82 | 1.78 | 1.35 | 0.14 | 12.01 | 0.0516 | 0.0016 | 1.429 | 0.179 |
| | 2218.8 | 0.843 | 0.842 | 1.47 | 0.14 | 1.30 | 1.26 | 0.96 | 0.13 | 11.46 | 0.0492 | 0.0015 | 1.533 | 0.261 |
| | 1676.9 | 1.200 | 1.199 | 2.09 | 0.14 | 0.87 | 0.84 | 0.64 | 0.12 | 9.00 | 0.0387 | 0.0014 | 3.261 | 0.661 |
| | 1185.5 | 1.087 | 1.082 | 1.89 | 0.14 | 3.58 | 3.45 | 2.62 | 0.15 | 40.46 | 0.1738 | 0.0022 | 0.721 | 0.066 |
| | 890.5 | 0.872 | 0.869 | 1.52 | 0.14 | 1.64 | 1.55 | 1.18 | 0.13 | 29.00 | 0.1246 | 0.0018 | 1.289 | 0.187 |
| | 445.3 | 0.944 | 0.939 | 1.64 | 0.14 | 2.91 | 2.76 | 2.10 | 0.15 | 46.20 | 0.1984 | 0.0026 | 0.781 | 0.086 |
| | 222.9 | 0.568 | 0.564 | 0.99 | 0.12 | 1.70 | 1.57 | 1.20 | 0.13 | 39.69 | 0.1705 | 0.0022 | 0.824 | 0.135 |
| | 98.9 | 0.553 | 0.550 | 0.96 | 0.25 | 6.65 | 6.56 | 4.99 | 0.16 | 27.70 | 0.1190 | 0.0018 | 0.193 | 0.050 |
| | 29.5 | 0.613 | 0.610 | 1.07 | 0.27 | 5.50 | 5.41 | 4.12 | 0.16 | 27.73 | 0.1191 | 0.0019 | 0.259 | 0.067 |
| | 5.8 | 0.584 | 0.583 | 1.02 | 0.14 | | | | | 9.92 | 0.0426 | 0.0015 | | |
| 38 | 1338.1 | 1.604 | 1.602 | 2.80 | 0.22 | 1.62 | 1.57 | 1.20 | 0.15 | 16.07 | 0.0690 | 0.0020 | 2.341 | 0.338 |
| 58.84ºN | 1303.6 | 1.336 | 1.333 | 2.33 | 0.11 | 3.34 | 3.25 | 2.47 | 0.20 | 29.18 | 0.1253 | 0.0026 | 0.943 | 0.087 |
| 31.27ºW | 1234.5 | 1.088 | 1.085 | 1.90 | 0.11 | 3.42 | 3.33 | 2.53 | 0.15 | 28.39 | 0.1219 | 0.0026 | 0.748 | 0.064 |
| | 1084 | 0.863 | 0.860 | 1.50 | 0.10 | 3.64 | 3.54 | 2.69 | 0.19 | 31.67 | 0.1360 | 0.0027 | 0.558 | 0.053 |
| | 494.2 | 1.621 | 1.616 | 2.82 | 0.25 | 3.16 | 3.03 | 2.30 | 0.16 | 39.87 | 0.1712 | 0.0033 | 1.226 | 0.137 |
| | 198.4 | 1.258 | 1.253 | 2.19 | 0.12 | 3.12 | 2.97 | 2.26 | 0.14 | 46.64 | 0.2003 | 0.0037 | 0.969 | 0.081 |
| | 108.7 | 1.274 | 1.269 | 2.22 | 0.19 | 2.78 | 2.65 | 2.01 | 0.14 | 42.89 | 0.1842 | 0.0036 | 1.102 | 0.121 |
| | 20.3 | 0.836 | 0.835 | 1.46 | 0.13 | 0.84 | 0.80 | 0.61 | 0.13 | 11.88 | 0.0510 | 0.0016 | 2.389 | 0.566 |
| | 5.3 | 1.230 | 1.228 | 2.15 | 0.25 | 0.80 | 0.76 | 0.58 | 0.15 | 11.59 | 0.0498 | 0.0016 | 3.719 | 1.055 |

| Station | Depth | $^{231}$Pa | $^{231}$Pa | $^{231}$Pa$_{corr}$ | 2se | $^{230}$Th | $^{230}$Th | $^{230}$Th$_{corr}$ | 2se | $^{232}$Th | $^{232}$Th | 2se | $^{231}$Pa/$^{230}$Th | 2se |
|---|---|---|---|---|---|---|---|---|---|---|---|---|---|---|
| | m | fg/kg | fg/kg | µBq/kg | | fg/kg | fg/kg | µBq/kg | | pg/kg | pmol/kg | | | |
| 44 | 2918.9 | 0.839 | 0.837 | 1.46 | 0.17 | 2.40 | 2.33 | 1.77 | 0.13 | 22.85 | 0.0981 | 0.0022 | 0.827 | 0.112 |
| 59.62ºN | 2878.5 | 0.875 | 0.873 | 1.52 | 0.15 | 2.50 | 2.42 | 1.84 | 0.14 | 24.44 | 0.1050 | 0.0023 | 0.829 | 0.102 |
| 38.95ºW | 2829 | 0.723 | 0.723 | 1.26 | 0.22 | | | | | | | | | |
| | 2681.9 | 0.651 | 0.649 | 1.13 | 0.24 | 2.65 | 2.57 | 1.96 | 0.15 | 25.27 | 0.1085 | 0.0024 | 0.580 | 0.129 |
| | 2561 | 1.164 | 1.161 | 2.03 | 0.23 | 4.87 | 4.78 | 3.64 | 0.17 | 26.47 | 0.1137 | 0.0025 | 0.557 | 0.069 |
| | 2216.6 | 0.910 | 0.906 | 1.58 | 0.26 | 6.18 | 6.08 | 4.63 | 0.18 | 31.25 | 0.1342 | 0.0028 | 0.342 | 0.058 |
| | 1776 | 1.416 | 1.412 | 2.47 | 0.39 | 5.87 | 5.75 | 4.37 | 0.15 | 39.66 | 0.1704 | 0.0032 | 0.564 | 0.090 |
| | 1382.5 | 1.186 | 1.182 | 2.07 | 0.27 | 5.85 | 5.72 | 4.35 | 0.15 | 41.00 | 0.1761 | 0.0034 | 0.475 | 0.065 |
| | 1087.5 | 0.795 | 0.790 | 1.38 | 0.18 | 3.83 | 3.70 | 2.81 | 0.15 | 40.73 | 0.1749 | 0.0033 | 0.491 | 0.071 |
| | 593.2 | 1.116 | 1.111 | 1.94 | 0.36 | 4.28 | 4.13 | 3.14 | 0.16 | 46.81 | 0.2010 | 0.0038 | 0.618 | 0.118 |
| | 297.3 | 1.083 | 1.078 | 1.88 | 0.20 | 4.10 | 3.97 | 3.02 | 0.22 | 42.29 | 0.1816 | 0.0036 | 0.625 | 0.081 |
| | 78.8 | 0.929 | 0.925 | 1.62 | 0.19 | 3.44 | 3.31 | 2.52 | 0.14 | 39.69 | 0.1705 | 0.0033 | 0.641 | 0.082 |
| | 25.5 | 0.998 | 0.997 | 1.74 | 0.16 | 1.37 | 1.32 | 1.00 | 0.15 | 16.50 | 0.0709 | 0.0019 | 1.737 | 0.297 |
| | 5.5 | 0.722 | 0.720 | 1.26 | 0.33 | 1.31 | 1.27 | 0.96 | 0.12 | 14.99 | 0.0644 | 0.0018 | 1.307 | 0.380 |
| | | | | | | | | | | | | | | |
| 60 | 1710.9 | 0.844 | 0.841 | 1.47 | 0.07 | 3.77 | 3.67 | 2.79 | 0.16 | 31.42 | 0.1349 | 0.0031 | 0.526 | 0.039 |
| 59.80ºN | 1652.6 | 0.881 | 0.878 | 1.53 | 0.07 | 4.03 | 3.92 | 2.98 | 0.16 | 33.28 | 0.1429 | 0.0032 | 0.514 | 0.036 |
| 42.01ºW | 1603.1 | 0.845 | 0.842 | 1.47 | 0.07 | 4.10 | 3.99 | 3.04 | 0.24 | 33.74 | 0.1449 | 0.0032 | 0.485 | 0.045 |
| | 1481.2 | 0.813 | 0.809 | 1.41 | 0.07 | 4.19 | 4.08 | 3.10 | 0.17 | 33.94 | 0.1458 | 0.0032 | 0.455 | 0.034 |
| | 989.7 | 1.087 | 1.082 | 1.89 | 0.08 | 5.29 | 5.15 | 3.91 | 0.20 | 45.60 | 0.1958 | 0.0040 | 0.483 | 0.031 |
| | 495.5 | 0.911 | 0.906 | 1.58 | 0.16 | 3.69 | 3.56 | 2.71 | 0.14 | 39.63 | 0.1702 | 0.0033 | 0.585 | 0.064 |
| | 247.9 | 0.851 | 0.847 | 1.48 | 0.23 | 3.16 | 3.03 | 2.30 | 0.14 | 40.61 | 0.1744 | 0.0034 | 0.643 | 0.109 |
| | 99.1 | 0.826 | 0.822 | 1.44 | 0.20 | 2.34 | 2.23 | 1.69 | 0.13 | 34.75 | 0.1492 | 0.0029 | 0.849 | 0.134 |
| | 19.5 | 0.887 | 0.885 | 1.55 | 0.32 | 1.36 | 1.30 | 0.99 | 0.16 | 18.14 | 0.0779 | 0.0020 | 1.561 | 0.405 |
| | 3.7 | 0.829 | 0.827 | 1.44 | 0.24 | 1.53 | 1.46 | 1.11 | 0.14 | 22.32 | 0.0959 | 0.0023 | 1.301 | 0.266 |

| Station | Depth | $^{231}$Pa | $^{231}$Pa | $^{231}$Pa$_{corr}$ | 2se | $^{230}$Th | $^{230}$Th | $^{230}$Th$_{corr}$ | 2se | $^{232}$Th | $^{232}$Th | 2se | $^{231}$Pa/$^{230}$Th | 2se |
|---|---|---|---|---|---|---|---|---|---|---|---|---|---|---|
| | m | fg/kg | fg/kg | µBq/kg | | fg/kg | fg/kg | µBq/kg | | pg/kg | pmol/kg | | | |
| 64 | 2466.8 | 0.759 | 0.756 | 1.32 | 0.19 | 3.24 | 3.16 | 2.40 | 0.14 | 25.71 | 0.1104 | 0.0025 | 0.551 | 0.087 |
| 59.07ºN | 2423.6 | 0.828 | 0.825 | 1.44 | 0.20 | 3.71 | 3.62 | 2.76 | 0.18 | 26.12 | 0.1122 | 0.0027 | 0.523 | 0.080 |
| 46.08ºW | 2374 | 0.958 | 0.955 | 1.67 | 0.19 | 3.76 | 3.68 | 2.80 | 0.13 | 27.16 | 0.1166 | 0.0025 | 0.597 | 0.074 |
| | 2226.6 | 1.051 | 1.048 | 1.83 | 0.27 | 5.33 | 5.24 | 3.99 | 0.15 | 29.28 | 0.1257 | 0.0028 | 0.460 | 0.070 |
| | 1775.6 | 1.155 | 1.152 | 2.01 | 0.20 | 5.50 | 5.38 | 4.10 | 0.17 | 34.64 | 0.1488 | 0.0038 | 0.491 | 0.054 |
| | 890 | 1.112 | 1.106 | 1.93 | 0.19 | 4.56 | 4.39 | 3.34 | 0.19 | 53.17 | 0.2283 | 0.0045 | 0.579 | 0.066 |
| | 395.2 | 0.898 | 0.894 | 1.56 | 0.17 | 3.61 | 3.48 | 2.65 | 0.18 | 40.79 | 0.1752 | 0.0110 | 0.590 | 0.074 |
| | 247.4 | 0.923 | 0.918 | 1.60 | 0.26 | 3.23 | 3.10 | 2.36 | 0.18 | 40.00 | 0.1718 | 0.0038 | 0.680 | 0.122 |
| | 99.2 | 1.044 | 1.039 | 1.81 | 0.23 | 3.45 | 3.30 | 2.51 | 0.16 | 46.12 | 0.1981 | 0.0041 | 0.723 | 0.105 |
| | 29.5 | 0.963 | 0.960 | 1.68 | 0.28 | 2.18 | 2.08 | 1.58 | 0.14 | 30.23 | 0.1298 | 0.0029 | 1.060 | 0.200 |
| | 5.1 | 0.768 | 0.766 | 1.34 | 0.21 | 1.54 | 1.46 | 1.11 | 0.17 | 23.73 | 0.1019 | 0.0024 | 1.202 | 0.269 |
| | | | | | | | | | | | | | | |
| 69 | 3676.5 | 0.404 | 0.402 | 0.70 | 0.10 | 1.94 | 1.88 | 1.43 | 0.21 | 19.60 | 0.0842 | 0.0025 | 0.491 | 0.099 |
| 55.84ºN | 3637.3 | 0.336 | 0.334 | 0.58 | 0.12 | 1.92 | 1.86 | 1.41 | 0.16 | 19.15 | 0.0822 | 0.0023 | 0.414 | 0.098 |
| 48.09ºW | 3589.5 | 0.438 | 0.436 | 0.76 | 0.11 | 2.15 | 2.08 | 1.58 | 0.19 | 21.39 | 0.0919 | 0.0024 | 0.482 | 0.091 |
| | 3444.7 | 0.528 | 0.525 | 0.92 | 0.11 | 3.58 | 3.49 | 2.65 | 0.19 | 27.76 | 0.1192 | 0.0028 | 0.346 | 0.048 |
| | 2951.8 | 0.581 | 0.578 | 1.01 | 0.11 | 5.75 | 5.64 | 4.29 | 0.23 | 33.80 | 0.1452 | 0.0032 | 0.235 | 0.029 |
| | 2462.6 | 0.604 | 0.599 | 1.05 | 0.11 | 6.44 | 6.31 | 4.80 | 0.21 | 39.48 | 0.1696 | 0.0037 | 0.218 | 0.025 |
| | 2168.1 | 0.855 | 0.850 | 1.49 | 0.12 | 6.03 | 5.90 | 4.49 | 0.22 | 40.95 | 0.1759 | 0.0039 | 0.331 | 0.031 |
| | 1481.3 | 0.670 | 0.661 | 1.16 | 0.10 | 4.98 | 4.73 | 3.60 | 0.20 | 79.22 | 0.3402 | 0.0066 | 0.321 | 0.033 |
| | 989 | 0.729 | 0.723 | 1.26 | 0.11 | 4.53 | 4.35 | 3.31 | 0.21 | 57.18 | 0.2456 | 0.0049 | 0.382 | 0.041 |
| | 445.6 | 0.692 | 0.686 | 1.20 | 0.07 | 4.23 | 4.05 | 3.08 | 0.23 | 55.48 | 0.2383 | 0.0047 | 0.389 | 0.037 |
| | 248.8 | 0.604 | 0.598 | 1.04 | 0.07 | 4.06 | 3.87 | 2.95 | 0.18 | 59.43 | 0.2552 | 0.0052 | 0.355 | 0.033 |
| | 99.5 | 0.513 | 0.507 | 0.89 | 0.10 | 3.58 | 3.40 | 2.58 | 0.22 | 56.06 | 0.2408 | 0.0049 | 0.343 | 0.047 |
| | 28.7 | 0.554 | 0.550 | 0.96 | 0.07 | 1.50 | 1.40 | 1.07 | 0.24 | 30.11 | 0.1293 | 0.0029 | 0.903 | 0.210 |
| | 8.2 | 0.457 | 0.453 | 0.79 | 0.07 | 1.42 | 1.31 | 0.99 | 0.17 | 34.28 | 0.1472 | 0.0031 | 0.797 | 0.153 |

| Station | Depth | $^{231}$Pa | $^{231}$Pa | $^{231}$Pa$_{corr}$ | 2se | $^{230}$Th | $^{230}$Th | $^{230}$Th$_{corr}$ | 2se | $^{232}$Th | $^{232}$Th | 2se | $^{231}$Pa/$^{230}$Th | 2se |
|---|---|---|---|---|---|---|---|---|---|---|---|---|---|---|
| | m | fg/kg | fg/kg | µBq/kg | | fg/kg | fg/kg | µBq/kg | | pg/kg | pmol/kg | | | |
| 77 | 2487.5 | | | | | 4.02 | 3.93 | 2.99 | 0.14 | 29.43 | 0.1264 | 0.0026 | | |
| 53ºN | 2462.8 | 1.261 | 1.258 | 2.20 | 0.15 | 4.07 | 3.97 | 3.02 | 0.14 | 30.06 | 0.1291 | 0.0026 | 0.728 | 0.059 |
| 51.10ºW | 2414.2 | 1.113 | 1.110 | 1.94 | 0.10 | 4.09 | 4.00 | 3.04 | 0.15 | 29.56 | 0.1270 | 0.0026 | 0.638 | 0.044 |
| | 2268.5 | 0.989 | 0.985 | 1.72 | 0.12 | 6.11 | 6.00 | 4.56 | 0.16 | 33.99 | 0.1460 | 0.0030 | 0.377 | 0.029 |
| | 2170.5 | 1.355 | 1.352 | 2.36 | 0.13 | 5.32 | 5.22 | 3.97 | 0.17 | 30.77 | 0.1322 | 0.0028 | 0.595 | 0.041 |
| | 1678.1 | 1.267 | 1.263 | 2.21 | 0.13 | 5.48 | 5.36 | 4.08 | 0.16 | 37.02 | 0.1590 | 0.0031 | 0.541 | 0.039 |
| | 1235.4 | 0.580 | 0.576 | 1.01 | 0.13 | 5.36 | 5.22 | 3.97 | 0.18 | 44.75 | 0.1922 | 0.0037 | 0.253 | 0.035 |
| | 989.6 | 1.772 | 1.767 | 3.09 | 0.23 | 4.96 | 4.79 | 3.64 | 0.17 | 53.23 | 0.2286 | 0.0041 | 0.848 | 0.073 |
| | 496.4 | | | | | 4.33 | 4.15 | 3.16 | 0.17 | 55.68 | 0.2391 | 0.0044 | | |
| | 297.9 | 0.501 | 0.494 | 0.86 | 0.07 | 4.71 | 4.50 | 3.42 | 0.23 | 64.41 | 0.2766 | 0.0055 | 0.252 | 0.026 |
| | 78.8 | 1.200 | 1.193 | 2.08 | 0.14 | 3.50 | 3.29 | 2.50 | 0.20 | 65.47 | 0.2812 | 0.0053 | 0.834 | 0.086 |
| | 2.5 | 0.926 | 0.923 | 1.61 | 0.12 | 1.34 | 1.25 | 0.95 | 0.17 | 28.01 | 0.1203 | 0.0026 | 1.699 | 0.332 |

$^{230}$Th and $^{231}$Pa are dissolved $^{230}$Th and $^{231}$Pa activities corrected for the ingrowth from seawater $^{234}$U and $^{235}$U, respectively, since the time of collection following equations:

$$^{230}Th = {^{230}Th_m} - {^{234}U} \times \left(1 - \exp\left(-\lambda_{^{230}Th} \times t\right)\right) \quad (1)$$

$$^{231}Pa = {^{231}Pa_m} - {^{235}U} \times \left(1 - \exp\left(-\lambda_{^{231}Pa} \times t\right)\right) \quad (2)$$

$^{230}$Th and $^{231}$Pa are further corrected for detrital, U-supported $^{230}$Th and $^{231}$Pa concentrations as follows:

$$^{230}Th_{corr} = {^{230}Th_m} - \left(0.6 \times {^{232}Th_m}\right) \quad (3)$$

$$^{231}Pa_{corr} = {^{231}Pa_m} - 0.046 \times \left(0.6 \times {^{232}Th_m}\right) \quad (4)$$

where $^{230}$Th$_m$, $^{231}$Pa$_m$ and $^{232}$Th$_m$ are activities obtained from measurement; $^{235}$U and $^{234}$U are their average activities in seawater, 1824 µBq/kg (112 dpm/1000l) and 45551µBq/kg (2801 dpm/1000l,) respectively, obtained from $^{238}$U activity of 39610 µBq/kg (2436 dpm/1000l) at salinity of 35 (Owens et al., 2011) and assuming natural $^{238}$U/$^{235}$U abundance ratio of 137.88 and seawater $^{234}$U/$^{238}$U activity ratio of 1.15; $\lambda_{^{230}Th}$ and $\lambda_{^{231}Pa}$ are decay constants of $^{230}$Th and $^{231}$Pa; t is the time between sample collection and chemical separation of U from $^{231}$Pa and $^{230}$Th. 0.6 is the average $^{238}$U/$^{232}$Th activity ratio in detrital material in the Atlantic (Henderson and Anderson, 2003) and 0.046 represents $^{235}$U/$^{238}$U activity ratio in seawater (Anderson et al., 1990). Half-lives for $^{231}$Pa, $^{230}$Th and $^{232}$Th are 32,760 yr, 75,584 yr and 1.40×10$^{10}$ yr (Cheng et al., 2013; Holden, 1990; Robert et al., 1969).

All errors are two standard errors (2se) and include the contribution from sample weighing, spike calibration, $^{231}$Pa, $^{230}$Th and $^{232}$Th in the respective $^{233}$Pa and $^{230}$Th spikes, blank correction, internal precision and related corrections of mass spectrometric measurement.

**S2. CFC-based Age determination**

5    CFC measurement are not available for the GEOVIDE cruise itself. However, with the availability of CFC measurements from OVIDE section in 2012 and water mass composition estimated using extended Optimum Multi-Parameter (eOMP) analysis for both OVIDE and GEOVIDE sections, CFC-based ages can be derived for GEOVIDE section.

1. CFC measurements were available along OVIDE section in 2012 (OVIDE/CATARINA cruise) (de la Paz et al., 2017). This allows the computing of the mean age of water masses using transient time distribution (TTD) method. A more detailed

10    description of TTD method is given in other studies (e.g. Steinfeldt et al., 2009; Waugh et al., 2003). It is important to note that this mean age (referred to as CFC-based age hereafter and in the manuscript) is different from the age calculated based on atmospheric history of CFC (referred to as CFC apparent age hereafter), and therefore is not limited by the time span of the presence of CFC in the atmosphere and inherently deals with age bias due to water mass mixing in CFC apparent age. Combining CFC-based ages computed with the TTD method for each water sample with water mass composition estimated

15    using eOMP analysis for OVIDE section in 2012 (García-Ibáñez et al., 2015), CFC-based age was calculated for each Source Water Type (SWT) defined in García-Ibáñez et al., 2015 by the equations,

$$\log [\text{CFC–based age}]^j = \sum_{i=1}^{12} SWT_i^j \times (log[\text{CFC–based age}]_i) + \varepsilon_j \ j = 1 \rightarrow 424 \ samples \quad (5)$$

$$[\text{CFC} - \text{Age}]^j = \text{anti} \log [\text{CFC} - \text{Age}]^j \ (6)$$

where $SWT_i^j$ is the fraction of SWT "$i$" to sample "$j$" (obtained through the eOMP analysis); [CFC–based age]$^j$ is CFC-

20    based age for each water sample computed with TTD method along OVIDE section 2012; and $\varepsilon_j$ is the residual, representing the portion of CFC–based age that can not be modelled by mixing of SWTs, i.e. the difference between $\log [\text{CFC–based age}]^j$ and that obtained as the sum of the contributions by mixing of the individual SWT, $\sum_{i=1}^{12} SWT_i^j \times$ ($log[\text{CFC–based age}]_i$).

The output of log[CFC-based age]$_i$ and its inversion ([CFC-age]$_i$) is given in Table S2. The squared correlation coefficient

25    ($r^2$) and standard deviation of the residual, $\varepsilon_j$, are 0.94 and 0.12, respectively.

2. CFC-based age for GEOVIDE section was then calculated employing equation (5) with water mass composition estimated using eOMP analysis along GEOVIDE section (García-Ibáñez et al., 2018) and the output of CFC-based age for SWT (Table S2).

**Table S2 Output of log(CFC-age) and the inversion [CFC-age]$_I$ (i.e. CFC-based age) for source water types (SWT)**

|  | log(CFC-age) | CFC-based age |
|---|---|---|
| **ENACW$_{16}$** | 1.05±0.20 | 11±5 |
| **ENACW$_{12}$** | 1.11±0.03 | 13±1 |
| **SPMW$_8$** | 1.69±0.04 | 49±5 |
| **SAIW** | 1.19±0.07 | 16±3 |
| **SPMW$_7$** | 1.26±0.05 | 18±2 |
| **IrSPMW** | 0.98±0.03 | 10±1 |
| **LSW** | 1.54±0.02 | 35±1 |
| **MW** | 1.96±0.04 | 91±8 |
| **PIW** | 1.33±0.15 | 22±8 |
| **DSOW** | 1.22±0.07 | 17±3 |
| **ISOW** | 1.70±0.03 | 50±4 |
| **NEADW$_L$** | 3.00±0.02 | 989±48 |
| r$^2$ | 0.94 | 0.943 |
| std(Resid) | 0.12 | 41 |

ENACW$_{16}$ and ENACW$_{12}$ = East North Atlantic Central Water of 16ºC and 12ºC; SPMW$_8$, SPMW$_7$, IrSPMW = Subpolar Mode Water of 8ºC, 7ºC and of the Irminger Sea; SAIW = Subarctic Intermediate Water; MW = Mediterranean Water; PIW = Polar Intermediate Water; ISOW=Iceland–Scotland Overflow Water; LSW=Labrador Sea Water; DSOW: Denmark Strait Overflow Waters; and NEADW$_L$: Lower
5  North East Atlantic Deep Water. r$^2$ and std (Resid) represents the squared correlation coefficient and standard deviation of the residual, $\varepsilon_j$, i.e. the difference between log [CFC–based age]$^j$ and that obtained as the sum of the contributions by mixing of the individual SWT, $\sum_{i=1}^{12} SWT_i^j \times (log[\text{CFC–based age}]_i)$.

**S3. Scavenging-mixing model and parameterization**

A more detailed description of the scavenging-mixing model used in this this study is given in Moran et al. (1997). Briefly,
10  the model takes into account reversible scavenging of the nuclides and water mass mixing. It describes the evolution of nuclides through time in a one-dimensional system, an ocean water column. In the Atlantic, the system is assumed to start at time t=0 in the far North Atlantic and moves southward with time. Transport of material downward relative to the direction of water flow is permitted, to represent the effect of scavenging of radionuclides by sinking particles. Lateral exchange with water outside of the system is not permitted.

15  The equations to derive the dissolved concentration of each nuclide follows those of Moran et al., (1997). Dissolved concentration of the nuclide is given by,

$$c_d = \frac{C_{pre,t} + P \quad \tau_w}{(K_d \ SPM + 1)} \times [1 - \exp(-\frac{(K_d \ SPM + 1)}{SK_d \ \tau_w SPM} \times z)] \quad (7)$$

where $C_d$ is the dissolved concentration of the nuclide ; $P$ is the production rate of [230]Th and [231]Pa, 0.42 µBq/kg/yr (2.57 × 10[-2] dpm/1000l/yr ) and 0.039 µBq/kg/yr (2.37 × 10[-3] dpm/1000l/yr), respectively; $K_d$ is the distribution coefficient of the nuclide; λ is the decay constant of the nuclide; $C_{pre,t}$ is the preformed total concentration of [230]Th (or [231]Pa); SPM is the suspended particle concentration; and S is the particle settling speed, which represents the net effect of particle sinking, disaggregation and aggregation; $\tau_w$ is water mass age; z is the water depth.

Initial parameterization was conducted using S=500-1000 m/yr, $K_d^{Th}$= 1× 10[7] ml/g, $K_d^{Pa}$= 5× 10[5] ml/g, SPM= 20-50 µg/l, for preformed concentrations set at 0 and surface average from GEOVIDE, i.e., $C_{pre}^{Th} = C_{surface\ average}^{Th}$ = 1.66 µBq/kg, $C_{pre}^{Pa}$=$C_{surface\ average}^{Pa}$ = 1.31 µBq/kg. With $\tau_w$ known from CFC measurements for every depth where [230]Th and [231]Pa was measured along GEOVIDE section, water-column profiles of both nuclides were simulated for GEOVIDE Station 13 and the parameters were adjusted for the best fit between the simulated and observed profiles (Fig. S1). This gives us the optimized parameters for the analysis in discussion section 4.3, which are listed in Table S3. Our optimized parameters are consistent with values reported by other studies (also listed in Table S3).

Adopting the optimized parameters and setting preformed component ($C_{pre}$) equal to the nuclide concentrations observed in the upper 100 m of the GEOVIDE section, the modelled evolution of nuclide concentrations with age between 0-500 years at 2000 m and 3500 m water depths, together with GEOVIDE data, is plotted in Figure S2.

[Figure]

**Figure S1: Modelled (dashed black lines) profiles with preformed value set at 0 and surface average concentration from GEOVIDE, and observed (solid blue lines) profiles of Station 13 from GEOVIDE section. The best fit was first sought for [230]Th, which gives us the optimized parameters S, SPM and $K_d^{Th}$. These parameters were then adopted for the simulation of [231]Pa profiles, adjusting only $K_d^{Pa}$ to obtain the best fit.**

[Figure]

**Figure S2: Results from a scavenging-mixing model of** $^{230}$**Th,** $^{231}$**Pa, Dissolved** $^{231}$**Pa/**$^{230}$**Th and Particulate** $^{231}$**Pa/**$^{230}$**Th compared to observations. Preformed concentration (C**$_{pre}$**) were set at 0 (dashed line) and at the average surface concentration (C**$_{surface\ average}$**) from GEOVIDE section (solid line), i.e.** $^{230}$**Th= 1.66 µBq/kg,** $^{231}$**Pa= 1.31 µBq/kg.**

**Table S3 Parameterization of the scavenging-mixing model**

|  | This study | Literature |
|---|---|---|
| S (m/yr) | 800 | 500-1000 (Moran et al., 1997) |
| SPM (μg/l) | 25 | 30 (Labrador Sea, Brewer et al., 1976) |
| $K_d^{Th}$ (ml/g) | $1.1 \times 10^7$ | $1.1 \times 10^7$ (Moran et al., 1997) |
| $K_d^{Pa}$ (ml/g) | $1.4 \times 10^6$ | $2.2 \times 10^5$ (pure carbonate)~$1.4 \times 10^6$ g/g (pure opal) (pseudo-$K_d$, Chase et al., 2002) |

**S4. Meridional transport of $^{230}$Th and $^{231}$Pa in the Atlantic**

**Table S4 Mass balance calculation of meridional transport of $^{230}$Th and $^{231}$Pa in the Atlantic**

|  | Net meridional transport $\times 10^{10}$ μBq/s | | Volume of seawater between two latitudes $\times 10^{17}$ m³ | Production in water column $\times 10^{10}$ μBq/s | | Removal to sediment $\times 10^{10}$ μBq/s | | Removal/Production % | |
|---|---|---|---|---|---|---|---|---|---|
|  | $^{230}$Th | $^{231}$Pa |  | $^{230}$Th | $^{231}$Pa | $^{230}$Th | $^{231}$Pa | $^{230}$Th | $^{231}$Pa |
| GEOVIDE-4.5ºS | -7.76 | -4.75 | 1.48 | 200.8 | 18.6 | 193.1 | 13.8 | 96.2 | 74.2 |
| 4.5ºS-45ºS | 0.33 | 0.017 | 1.02 | 138.4 | 12.8 | 138.7 | 12.8 | 99.8 | 100.0 |

Positive value indicates northward transport; negative value indicates southward transport. Production rate of $^{230}$Th and $^{231}$Pa in water column are 0.42 and 0.039 μBq/kg/yr, respectively.

---

## Author Response (AR3)

**"Evolution of 231Pa and 230Th in overflow waters of the North Atlantic"**
**by Feifei Deng et al.**

**Response to referees**

We would like to thank all referees and the editor for their time reading the revised manuscript and giving constructive suggestions to further improve the paper. We are pleased that the referees appreciate our clarification and revision of the initial manuscript, and have accepted the manuscript for publication.

As we were preparing for the final production files, we made changes to the accepted manuscript to meet the requirements for publication as outlined below. The following page and line numbers refer to those in the accepted version of the revised manuscript.

1. We added data availability (and the reference: Deng, F., Henderson, G., Castrillejo, M., Browning, T.J.: GEOTRACES GA01 (cruise GEOVIDE) $^{231}$Pa, $^{230}$Th and $^{232}$Th activities, British Oceanographic Data Centre - Natural Environment Research Council, UK, doi:10/cws2, 2018.), author contributions, competing interests, and special issue statements.

2. p.2, line18, change 'ceased' to 'ceases'.

3. p.3, line 5, change 'suggested' to 'have suggested'.

4. p.6, line25, change 'west' to 'east'.

5. p.7, line 24, change '$^{238}/^{235}$U' to '$^{238}$U/$^{235}$U'.

6. p.11, line 13, change 'Use' to 'Using'.

7. We added the reviewers, editor and Jerry McManus who have helped greatly to improve the paper in the acknowledgement section.

8. We replotted Figure 9 so that major units of x-axis for all panels are the same. In the original figure of the accepted manuscript, the major units of x-axis for Fig. 9 (a) and (b) (being 5) do not match those of (c) and (d) (being 10).

[revised manuscript text omitted]